# Implicit Regularization or Implicit Conditioning? Exact Risk Trajectories of SGD in High Dimensions

**Courtney Paquette**
Google Research, Brain and McGill University
courtney.paquette@mcgill.ca

**Elliot Paquette**
McGill University
elliot.paquette@mcgill.ca

**Ben Adlam**
Google Research, Brain

**Jeffrey Pennington**
Google Research, Brain

## Abstract

Stochastic gradient descent (SGD) is a pillar of modern machine learning, serving as the go-to optimization algorithm for a diverse array of problems. While the empirical success of SGD is often attributed to its computational efficiency and favorable generalization behavior, neither effect is well understood and disentangling them remains an open problem. Even in the simple setting of convex quadratic problems, worst-case analyses give an asymptotic convergence rate for SGD that is no better than full-batch gradient descent (GD), and the purported implicit regularization effects of SGD lack a precise explanation. In this work, we study the dynamics of multi-pass SGD on high-dimensional convex quadratics and establish an asymptotic equivalence to a stochastic differential equation, which we call homogenized stochastic gradient descent (HSGD), whose solutions we characterize explicitly in terms of a Volterra integral equation. These results yield precise formulas for the learning and risk trajectories, which reveal a mechanism of implicit conditioning that explains the efficiency of SGD relative to GD. We also prove that the noise from SGD negatively impacts generalization performance, ruling out the possibility of any type of implicit regularization in this context. Finally, we show how to adapt the HSGD formalism to include streaming SGD, which allows us to produce an exact prediction for the excess risk of multi-pass SGD relative to that of streaming SGD (bootstrap risk).

## 1 Introduction

Stochastic gradient descent (SGD) is the algorithm of choice for optimization in modern machine learning and has been hailed as a major reason for deep learning's success [11, 21]. Explanations for the effectiveness of SGD typically refer to its computational efficiency and to its favorable generalization properties, but theoretical understanding of these purported benefits is far from complete.

The efficiency of SGD has been the subject of extensive research, dating back to the original work of Robbins and Monro [68] and extending to modern large-scale machine learning applications (see e.g. [10, 12]). However, despite its widespread adoption and algorithmic simplicity, surprisingly little is known about how SGD performs in the types of high-dimensional optimization problems that occur in practice. Part of the challenge in deriving robust high-level conclusions about the efficiency of SGD is simply that those conclusions can depend on precisely which quantities are measured and what assumptions are leveraged. For example, in the extreme setting where the samples are one-hot vectors, running SGD on a quadratic function is actually identical to running full-batch gradient descent; as such, any statements about the two algorithms' relative efficiency must be data-dependent.

Furthermore, the majority of prior analyses focus on the streaming or single-pass setting, where each sample is seen a single time. While this setting is appropriate when the number of samples $n$ is much larger than the dimensionality $d$, it does not adequately describe the practically-relevant overparameratized or high-dimensional settings where $d \gtrsim n$.

Moreover, the practical success of SGD has been so remarkable in recent years that a growing body of literature has suggested that its benefit to generalization extends beyond what any improved efficiency might reasonably afford [76, 31, 14, 72, 74]. Some of the myriad explanations for SGD's favorable generalization properties include the local geometry of minimizers [32, 26, 82, 24], connections to approximate Bayesian inference [49], and the regularization properties of noise [75], among many others. While some of the these perspectives are intuitive and compelling, they are often difficult to rigorously establish from either an empirical or a theoretical perspective. Empirically, simulations at large scale command significant computational resources, and it can be challenging to push to sufficiently late times or sufficiently large batches to establish the appropriate baselines [72, 75]. Theoretically, the strongest existing results are again in the single-pass setting, for which a number of works have established excess risk bounds for quadratic problems [6, 18, 20, 81]. Much less is known in the multi-pass setting, though stability results were established by [25], and some recent works have begun examining generalization [39].

In this work, we study the dynamics of multi-pass SGD on high-dimensional convex quadratic functions and derive exact asymptotic predictions for the learning and risk trajectories. Our analysis establishes an asymptotic equivalence to a stochastic differential equation, which we call homogenized stochastic gradient descent (HSGD), whose solutions we characterize explicitly in terms of a Volterra integral equation. These results allow us to define a precise data-dependent implicit-conditioning ratio (ICR) that determines whether SGD is more efficient than its full-batch cousins. The ICR favors SGD for many practical datasets, providing some explanation for the observed superior efficiency of SGD; interestingly, we also highlight settings for which SGD is less efficient than full-batch momentum gradient descent, underscoring the data-dependence of the conclusions. Moreover, our results also show that SGD does not improve generalization performance, whether measured in-distribution or out-of-distribution, and therefore that SGD does not offer any form of implicit regularization in this setting. We emphasize that our results do not rule out possible benefits for non-convex problems, but they do provide some of the first explicit negative results in the convex quadratic case.

## 1.1 Contributions

Our primary contributions are to:

1. Establish the equivalence of quadratic statistics computed on the iterates of SGD and on a particular stochastic Langevin diffusion process called homogonized SGD (Theorem 1);

2. Exactly characterize the asymptotic training and risk trajectories as the solutions of a deterministic Volterra integral equation (Theorem 2);

3. Prove that the noise from SGD negatively impacts generalization performance, both in- and out-of-distribution (Section 3.1), but explain why the impact is often minimal in practice;

4. Introduce the implicit-conditioning ratio that describes when and by how much SGD accelerates convergence relative to the best full-batch methods (Section 3.2);

5. Analyze the limit of streaming SGD to show its inability to capture many salient features of the dynamics of multi-pass SGD (Appendix C).

## 2 Preliminaries and background

**Problem setting.** We consider high-dimensional $\ell^2$-regularized least squares problems defined by,

$$\min_{\boldsymbol{x} \in \mathbb{R}^d} \left\{ f(\boldsymbol{x}) \stackrel{\text{def}}{=} \frac{1}{2} \|\boldsymbol{A}\boldsymbol{x} - \boldsymbol{b}\|_2^2 + \frac{\delta}{2}\|\boldsymbol{x}\|^2 = \sum_{i=1}^n \underbrace{\frac{1}{2}\left( (\boldsymbol{a}_i\boldsymbol{x} - b_i)^2 + \frac{\delta}{n}\|\boldsymbol{x}\|^2 \right)}_{\stackrel{\text{def}}{=} f_i(\boldsymbol{x})} \right\}, \tag{1}$$

where $\delta \geq 0$ is the ridge-regularization parameter. We denote the ridgeless empirical risk as

$$\mathcal{L}(\boldsymbol{x}) \stackrel{\text{def}}{=} \frac{1}{2}\|\boldsymbol{A}\boldsymbol{x} - \boldsymbol{b}\|_2^2. \tag{2}$$

On the problem (1), the steps taken by gradient decent (GD) can be written recursively as

$$\boldsymbol{x}_{k+1}^{\text{m-gd}} = \boldsymbol{x}_k^{\text{m-gd}} - \gamma_k \nabla f(\boldsymbol{x}_k^{\text{m-gd}}) = \boldsymbol{x}_k^{\text{m-gd}} - \gamma_k \boldsymbol{A}^T(\boldsymbol{A}\boldsymbol{x}_k^{\text{m-gd}} - \boldsymbol{b}) - \gamma_k \delta \boldsymbol{x}_k^{\text{m-gd}} + \Delta(\boldsymbol{x}_k^{\text{m-gd}} - \boldsymbol{x}_{k-1}^{\text{m-gd}}), \quad (3)$$

where $\Delta > 0$ is the momentum parameter, $\gamma_k$ is the learning rate schedule, and $\boldsymbol{x}_0 \in \mathbb{R}^d$ is an initial vector assumed to be independent of all other randomness and having norm at most 1. When $A$ is large, computing these updates can be expensive, so an unbiased estimator for the true gradient is often used, where a subset of the data points are selected uniformly at random. We focus on the setting with batch size equal to one and without momentum, which we refer to as stochastic gradient descent (SGD), and for which the iterates can be written recursively as

$$\boldsymbol{x}_{k+1}^{\text{sgd}} = \boldsymbol{x}_k^{\text{sgd}} - \gamma_k \nabla f_{i_k}(\boldsymbol{x}_k^{\text{sgd}}) = \boldsymbol{x}_k^{\text{sgd}} - \gamma_k \boldsymbol{A}^T \boldsymbol{e}_{i_k} \boldsymbol{e}_{i_k}^T(\boldsymbol{A}\boldsymbol{x}_k^{\text{sgd}} - \boldsymbol{b}) - \frac{\gamma_k \delta}{n} \boldsymbol{x}_k^{\text{sgd}}, \quad (4)$$

where the $i_k \sim \text{Unif}([n])$ iid. While it would also be possible to consider mini-batch SGD, previous work has shown that batch sizes that are vanishingly small as a fraction of the number of samples are equivalent to the single-batch analysis, after appropriately adjusting the time by a factor of the batch size [60, Theorem 1]; similarly, we do not consider high-dimensional SGD with momentum as it degenerates to SGD [58]. See also [30].

**Diffusion approximations and homogenized SGD.** A common paradigm for understanding SGD is through stochastic Langevin diffusions (SLD), i.e. solutions of equations of the form

$$d\boldsymbol{X}_t = -\gamma(\nabla f(\boldsymbol{X}_t)\, dt + \sqrt{\boldsymbol{\Sigma}_t}\, d\boldsymbol{B}_t), \quad (5)$$

where $\gamma$ is the step size of SGD, $f$ is the loss function, $\boldsymbol{B}_t$ is a $d$-dimensional standard Brownian motion, and the matrix $0 \preceq \boldsymbol{\Sigma}_t \in \mathbb{R}^{d \times d}$ models the noise covariance. In many analyses, no concrete connection between SGD and SLD is developed, and the diffusion is merely used to build intuition. A common example is the isotropic case ($\boldsymbol{\Sigma}_t \propto \boldsymbol{I}_d$), for which the Fokker-Planck equation implies that the dynamics are reversible with respect to a density proportional to $e^{-C_\gamma f(x)}$ with $C_\gamma > 0$ some constant. Consequently, the process can escape local minima, exhibiting a trade-off between the entropy and depth of minima and thereby highlighting a possible mechanism of implicit regularization. In the general anisotropic case, describing the stationary distribution is more difficult; nonetheless, the local geometry near minima of $f$ can be analyzed, see [14, 35].

While this type of implicit entropic regularization might ultimately underlie the generalization benefits of SGD for nonconvex problems, currently we lack a precise connection between a concrete SLD and a practical nonconvex learning problem. As such, the implicit regularization effects of SGD on nonconvex losses remains a largely unsolved problem.

For convex quadratics, however, the implications of Eq. (5) are quite clear: there is no notion of implicit regularization as the noise in SLD negatively impacts generalization performance. Note that because the noise is mean zero, any SLD is centered around *gradient flow* (GF) $\mathcal{X}_t^{\text{gf}}$, which solves

$$d\mathcal{X}_t^{\text{gf}} = -\nabla f(\mathcal{X}_t^{\text{gf}}), \quad (6)$$

leading to the following conclusion for generalization (see also [88]):

**Lemma 1.** *Suppose the objective function is $f(\boldsymbol{x}) = \frac{1}{2}\left(\|\boldsymbol{A}\boldsymbol{x} - \boldsymbol{b}\|^2 + \delta\|\boldsymbol{x}\|^2\right)$. Suppose $(\boldsymbol{X}_t : t \in [0, \infty))$ is an SLD (i.e. $\boldsymbol{X}_t$ solves (5)) with $\|\boldsymbol{\Sigma}_t\|_{op}$ almost surely bounded by some $C < \infty$. Suppose the population risk $\mathcal{R} : \mathbb{R}^d \to \mathbb{R}$ is a convex function and denote $\boldsymbol{x}_* \stackrel{\text{def}}{=} \lim_{t \to \infty} \mathcal{X}_t^{gf}$, then*

$$\underbrace{\mathbb{E}[\mathcal{R}(\boldsymbol{X}_t)]}_{\text{pop. risk of SLD}} \geq \underbrace{\mathcal{R}(\mathcal{X}_{\gamma t}^{gf})}_{\substack{\text{pop. risk of} \\ \text{gradient flow}}} \quad \text{for all } t \geq 0 \text{ and hence,} \quad \underbrace{\liminf_{t \to \infty} \mathbb{E}[\mathcal{R}(\boldsymbol{X}_t)]}_{\text{limiting pop. risk of SLD}} \geq \underbrace{\mathcal{R}(\boldsymbol{x}_*)}_{\substack{\text{limiting pop. risk} \\ \text{gradient flow}}} .$$

*If in addition $\mathcal{R}$ is strictly convex, and $\boldsymbol{\Sigma}_t \to \boldsymbol{\Sigma}_\infty$ with $\boldsymbol{\Sigma}_\infty \succ 0$, then the inequality is strict.*

*Proof.* The mean $\mathbb{E}[\boldsymbol{X}_t]$, by the linearity of the gradient $\nabla f$, is GF. Under the conditions given, the law of $\boldsymbol{X}_t$ converges to a Gaussian variable centered at $\boldsymbol{x}_*$. Hence by Fatou's lemma and Jensen's inequality, the inequality follows. $\qquad\square$

We emphasize that this conclusion applies even under general distribution shifts, so long as the risk remains a convex function. Still, the utility of Lemma 1 may not be immediately clear, as it pertains

to SLD and we have not yet established any concrete connection between SLD and the process of interest, SGD. Nor is it evident what form such a connection should take—the agreement between SGD and an SLD cannot occur at the level of individual states since the randomness from each process is not assumed to be coupled. Instead, the most we can hope for is that statistics of the processes agree. Specifically, we might hope that matching the noise structure of SGD with a careful choice of SLD will cause relevant statistics, like the population risk, to be equal.

It turns out that such a choice of SLD exists for convex quadratic problems in high dimensions, and is given by *homogenized SGD* (HSGD), introduced simultaneously in [52, 58]. Both the empirical and population risks ($\mathcal{L}$, $\mathcal{R}$ resp.) of HSGD agree with the same of SGD in the high-dimensional limit (see Thm. 1). Mathematically, HSGD is the strong solution of the stochastic differential equation:

$$\mathrm{d}\boldsymbol{X}_t \stackrel{\text{def}}{=} -\gamma(t)\nabla\mathcal{L}(\boldsymbol{X}_t)\,\mathrm{d}t + \gamma(t)\sqrt{\tfrac{2}{n}\mathcal{L}(\boldsymbol{X}_t)\nabla^2\mathcal{L}(\boldsymbol{X}_t)}\,\mathrm{d}\boldsymbol{B}_t, \quad \text{for quadratic } \mathcal{L}, \tag{7}$$

where again $\boldsymbol{B}_t$ is a $d$-dimensional standard Brownian motion, $\gamma(t)$ is the learning rate schedule, and the initial condition is $\boldsymbol{X}_0 = \boldsymbol{x}_0$. Roughly, HSGD is a diffusion approximation to SGD that gains explanatory power when the *dimensionality* is large. In particular, it does not require the step size $\gamma$ to be small, in contrast to the usual paradigm of SLD approximations. Note that as with other universality results, the details of the noise distribution are not relevant and only the second-order correlations contribute, which are carefully matched by HSGD to SGD.

The precise sense of the comparison requires us to evaluate low-dimensional statistics of the high-dimensional dynamics; "low-dimensional" must be effective, in that the univariate statistics of the SGD iterates concentrate around the same statistic evaluated on HSGD. For understanding generalization or implicit regularization properties, a important statistic is the population risk, $\mathcal{R}$.

**Assumptions.** For all parts of our analysis to hold, the pair $(\boldsymbol{A}, \boldsymbol{b})$ of the data matrix $\boldsymbol{A} \in \mathbb{R}^{n \times d}$ and target vector $\boldsymbol{b} \in \mathbb{R}^n$ must satisfy some *quasi-random* assumptions—a set of deterministic conditions on the pair $(\boldsymbol{A}, \boldsymbol{b})$ that are satisfied with high probability by natural classes of random matrix-vector pairs (see Appendix B for specifics). We use the convention that the target and initialization vectors are bounded independent of $n$, $\|\boldsymbol{b}\|_2^2 \leq C$ and $\|\boldsymbol{x}_0\|_2^2 \leq C$, respectively.

We illustrate some examples below that we have shown to satisfy the quasi-random assumptions.

- Gaussian linear regression. Here the rows of $\boldsymbol{A}$ are iid and drawn from a Gaussian with norm-bounded covariance $\Sigma$ and the target $\boldsymbol{b}$ is drawn from a generative model, $\boldsymbol{b} = \boldsymbol{A}\widetilde{\boldsymbol{x}} + \boldsymbol{\eta}$ for some unknown signal $\widetilde{\boldsymbol{x}} \in \mathbb{R}^d$ and independent noise $\boldsymbol{\eta} \in \mathbb{R}^n$.

- Subgaussian linear designs. In the example above, we can relax the Gaussian assumption to be of the form $\boldsymbol{x}\Sigma^{1/2}$ for $\boldsymbol{x}$ a vector of iid centered subgaussian random variables [88, 30].

- Gaussian random features with a linear ground truth [51, 3, 66, 4]. Suppose $\boldsymbol{A}$ is given by $\sigma(\boldsymbol{X}\boldsymbol{W})$ for an iid standard Gaussian weight matrix $\boldsymbol{W}$ and Gaussian data matrix $\boldsymbol{X}$. Suitable assumptions on the activation function $\sigma$ and the covariance $\Sigma$ of $\boldsymbol{X}$ added.

**Assumption 1.** *The population risk $\mathcal{R} : \mathbb{R}^d \to \mathbb{R}$ is a quadratic, that is, it is a degree-2 polynomial or, equivalently, can be represented by*

$$\mathcal{R}(\boldsymbol{x}) = \frac{1}{2}\boldsymbol{x}^T\boldsymbol{T}\boldsymbol{x} + \boldsymbol{u}^T\boldsymbol{x} + c$$

*for some $d \times d$ symmetric matrix $\boldsymbol{T}$, vector $\boldsymbol{u} \in \mathbb{R}^d$, and scalar $c \in \mathbb{R}$. We further assume that $\|\nabla^2\mathcal{R}\|_{op} \leq C$, $\|\nabla\mathcal{R}(0)\|_2^2 \leq C$, and $|\mathcal{R}(0)| \leq C$.*

A natural population risk is given by $\mathcal{R}(\boldsymbol{x}) = \frac{1}{2}\mathbb{E}\left[(\boldsymbol{a} \cdot \boldsymbol{x} - b)^2\right]$ where $(\boldsymbol{a}, b) \sim \mathcal{D}$. This distribution $\mathcal{D}$ may or may not be the same as the distribution that generated the data $[\boldsymbol{A} \mid \boldsymbol{b}]$ used in training.

As we work in the high-dimensional limit, we suppose that $\gamma_k = \gamma(k/n)$ for a smooth, bounded function $\gamma(\cdot)$ such that $\gamma(t) \to \gamma \in [0, \infty)$ and $\widehat{\gamma} \stackrel{\text{def}}{=} \sup_{t \geq 0} \gamma(t) < \infty$.

## 3 Main results

Our main results are analyzable (non-asymptotic) expressions for the empirical risk $\mathcal{L}$ and the population risk $\mathcal{R}$ of SGD at any time $t$ for the high-dimensional least squares problem (1). To begin, we first establish the following equivalence between SGD and HGSD.

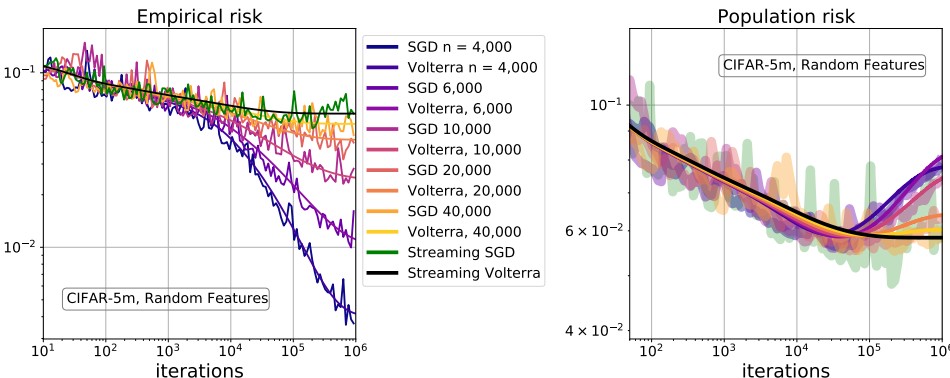

Figure 1: **Single runs of SGD vs. HSGD (Volterra) in streaming** on standarized CIFAR-5M [54] with car/plane class vector (1,000,000 samples); a standarized ReLu (74) random features model (see Appendix B.1) was applied with increasing number of samples $n$ and fixed $d = 6000$. The predicted behavior from HSGD (denoted by Volterra) matches the performance of single runs of SGD for finite $n$ and streaming ($n = \infty$). Shaded region (right) is the moving average of a single run of SGD. Empirical risk (left) increases monotonically with $n$ to its limit while population risk generally decreases with $n$. Streaming corresponds to $n = \infty$ (see Appendix C). For consistency across sample sizes, time is measured in iterations. Additional details in App G.

**Theorem 1** (Equivalence of SGD and HSGD). *Suppose the pair $(\boldsymbol{A}, \boldsymbol{b}) \in \mathbb{R}^{n \times d} \times \mathbb{R}^d$ satisfy the quasi-random assumptions with $d^\varepsilon \leq n \leq d^{1/\varepsilon}$ for some $\varepsilon \in (0, 1]$. Let the iterates $\boldsymbol{x}_t = \boldsymbol{x}^{sgd}_{\lfloor t \rfloor}$ be generated from multi-pass SGD Eq. (4) and $\boldsymbol{X}_t$ be the solution of Eq. (7). Then for any deterministic $T > 0$ and any $D > 0$, there is a $C > 0$ such that*

$$\Pr\left[\sup_{0 \leq t \leq T} \left\| \begin{pmatrix} \mathcal{L}(\boldsymbol{x}_{\lfloor tn \rfloor}) \\ \mathcal{R}(\boldsymbol{x}_{\lfloor tn \rfloor}) \end{pmatrix} - \begin{pmatrix} \mathcal{L}(\boldsymbol{X}_t) \\ \mathcal{R}(\boldsymbol{X}_t) \end{pmatrix} \right\|_2 > d^{-\varepsilon/2} \right] \leq C d^{-D}.$$

The rigorous proof is given in [61]. For the rest of this paper, we will use homogenized SGD to analyze the behavior of multi-pass SGD. See also Appendix C where we heuristically extend this to the case of streaming SGD.

While the comparison of SGD to HSGD requires relatively strong assumptions on $\boldsymbol{A}$ and $\boldsymbol{b}$, the analysis of HSGD can be performed under weaker assumptions (no quasirandomness assumptions are needed). It suffices to suppose the problem is high dimensional in the following sense:

**Assumption 2.** *The empirical risk $\mathcal{L}$ satisfies $\operatorname{tr} \nabla^2 \mathcal{L} = n$ and $0 \preceq \nabla^2 \mathcal{L} \preceq n d^{-\epsilon}$ for some $\epsilon > 0$.*

This corresponds to the normalization where $\nabla^2 \mathcal{L} = \boldsymbol{A}^T \boldsymbol{A}$ and each row of $\boldsymbol{A}$ is length 1 and hence $\operatorname{tr} \nabla^2 \mathcal{L} = n$.

Under Assumptions 1 and 2, the dynamics of the empirical and population risk under HSGD concentrate around a deterministic dynamical system driven by a Volterra integral equation:

> **Volterra Dynamics, Multi-pass.** The following deterministic dynamical system is the high-dimensional limit for $\mathcal{L}(\boldsymbol{X}_t)$ and $\mathcal{R}(\boldsymbol{X}_t)$, respectively
>
> $$\Psi_t = \mathcal{L}\big(\mathfrak{X}^{\mathrm{gf}}_{\Gamma(t)}\big) + \int_0^t K(t,s; \nabla^2 \mathcal{L}) \Psi_s \, \mathrm{d}s \qquad \text{(Empirical risk)} \qquad (8)$$
>
> $$\Omega_t = \mathcal{R}\big(\mathfrak{X}^{\mathrm{gf}}_{\Gamma(t)}\big) + \int_0^t K(t,s; \nabla^2 \mathcal{R}) \Psi_s \, \mathrm{d}s \qquad \text{(Population risk)} \qquad (9)$$
>
> where the *integrated learning rate* $\Gamma$ and *kernel* $K$, for any $d \times d$ matrix $\boldsymbol{P}$, respectively are
>
> $$\Gamma(t) = \int_0^t \gamma(s) \, \mathrm{d}s, \ \ K(t,s; \boldsymbol{P}) = \tfrac{\gamma^2(s)}{n} \operatorname{tr}\big((\nabla^2 \mathcal{L}) \boldsymbol{P} \exp\big(-2(\nabla^2 \mathcal{L} + \delta \mathbf{I}_d)(\Gamma(t) - \Gamma(s))\big)\big). \quad (10)$$

**Theorem 2** (Concentration of HSGD around Volterra dynamics). *Under Assumptions 1 and 2, for any $T > 0$ and for any $D > 0$ there exists sufficiently large $C > 0$ such that for all $d > 0$*

$$\Pr\left[\sup_{0 \le t \le T} \left\| \begin{pmatrix} \mathcal{L}(\boldsymbol{X}_t) \\ \mathcal{R}(\boldsymbol{X}_t) \end{pmatrix} - \begin{pmatrix} \Psi_t \\ \Omega_t \end{pmatrix} \right\| > d^{-\epsilon/2} \right] \le C d^{-D},$$

*where $\Psi_t$ and $\Omega_t$ solve (8) and (9).*

We give a formal proof of the concentration result in Appendix D.1 in Theorem 11.

### 3.1 No implicit regularization from SGD

From (9), for convex $\mathcal{R}$ we observe immediately that the population risk $\Omega_t$ is only larger than the population risk of GF. Moreover, we have an explicit formula for the excess risk due to SGD noise,

$$\underbrace{\Omega_t - \mathcal{R}\big(\mathfrak{X}^{\mathrm{gf}}_{\Gamma(t)}\big)}_{\text{excess risk due to SGD}} \overset{\text{def}}{=} \int_0^t K(t,s; \nabla^2 \mathcal{R}) \times \underbrace{\Psi_s}_{\text{limiting loss } \mathcal{L}} \, \mathrm{d}s.$$

Note that the population risk of SGD tracks that of GF. If GF overfits, SGD overfits as well; there is no statistical regularization due to the noise of SGD applied to empirical risk minimization (ERM).

We can further analyze the long-time behavior of SGD with exact limiting values for this excess risk.

**Theorem 3** (Time infinity risk values). *If $\gamma(t) \to 0$ as $t \to \infty$ but $\Gamma(t) \to \infty$ (i.e. the usual Robbins-Monro setting), then the excess population risk of SGD over GF tends to $0$. If on the other hand $\gamma(t) \to \gamma \in (0, 2(\frac{1}{n} \operatorname{tr}\{\frac{(\boldsymbol{A}^T \boldsymbol{A})^2}{\boldsymbol{A}^T \boldsymbol{A} + \delta \boldsymbol{I}_d}\})^{-1}$, then with $\Psi_\infty$ given by the limiting empirical risk,*

$$\Psi_\infty = \mathcal{L}\big(\mathfrak{X}^{\mathrm{gf}}_\infty\big) \times \left(1 - \frac{\gamma}{2n} \operatorname{tr}\left\{\frac{(\nabla^2 \mathcal{L})^2}{\nabla^2 \mathcal{L} + \delta \boldsymbol{I}_d}\right\}\right)^{-1}$$

*the excess risk due to SGD converges to*

$$\Omega_t - \mathcal{R}\big(\mathfrak{X}^{\mathrm{gf}}_{\Gamma(t)}\big) \to \frac{\gamma \Psi_\infty}{2n} \times \operatorname{tr}\left\{\frac{(\nabla^2 \mathcal{R})(\nabla^2 \mathcal{L})}{\nabla^2 \mathcal{L} + \delta \boldsymbol{I}_d}\right\}.$$

There are a few conclusions to draw directly from this. In the interpolation regime, that is where $\Psi_\infty = 0$, there is no excess risk due to SGD and there is no need to send $\gamma$ to $0$. Moreover, if the empirical risk $\Psi_\infty$ is small, the excess risk due to SGD is proportional to $\gamma \Psi_\infty$, and hence it is frequently orders of magnitude smaller than other potential sources of error. Furthermore, the excess risk is affected by how similar the population and empirical risks are, in the large directions. Ridge regularization can substantially reduce excess risk due to SGD in cases where population risk has many small eigenvalues. In summary, either by sending $\gamma \to 0$, working in the interpolation regime, or otherwise in a regime $\Psi_\infty$ is small, *the excess risk incurred by running SGD is minimal.*

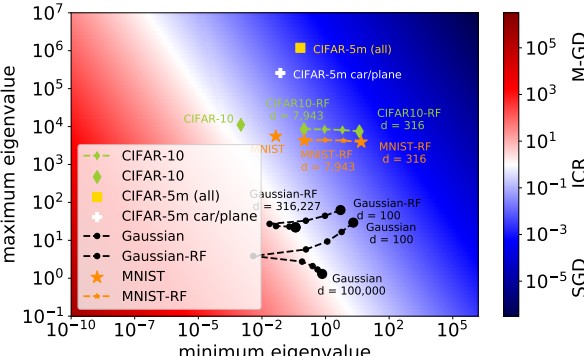

Figure 2: ICR as function of largest and smallest eigenvalues of $A$ with trace normalized to be 1; blue (smaller, SGD favored) and red (larger, full batched MGD favored). Points indicate ICR for image datasets (MNIST, CIFAR-10, CIFAR-5m) as well as their images under ReLu random feature maps of various dimensions (lines). Gaussian random features ($n = 2000$, $n_0 = 100$, various $d$) and Gaussian data ($n = 2000$, various $d$) shows ICR for over- and under-parameterized models. Natural datasets tend to favor SGD.

## 3.2 Implicit conditioning of SGD

In contrast, the algorithmic advantages of SGD are substantial. To simplify the discussion, we consider only the case of constant learning rate $\gamma$. In this case, the kernel in (8) and (9) simplifies to a convolution kernel, which has a much simpler theory. To characterize the rates, we define $\lambda_{\min}$ as the smallest non-zero eigenvalue of $\nabla^2 \mathcal{L}$. Then for generic initial conditions, (in particular almost surely if $X_0$ is nonzero isotropic), GF has the following convergence rate

$$\lim_{t \to \infty} \left( \mathcal{L}(\mathcal{X}^{\mathrm{gf}}_{\gamma t}) - \mathcal{L}(\mathcal{X}^{\mathrm{gf}}_{\infty}) \right)^{1/t} = \begin{cases} e^{-\gamma(\lambda_{\min}(\nabla^2 \mathcal{L}) + \delta)}, & \text{if } \delta > 0, \\ e^{-2\gamma \lambda_{\min}(\nabla^2 \mathcal{L})}, & \text{otherwise.} \end{cases}$$

Here we use the notation that $\lambda_{\min}(H)$ and $\lambda_{\max}(H)$ are the smallest and largest eigenvalues of the matrix $H$. The rate of convergence of $\Psi_t$ to $\Psi_\infty$ can be no faster than the underlying GF, given by the rate above. On the other hand, for larger $\gamma$ the Volterra term in (8) can frustrate the convergence. The *Malthusian exponent* of the convolution Volterra equation is given by

$$\lambda_* = \inf\left\{ x : 1 = \int_0^\infty e^{xt} K(t; \nabla^2 \mathcal{L}) \, \mathrm{d}t \stackrel{\mathrm{def}}{=} \gamma^2 \int_0^\infty e^{xt} \operatorname{tr}\left( (\nabla^2 \mathcal{L})^2 \exp(-2\gamma(\nabla^2 \mathcal{L} + \delta \mathbf{I}_d) t) \right) \, \mathrm{d}t \right\}. \quad (11)$$

As $\nabla^2 \mathcal{L}$ is finite dimensional, we have that $\lambda_* \le 2\gamma(\lambda_{\min}(\nabla^2 \mathcal{L}) + \delta)$, owing to the divergence of the integral as $x$ approaches this value from below. Note that in principal the Malthusian exponent can be negative, in which case SGD is *divergent*. The Malthusian exponent gives the effective rate of convergence of constant learning rate SGD. Define

$$\Xi(\gamma) \stackrel{\mathrm{def}}{=} \begin{cases} \min\{\gamma(\lambda_{\min}(\nabla^2 \mathcal{L}) + \delta), \lambda_*(\gamma)\} & \text{if } \delta > 0, \\ \lambda_*(\gamma) & \text{if } \delta = 0. \end{cases} \quad (12)$$

**Theorem 4** (SGD convergence rates, average-case). *Then the rates of convergence of both the empirical and population risk are controlled by this parameter*

$$\lim_{t \to \infty} \left( \Psi_t - \Psi_\infty \right)^{1/t} = e^{-\Xi(\gamma)} = \lim_{t \to \infty} \left( \Omega_t - \Omega_\infty \right)^{1/t}.$$

*Furthermore, when $\gamma = n(\operatorname{tr}(A^T A))^{-1}$, we have the rate guarantee $\Xi(\gamma) \ge \frac{\lambda_{\min}(\nabla^2 \mathcal{L}) + \delta}{2}$.*

The major difference between SGD and full batch methods such as momentum gradient descent (MGD; see Appendix F.2 for definitions) is that they have different sensitivities to the Hessian spectrum of the empirical risk $\mathcal{L}(x) = \frac{1}{2}\|Ax - b\|^2$. Define the condition numbers

$$\kappa \stackrel{\mathrm{def}}{=} \frac{\lambda_{\max}(\nabla^2 \mathcal{L}) + \delta}{\lambda_{\min}(\nabla^2 \mathcal{L}) + \delta} \quad \text{and} \quad \overline{\kappa} \stackrel{\mathrm{def}}{=} \frac{\frac{1}{n}\operatorname{tr}(\nabla^2 \mathcal{L})}{\lambda_{\min}(\nabla^2 \mathcal{L}) + \delta}.$$

The first of these is the classical condition number of the ridge problem, while the second is the averaged condition number that regulates the behavior of SGD in the high-dimensional limit. MGD has been long established to have a rate of convergence, with proper tuning, controlled by the square root of the condition number [65], which is known to be optimal amongst first order algorithms.

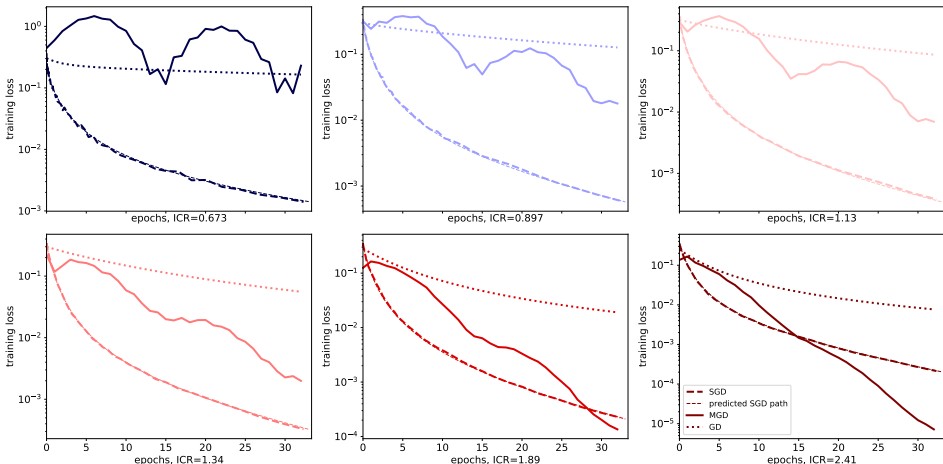

Figure 3: **ICR effect on full batch MGD versus SGD** in a synthetic least-squares setting. We consider minimizing, for an $n \times d$ matrix $\boldsymbol{A}$ (with $n = 2400, d = 3600$) $\min_{x \in \mathbb{R}^d} \frac{1}{2} \| \boldsymbol{Ax} - \boldsymbol{b} \|^2$, where $\boldsymbol{A}$ is a Gaussian matrix, with correlated rows and $\boldsymbol{b}$ is given by $\boldsymbol{A\beta} + \boldsymbol{\eta}$ for a ground truth, which is isotropic normal of expected norm-square 1, and $\boldsymbol{\eta}$ is isotropic normal of expected norm-square 0.02.

**Theorem 5** (Convergence rates for MGD). *For isotropic random initialization $\boldsymbol{x}_0$ or noisy $\boldsymbol{b}$, $\delta > 0$, and strictly convex population risk $\mathcal{R}$*

$$\left(\mathcal{L}(\boldsymbol{x}_k^{m\text{-}gd}) - \mathcal{L}(x_*)\right)^{1/k} \xrightarrow[k \to \infty]{\text{a.s.}} \left(\frac{\sqrt{\kappa} - 1}{\sqrt{\kappa} + 1}\right) \quad and \quad \left(\mathcal{R}(\boldsymbol{x}_k^{m\text{-}gd}) - \mathcal{R}(x_*)\right)^{1/k} \xrightarrow[k \to \infty]{\text{a.s.}} \left(\frac{\sqrt{\kappa} - 1}{\sqrt{\kappa} + 1}\right).$$

See Appendix F.2 for elaboration.

In light of Theorems 4 and 5, we can define the *implicit-conditioning ratio* as

$$\text{ICR} \stackrel{\text{def}}{=} \frac{\overline{\kappa}}{\sqrt{\kappa}} \approx \log\left(\frac{\sqrt{\kappa} - 1}{\sqrt{\kappa} + 1}\right) \overline{\kappa},$$

which measures the efficiency of SGD over MGD in that SGD with constant learning rate $n/\operatorname{tr}(\nabla^2 \mathcal{L})$ trains in an ICR-multiple of the number of epochs that MGD requires (lower is better for SGD).

Problems favor SGD when there are large outlier eigenvalues, a common feature of Hessian spectra in practice [69, 70, 1]. Indeed, if the largest eigenvalues are on the same order as the *unnormalized trace*, individual SGD iterates are as effective as full-batch gradient. In contrast, when the Hessian spectrum is tightly packed, which is less common in practice but can occur after some preprocessing techniques or e.g. for uncorrelated Gaussian samples, then MGD is favored. See Fig. 2.

### 3.3 Numerical results on ICR

In Fig. 3, we illustrate how ICR affects the relative performance of full batch MGD versus single batch SGD in a synthetic least squares setting where we have tight control over the all the particulars of the problem and in a neural network setting. We control the ICR by controlling the covariance singular value spectrum of the rows of $\boldsymbol{A}$, which we take as Pareto distributed with exponent $s$ for varying $s > 2$. This choice allows us to affect the ICR of the problem without changing the minimum curvature of the Hessian. The results, comparing SGD, MGD, and GD, are given in Figure 3.

We note a few key qualitative observations. First, even in MGD favored configurations, SGD will outperform MGD on short time scales. When optimizing the hyperparameters in MGD for long-time performance, minimal curvature (which in this case is just the minimal eigenvalue of $\boldsymbol{AA}^T$) plays a major role in the choices; being tuned for long-time performance, MGD typically performs suboptimally at initialization. In contrast, the learning rate in SGD only depends on average curvature, and so it generally performs better at initialization on problems with a larger interval of Hessian spectra.

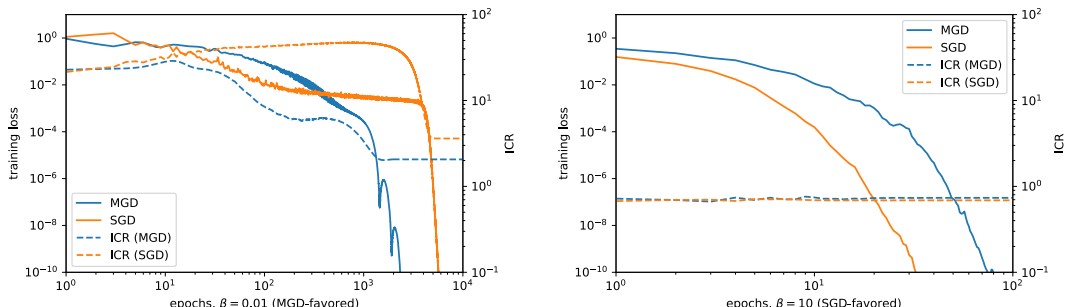

Figure 4: **ICR and full-batch MGD versus SGD** with a fully connected 2-layer neural network with activation function $f(x) = \frac{1}{\sqrt{\beta}} \operatorname{erf}(\sqrt{\beta}x)$ on a 500-sample subset of CIFAR-10 with targets car/plane. Momentum hyper-parameters were tuned empirically to give best performance. We note that the loss curve of MGD also displays the characteristic cycloidal oscillations of tuned momentum on inhomogeneous problems. *Left:* $\beta = 0.01$. ICR varies over the course of training but stays above 1, favoring momentum as illustrated in the figure. *Right:* $\beta = 10$. ICR varies slightly but always stays below 1 throughout the training and SGD is favored over full-batch momentum.

Second, we note that the problem setup was chosen to hold the minimum curvature roughly constant while varying $s$. When $s$ tends to 2 from above, the largest eigenvalues of $\boldsymbol{A}\boldsymbol{A}^T$ grows with feature dimension $d$, but the average and minimum eigenvalue stays bounded with feature dimension. Hence we can send the ICR to 0 by choosing an $s$ above 2 and increasing $d$ (or $n$).[1] On the other hand, by sending $s \to \infty$, we send the covariance matrix to the identity, which tends to be momentum favored.[2]

In Fig. 4, we run SGD on a fully-connected 2-layer neural network on a subset of CIFAR-10 in order to examine the dynamics of the ICR for a non-trivial problem and to see how our insights might play out in practice. Owing to the non-convexity of this problem, we define the ICR in terms of the Gauss-Newton approximation to the Hessian, or equivalently in terms of the Neural Tangent Kernel [29]. By changing the activation function of the network, we can vary the initial ICR from an SGD-favored to a momentum-favored value. While the ICR does change over the course of training, we find that, at least in this setting, the initial ICR can nevertheless predict the relative performance of SGD versus MGD. Indeed, for activation functions for which the ICR remains above 1.0, the training remains MGD-favored over sufficiently long times, and we observe that MGD with optimal parameters does converge faster than SGD. In contrast, when the ICR remains below 1.0, we find that SGD outperforms MGD.

## 4 Conclusion.

Using a specific type of SLD (called HSGD) that matches the second-order correlations in the noise of SGD, we demonstrated that their empirical and population risks match in the high-dimensional limit. Moreover, the risks of HSGD behavior deterministically, as described by a Volterra equation. With this connection, we investigated the benefits of SGD on a convex objective. While there is no statistical benefit to generalization from the noise of SGD, in overparameterized, interpolating settings little is lost compared to GD. Moreover, when computational restrictions are imposed, SGD can be radically faster than GD because of its dependence on a different condition number of the Hessian. We characterized this speed up using the ICR, which when calculated for datasets common in deep learning clearly favors SGD. This should highlight the difficulty in studying implicit regularization for SGD empirically: any experiment necessarily has a finite computational budget and may find

---

[1]The relevance of $s > 2$ is that the Pareto has moments up to and including the second moment. For values less than 2, the covariance spectra is sufficiently heavy that the maximum eigenvalue of $\boldsymbol{A}\boldsymbol{A}^T$ dominates the trace. In that regime, the problem becomes effectively sparse, with an intrinsic dimension depending only on $s$, and it should be expected that GD/SGD are approximately equivalent and outperform MGD.

[2]Furthermore, the 'average curvature' speedup of SGD on short time scales becomes muted, owing to all curvatures being the same.

lower population risks with SGD simply via its improved conditioning. Finally, we demonstrated limitations in using streaming SGD alone as a tool for studying generalization.

As future work, a major outstanding problem (both theoretically and empirically) is extending the analysis above to non-quadratic losses, both train and test, and especially to other high-dimensional problems not in the kernel regime. Finally, data augmentation can naturally be considered by randomly augmenting each sample from $\widehat{\mathcal{D}}_n$ in Eq. (30).

## Acknowledgments and Disclosure of Funding

C. Paquette's research was supported by CIFAR AI Chair, MILA, a Discovery Grant from the Natural Science and Engineering Council (NSERC), and the FRQNT New University Researcher's Start-up Program. Research by E. Paquette was supported by a Discovery Grant from the Natural Science and Engineering Council (NSERC). Additional revenues related to this work: C. Paquette has part-time employment at Google Research, Brain Team, Montreal, QC.

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
