# Implicit Regularization or Implicit Conditioning?
# Exact Risk Trajectories of SGD in High Dimensions

## Supplementary material

The appendix is organized into five sections as follows:

1. Appendix A builds on the background and preliminaries discussed in the main text. We include additional related work, an extended discussion on diffusion processes and Volterra equations.

2. Appendix B expands upon the assumptions/setting around (1) and it discusses some motivating applications such as in- and out- of distribution expected risk and random features. Moreover, we discuss the equivalence of homogenized SGD and SGD, see Theorem 1.

3. Appendix C gives results for the streaming SGD setting. In particular, we establish a Volterra equation.

4. Appendix D introduces a general Volterra class of equations, called the Volterra SLD class, that encompasses homogenized SGD and its Volterra dynamics in the multi-pass setting (Section 3) and streaming (Section C). This general Volterra class allows for more types of additive noise. We prove in this section that the Volterra SLD class concentrates around its mean; thereby deriving the proof of Theorem 2.

5. We prove in Appendix E the limiting risk values for the Volterra SLD class (Theorem 12 (constant learning rate) and Theorem 13 (time dependent learning rate)). These two theorems immediately imply the limiting risk values for homogenized SGD in the multi-pass and streaming settings, see Theorems 3 and 10 respectively.

6. Appendix F discusses the exact asymptotic convergence rates for SGD and full batch momentum algorithms on high-dimensional $\ell^2$-regularized least squares problems. The results in this section (e.g., Theorems 4 and 5) were shown in a series of papers [58–60] that explored exact trajectories of loss function.

7. Appendix G contains details on the simulations.

**Potential societal impacts.**  The results presented in this paper concern the analysis of existing methods on a simple $\ell^2$-regularized least squares problems. The results are theoretical and we do not anticipate any direct ethical and societal issues. We believe the results will be used by machine learning practitioners and we encourage them to use it to build a more just, prosperous world.

## A  Extended preliminaries and background.

We include some additional preliminary background information.

**Related work.**  We highlight recent progress of advances in analyzing the excess risk of SGD. As these areas are highly active, we present below a non-exhaustive list of the current progress, and particularly focus on the least-squares setting.

In the literature, convergence guarantees and risk bounds are available for analyzing SGD and its variants [71, 56, 53, 16, 79]. A popular paradigm for analyzing risk bounds of SGD is the one-pass or streaming setting where one supposes that the gradient estimators are independent with a common distribution [24, 30]. Such a setting was considered in a series of works [30, 18] which explored 'one-pass' SGD on a least-squares under a design condition on the data matrix. Extending this idea, [87] provide upper and lower excess risk bounds for constant stepsize SGD on the $\ell^2$-regularized least-squares problem which extends the work of [8, 78]. These bounds are characterized by the full eigenspectrum of the population covariance matrix. Beyond the confines of the streaming setting, much less is known about the risk bounds for multi-pass SGD [45, 64, 39, 88]. Like in this work, [45, 64, 88] consider the simplified setting of analyzing the behavior of multi-pass SGD on a (high-dimensional) $\ell^2$-regularized least-squares problem. In contrast to our exact dynamics of the excess risk, previous works provide only bounds. To the best of knowledge, the only other paper which

rigorously computed exact dynamics in the high-dimensional setting was [59]. They were the first to show dynamics on the *training loss* of a least-squares problem, assuming a left orthogonal invariance condition. In this paper, we remove the left orthogonal invariance condition and extend dynamics to generalization performance. Other approaches to analyzing generalization performance of multi-pass SGD include uniform stability (see c.f. [25]). Excess risk and uniform stability are related through a (loose) triangle inequality; we preferred to focus on the excess risk in this paper and leave discussions of uniform stability for future directions.

The exact features of SGD that are responsible for the success of SGD on high-dimensional problems are the subject of extensive research. These features are often labelled under the umbrella term the *implicit regularization effects of SGD*. The implicit regularization mechanism has been primarily hypothesized to exist in highly nonconvex settings. In such a setting, empirical observations have led to the conclusion that the noise generated by small batch [32, 27] and/or large learning rate [40] SGD leads to better generalization performance. A proposed mechanism for this improvement is that the stochasticity inherent in SGD allows the optimizer to escape traps (c.f. [13] or [85]) which poorly generalize and are stationarity points for gradient flow. A related point of view on the proposed escape mechanism is that SGD demonstrates a preference for flat minima, long considered to be a preferable solution for generalization properties [26, 80]. Using a continuous time model, in a certain noise regime, [80] shows that SGD prefers flat minima which is different from the flat minimia flat minimum selection of continuous time SGD with homogeneous noise. For the least-squares problem, multi-pass SGD converges to the minimal norm solution [23, 57, 84] which is widely cited as the implicit bias of SGD [33, 17]. Moreover other works have shown for the least-squares problem with $\ell^2$-regularization results similar to Lemma 1, saying that the generalization performance of SGD always performs worse than gradient descent [88].

**Diffusion approximations and SGD.**    In this work, we analyze a new mathematical tool, homogenized SGD, introduced simultaneously in [58, 52] (a similar related SDE was also analyzed in [63]). This SDE is the high-dimensional equivalence of SGD. The technique of using SDEs to analyze SGD is not new (see, for example, stochastic modified equation (SME) [42, 48] and Langevin dynamics [15] and other SDE formulations and interpretations [31, 36, 46, 7]). Diffusion approximations to SGD have a long history. In the stochastic approximation literature, it appears as a natural counterpart to ODE methods (c.f. [36], [47]). However, these are methods that require the vanishing learning rate (such as $\gamma_k = 1/k$) and moreover, in the setup we have suggested here, the resulting SDE (which only arises in an asymptotic comparison, as is standard with stochastic approximation theory) has a vanishing diffusion term for such an aggressive learning-rate decay — the asymptotic trajectory of SGD is also approximated by the ODE gradient flow with time change $\Gamma(n) \approx \log n$.

A more natural point of comparison is the *stochastic modified equation* of [48, 42, 41] which has been rigorously compared to the behavior of SGD [43]. To make a comparison with [43, 28], we fix the learning rate $\gamma$ and we rescale time to be on the order of epochs. With these changes, the SME solves,

$$\mathrm{d}\boldsymbol{\mathcal{M}}_t \stackrel{\text{def}}{=} -\gamma \nabla f(\boldsymbol{\mathcal{M}}_t)\,\mathrm{d}t + \gamma\sqrt{n\boldsymbol{\Sigma}(\boldsymbol{\mathcal{M}_t})}\,\mathrm{d}\boldsymbol{B}_t, \quad \text{where} \quad \begin{cases} \boldsymbol{\Sigma}(\boldsymbol{x}) \stackrel{\text{def}}{=} \mathbb{E}\big(\nabla g_I(\boldsymbol{x}) \otimes \nabla g_I(\boldsymbol{x})\big), \\ g_I(\boldsymbol{x}) \stackrel{\text{def}}{=} f_I(\boldsymbol{x}) - \mathbb{E}\,f_I(\boldsymbol{x}), \\ I \stackrel{\text{def}}{=} \text{Uniform}\{1, 2, \ldots, n\}. \end{cases} \tag{13}$$

The diffusion matrix $\boldsymbol{\Sigma}$ of the SME is chosen to exactly match the covariance of the increments of SGD (4). When applied to the $\ell^2$-regularized problem (1), this matrix becomes (with $\boldsymbol{a}_i$ the $i$-th row of $\boldsymbol{A}$)

$$\boldsymbol{\Sigma}(\boldsymbol{x}) = \frac{1}{n}\sum_{i=1}^{n}(\boldsymbol{a}_i \cdot \boldsymbol{x} - b_i)^2 \boldsymbol{a}_i^T \boldsymbol{a}_i - \frac{1}{n^2}\boldsymbol{A}^T(\boldsymbol{A}\boldsymbol{x} - \boldsymbol{b})(\boldsymbol{A}\boldsymbol{x} - \boldsymbol{b})^T\boldsymbol{A}.$$

HSGD and the SME can be compared by replacing second term by 0 and the first term by

$$n\boldsymbol{\Sigma}(\boldsymbol{x}) \approx \frac{1}{n}\bigg(\sum_{i=1}^{n}(\boldsymbol{a}_i \cdot \boldsymbol{x} - b_i)^2\bigg) \times \bigg(\sum_{i=1}^{n}\boldsymbol{a}_i^T\boldsymbol{a}_i\bigg) = \frac{2}{n}\mathcal{L}(\boldsymbol{x})\nabla^2\mathcal{L}(\boldsymbol{x}),$$

which is the diffusion coefficient in HSGD.

The SME has been used for a variety of purposes, such as optimal learning rate scheduling [41], analysis of momentum terms [42], and prediction of test risk behavior [75]; however, analysis of

the SME is itself difficult, as the diffusion coefficient involves interactions between the functions $f_i$, and, to our knowledge, while the theory developed in [43] provides dimension-independent comparisons, the resulting SME has not been analyzed in any high-dimensional setting. Furthermore, the mathematical comparison which is proven in [43], on the time scale in (13), gives a comparison for time of order $\mathcal{O}(1/n)$ and with an error which is bounded by $\mathcal{O}(\gamma)$. As such, in a high-dimensional setting, the comparison that exists between SME and SGD only provides a non-vanishing error over a vanishing window of time.

Because the SME naturally matches the drift and diffusion matrix of SGD, the above remarks might lead to a conjecture that in fact no comparison is possible between SGD and SDEs; indeed, in a fixed-dimensional analysis, Yaida [83] showed that there is no small learning-rate limit of SGD that produces nontrivial stochastic behavior. In contrast, we will show that in a high-dimensional limit, this is precisely what occurs (although when univariate statistics of this high-dimensional SDE are taken, almost deterministic behavior is seen).

**Background on Volterra equations.**    Volterra dynamics appear frequently in filtering, population dynamics, and the renewal problem, to name a few, and consequently their properties, particularly convergence and limiting behavior, are well-studied in the literature [22, 67]. Convolution-type volterra equation, that is, equations of the form

$$\psi(t) = F(t) + \int_0^t K(t - s)\psi(s)\ \mathrm{d}s, \quad t \geq s, \tag{14}$$

where the forcing term $F : \mathbb{R}_{\geq 0} \to \widehat{\boldsymbol{R}}$ and the kernel is $K : \mathbb{R}_{\geq 0} \to \mathbb{R}$ with $\mathbb{R}_{\geq 0} = \{t \geq 0\}$, in particular, can be "solved" using Laplace transforms (must be able to invert the Laplace transform). For general Volterra equations, one replaces (14) is replaced with a general function on two variables, $K : \mathbb{R}_{\geq 0} \times \mathbb{R}_{\geq 0} \to \mathbb{R}$.

The solution of $\psi$ can be found by repeatedly convolving the forcing term $F(t)$ with the kernel $K$ (provided $\sup_{t \geq 0} \sup_{s \geq 0} \mathcal{K}(t, s; \nabla^2 \mathcal{L})$ is bounded [22]), that is,

$$\psi(t) = F(t) + (K \star F)(t) + (K \star K \star F)(t) + \cdots \ \text{ where } \begin{cases} (K \star h)(t) \stackrel{\text{def}}{=} \int_0^t K(t, s)h(s)\ \mathrm{d}s, \\ \qquad \forall\, h\, \in C([0, \infty)). \end{cases} \tag{15}$$

Moreover, numerical approximations to (15) can be found by taking a large but finite number of convolutions in the expression above. The boundedness of this solution corresponds precisely to learning rate choices for which SGD is convergent.

## B    Quasi-random assumptions on the data matrix, targets, and initialization

The data matrix $\boldsymbol{A} \in \mathbb{R}^{n \times d}$, target $\boldsymbol{b} \in \mathbb{R}^n$, and initialization $\boldsymbol{x}_0 \in \mathbb{R}^d$ may be deterministic or random; we formulate our theorems for deterministic matrix $\boldsymbol{A}$ and vectors $\boldsymbol{b}$ and $\boldsymbol{x}_0$ satisfying various assumptions, and in the applications of these theorems to statistical settings, we shall show that random $\boldsymbol{A}$ and $\boldsymbol{b}$ satisfy those assumptions. These assumptions are motivated by ERM and, in particular, when the augmented matrix $[\boldsymbol{A} \,|\, \boldsymbol{b}]$ has rows that are independent and sampled from some common distribution. We call these assumptions *quasi-random*.

As the problem (1) is homogeneous, we adopt the following normalization convention without loss of generality.

**Assumption 3** (Data-target normalization)**.** *There is a constant $C > 0$ independent of $d$ and $n$ such that the spectral norm of $\boldsymbol{A}$ is bounded by $C$ and the target vector $\boldsymbol{b} \in \mathbb{R}^n$ is normalized so that $\|\boldsymbol{b}\|^2 \leq C$.*

More importantly, we also assume that the data and targets resemble typical unstructured high-dimensional random matrices. One of the principal qualitative properties of high-dimensional random matrices is the *delocalization of their eigenvectors*, which refers to the statistical similarity of the eigenvectors to uniform random elements from the Euclidean sphere. The precise mathematical

description of this assumption is most easily given in terms of resolvent bounds. The resolvent $R(z; \boldsymbol{M})$ of a matrix $\boldsymbol{M} \in \mathbb{R}^{d \times d}$ is

$$R(z; \boldsymbol{M}) = (z\mathbf{I}_d - \boldsymbol{M})^{-1} \quad \text{for } z \in \mathbb{C}.$$

In terms of the resolvent, we suppose the following:

**Assumption 4.** *Suppose $\Omega$ is the contour enclosing $[0, 1 + \|\boldsymbol{A}\|^2]$ at distance $1/2$. Suppose there is a $\theta \in (0, \frac{1}{2})$ for which*

1. $\max_{z \in \Omega} \max_{1 \leq i \leq n} |e_i^T R(z; \boldsymbol{A}\boldsymbol{A}^T)\boldsymbol{b}| \leq n^{\theta - 1/2}.$

2. $\max_{z \in \Omega} \max_{1 \leq i \neq j \leq n} |e_i^T R(z; \boldsymbol{A}\boldsymbol{A}^T)e_j^T| \leq n^{\theta - 1/2}.$

3. $\max_{z \in \Omega} \max_{1 \leq i \leq n} |e_i^T R(z; \boldsymbol{A}\boldsymbol{A}^T)e_i - \frac{1}{n} \operatorname{tr} R(z; \boldsymbol{A}\boldsymbol{A}^T)| \leq n^{\theta - 1/2}.$

Only the resolvent of $\boldsymbol{A}\boldsymbol{A}^T$ appears in these assumptions, and so in effect we are only assuming statistical properties on the left singular-vectors of $\boldsymbol{A}$. This assumption reflects the common formulation of ERM in which the rows of $\boldsymbol{A}$ are independent, and so the left singular-vectors of $\boldsymbol{A}$ are expected to be delocalized (under some mildness assumptions on the distributions of the rows). The first condition, which involves the interaction between $\boldsymbol{A}\boldsymbol{A}^T$ and $\boldsymbol{b}$, can be understood as requiring that $\boldsymbol{b}$ is not too strongly aligned with the left singular-vectors of $\boldsymbol{A}$. The other two conditions can be viewed as corollaries of delocalization of the left singular-vectors.

As for the initialization $\boldsymbol{x}_0$, we need to suppose that it, like $\boldsymbol{b}$, does not interact too strongly with the left singular-vectors of $\boldsymbol{A}^T \boldsymbol{A}$. In the spirit of Assumption 4, it suffices to assume the following:

**Assumption 5.** *Let $\Omega$ be the same contour as in Assumption 4 and let $\theta \in (0, \frac{1}{2})$. Then*

$$\max_{z \in \Omega} \max_{1 \leq i \leq d} |e_i^T R(z; \boldsymbol{A}^T \boldsymbol{A})\boldsymbol{x}_0| \leq n^{\theta - 1/2}.$$

Note that, as a simple but common case, this assumption is surely satisfied for $\boldsymbol{x}_0 = \boldsymbol{0}$. In principle, this assumption is general enough to allow for $\boldsymbol{x}_0$ that are correlated with $\boldsymbol{A}$ in a nontrivial way, but we do not have an application for such an initialization. For a large class of nonzero initializations independent from $(\boldsymbol{A}, \boldsymbol{b})$, this assumption is satisfied, as a corollary of Assumption 4:

**Lemma 2.** *Suppose that Assumption 4 holds with some $\theta_0 \in (0, \frac{1}{2})$ and that $\boldsymbol{x}_0$ is chosen randomly, independent of $(\boldsymbol{A}, \boldsymbol{b})$, and with independent coordinates in such a way that for some $C$ independent of $d$ or $n$*

$$\| \mathbb{E}\, \boldsymbol{x}_0 \|_\infty \leq C/n \quad \text{and} \quad \max_i \|(\boldsymbol{x}_0 - \mathbb{E}\, \boldsymbol{x}_0)_i\|_{\psi_2}^2 \leq Cn^{2\theta_0 - 1}.$$

*For any $\theta > \theta_0$, Assumption 5 holds with any $\theta > \theta_0$ on an event of probability tending to 1 as $n \to \infty$.*

Note that this assumption allows for deterministic $\boldsymbol{x}_0$ having maximum norm $\mathcal{O}(1/n)$, as well as iid centered subgaussian vectors of Euclidean norm $\mathcal{O}(1)$.

To execute the mathematical comparison between SGD and HSGD, we require an additional assumption on the quadratic in the same spirit as Assumption 3.

**Assumption 6** (Quadratic statistics). *Suppose $\mathcal{R} : \mathbb{R}^d \to \mathbb{R}$ is quadratic, i.e. there is a symmetric matrix $\boldsymbol{T} \in \mathbb{R}^{d \times d}$, a vector $\boldsymbol{u} \in \mathbb{R}^d$, and a constant $c \in \mathbb{R}$ so that*

$$\mathcal{R}(\boldsymbol{x}_t) = \tfrac{1}{2}\boldsymbol{x}_t^T \boldsymbol{T} \boldsymbol{x}_t + \boldsymbol{u}^T \boldsymbol{x}_t + c. \tag{16}$$

*We also assume that $\mathcal{R}$ satisfies Assumption 1. Moreover, we assume the following (for the same $\Omega$ and $\theta$) as in Assumption 4:*

$$\max_{z,y \in \Omega} \max_{1 \leq i \leq n} |e_i^T \boldsymbol{A}\widehat{\boldsymbol{T}}\boldsymbol{A}^T e_i - \tfrac{1}{n} \operatorname{tr}(\boldsymbol{A}\widehat{\boldsymbol{T}}\boldsymbol{A}^T)| \leq \|\boldsymbol{T}\|_{op} n^{-\epsilon} \text{ where } \begin{cases} \widehat{\boldsymbol{T}} = R(z)\boldsymbol{T}R(y) + R(y)\boldsymbol{T}R(z), \\ R(z) = R(z; \boldsymbol{A}^T \boldsymbol{A}) \end{cases}$$

$$\tag{17}$$

This assumption ensures that quadratic $\mathcal{R}$ has a Hessian that is not too correlated with any of the left singular vectors of $\boldsymbol{A}$. Establishing Assumption 6 can be nontrivial in the cases when the quadratic has complicated dependence on $\boldsymbol{A}$. In simple cases, (especially for the case of the empirical risk and the norm) it follows automatically from Assumption 4.

**Lemma 3.** *Suppose that $\mathcal{R}$ satisfies (16) with $\boldsymbol{T}$ given by a polynomial $p$ in $\boldsymbol{A}^T\boldsymbol{A}$ (especially $\boldsymbol{I}$ and the monomial $\boldsymbol{A}^T\boldsymbol{A}$) having bounded coefficients, and suppose $\boldsymbol{u}$ and $c$ are norm bounded independently of $n$ or $d$. Then supposing Assumptions 3 and 4 for some $\theta_0 \in (0, \frac{1}{2})$, for all $n$ sufficiently large and for any $\theta > \theta_0$, Assumption 6 holds.*

For proofs of Lemma 2 and 3, see Section 2 in [61].

### B.1  Motivating applications

**Training loss and sample covariance matrices.**  One important (nonstatistical) quadratic statistic, which allows analysis of the optimization aspects of SGD in high dimensions, is the $\ell^2$-regularized loss function $f$ in (1). Then provided that $\boldsymbol{A}, \boldsymbol{b}$ satisfy Assumptions 3 and 4, $\boldsymbol{x}_0$ is iid subgaussian, Lemmas 2 and 3 and Theorems 1 and 2 show that $f(\boldsymbol{x}_k)$ concentrates around the solution of a Volterra integral equation. A natural setup under which Assumptions 3 and 4 are satisfied is the following:

**Assumption 7.** *Suppose $M > 0$ is a constant. Suppose that $\boldsymbol{\Sigma}$ is a positive semi-definite $d \times d$ matrix with $\operatorname{tr}\boldsymbol{\Sigma} = 1$ and $\|\boldsymbol{\Sigma}\|_{op} \leq M/\sqrt{d} < \infty$. Suppose that $\boldsymbol{A}$ is a random matrix $\boldsymbol{A} = \boldsymbol{Z}\sqrt{\boldsymbol{\Sigma}}$ where $\boldsymbol{Z}$ is an $n \times d$ matrix of independent, mean 0, variance 1 entries with subgaussian norm at most $M < \infty$, and suppose $n \leq Md$. Finally suppose that $\boldsymbol{b} = \boldsymbol{A}\boldsymbol{\beta} + \boldsymbol{\xi}$ for $\boldsymbol{\beta}, \boldsymbol{\xi}$ iid centered subgaussian satisfying $\|\boldsymbol{\beta}\|^2 = R$ and $\|\boldsymbol{\xi}\|^2 = \frac{n}{d}\widetilde{R}$.*

These assumptions naturally lead to random matrices that satisfy Assumption 7 with good probability:

**Lemma 4.** *If $(\boldsymbol{A}, \boldsymbol{b})$ satisfy Assumption 7, then $(\boldsymbol{A}, \boldsymbol{b})$ satisfies Assumptions 3 and 4 with probability tending to $1 - e^{-\Omega(d)}$.*

Hence, under these assumptions, we conclude:

**Theorem 6.** *Suppose $(\boldsymbol{A}, \boldsymbol{b})$ satisfy Assumption 7, $\delta > 0$ and $\boldsymbol{x}_0$ is iid centered subgaussian with $\mathbb{E}\|\boldsymbol{x}_0\|^2 = \widehat{R}$. Then for some $\epsilon > 0$, for all $T > 0$, and for all $D > 0$ there is a $C > 0$ such that*

$$\Pr\left(\sup_{0 \leq t \leq T}\left\|\begin{pmatrix}\mathcal{L}(\boldsymbol{x}^{sgd}_{\lfloor tn \rfloor}) \\ \frac{1}{2}\|\boldsymbol{x}^{sgd}_{\lfloor tn \rfloor} - \boldsymbol{\beta}\|_2^2\end{pmatrix} - \begin{pmatrix}\Psi_t \\ \Omega_t\end{pmatrix}\right\|_2 > d^{-\epsilon}\right) \leq Cd^{-D},$$

*where $\Psi_t$ solves (10) and $\Omega_t$ solves (9) with $\mathcal{R} = \frac{1}{2}\| \cdot -\boldsymbol{\beta}\|_2^2$.*

We discuss generalization implications in the the next section.

Theorem 6 generalizes [59] in that it allows for varying training rates, adds a regularization parameter, and allows for non-orthogonally-invariant designs $\boldsymbol{A}$. We further note that under the assumptions of Theorem 6, we can further approximate the behavior of GF to show that

$$\Psi_t = \mathcal{L}(\boldsymbol{\mathcal{X}}^{\text{gf}}_{\Gamma(t)}) + \frac{1}{n}\int_0^t \gamma^2(s)\operatorname{tr}\left((\boldsymbol{A}^T\boldsymbol{A})^2 e^{-2(\boldsymbol{A}^T\boldsymbol{A}+\delta\mathbf{I}_d)(\Gamma(t)-\Gamma(s))}\right)\Psi_s \, \mathrm{d}s$$

where 
$$\mathcal{L}(\boldsymbol{\mathcal{X}}^{\text{gf}}_{\Gamma(t)}) = \frac{R}{2d}\operatorname{tr}\left[(\boldsymbol{A}^T\boldsymbol{A})\left(\boldsymbol{A}^T\boldsymbol{A}(\boldsymbol{A}^T\boldsymbol{A}+\delta\mathbf{I}_d)^{-1}\left(\mathbf{I}_d - e^{-(\boldsymbol{A}^T\boldsymbol{A}+\delta\mathbf{I}_d)\Gamma(t)}\right) - \mathbf{I}_d\right)^2\right]$$
$$+ \frac{\widetilde{R}}{2d}\operatorname{tr}\left[\left(\boldsymbol{A}(\boldsymbol{A}^T\boldsymbol{A}+\delta\mathbf{I}_d)^{-1}\left[\mathbf{I}_d - e^{-(\boldsymbol{A}^T\boldsymbol{A}+\delta\mathbf{I}_d)\Gamma(t)}\right]\boldsymbol{A}^T - \mathbf{I}_n\right)^2\right]$$
$$+ \frac{\widehat{R}}{2d}\operatorname{tr}\left(\boldsymbol{A}^T\boldsymbol{A}e^{-2(\boldsymbol{A}^T\boldsymbol{A}+\delta\mathbf{I}_d)\Gamma(t)}\right).$$

(18)

For the risk $\mathcal{R}(\cdot) = 1/2\| \cdot -\boldsymbol{\beta}\|_2^2$, we have following expression

$$\Omega_t = \mathcal{R}(\mathfrak{X}_{\Gamma(t)}^{\mathrm{gf}}) + \frac{1}{n} \int_0^t \gamma^2(s) \operatorname{tr}\left( (\boldsymbol{A}^T\boldsymbol{A}) e^{-2(\boldsymbol{A}^T\boldsymbol{A} + \delta\mathbf{I}_d)(\Gamma(t) - \Gamma(s))} \right) \Psi_s \, \mathrm{d}s$$

where $\quad \mathcal{R}(\mathfrak{X}_{\Gamma(t)}^{\mathrm{gf}}) = \dfrac{R}{2d} \operatorname{tr}\left[ \left( \boldsymbol{A}^T\boldsymbol{A}(\boldsymbol{A}^T\boldsymbol{A} + \delta\mathbf{I}_d)^{-1} \left( \mathbf{I}_d - e^{-(\boldsymbol{A}^T\boldsymbol{A} + \delta\mathbf{I}_d)\Gamma(t)} \right) - \mathbf{I}_d \right)^2 \right]$

$$+ \frac{\widetilde{R}}{2d} \operatorname{tr}\left[ \left( (\boldsymbol{A}^T\boldsymbol{A} + \delta\mathbf{I}_d)^{-1} \left[ \mathbf{I}_d - e^{-(\boldsymbol{A}^T\boldsymbol{A} + \delta\mathbf{I}_d)\Gamma(t)} \right] \boldsymbol{A}^T \right)^2 \right]$$

$$+ \frac{\widehat{R}}{2d} \operatorname{tr}\left( e^{-2(\boldsymbol{A}^T\boldsymbol{A} + \delta\mathbf{I}_d)\Gamma(t)} \right).$$

(19)

Under the learning rate assumptions in Theorem 3, the limiting GF terms simplify

$$\mathcal{L}(\mathfrak{X}_\infty^{\mathrm{gf}}) = \frac{R}{2d} \operatorname{tr}\left[ (\boldsymbol{A}^T\boldsymbol{A}) \left( \boldsymbol{A}^T\boldsymbol{A}(\boldsymbol{A}^T\boldsymbol{A} + \delta\mathbf{I}_d)^{-1} - \mathbf{I}_d \right)^2 \right]$$

$$+ \frac{\widetilde{R}}{2d} \operatorname{tr}\left[ \left( \boldsymbol{A}(\boldsymbol{A}^T\boldsymbol{A} + \delta\mathbf{I}_d)^{-1}\boldsymbol{A}^T - \mathbf{I}_n \right)^2 \right]$$

(20)

$$\mathcal{R}(\mathfrak{X}_\infty^{\mathrm{gf}}) = \frac{R}{2d} \operatorname{tr}\left[ \left( \boldsymbol{A}^T\boldsymbol{A}(\boldsymbol{A}^T\boldsymbol{A} + \delta\mathbf{I}_d)^{-1} - \mathbf{I}_d \right)^2 \right] + \frac{\widetilde{R}}{2d} \operatorname{tr}\left[ \left( (\boldsymbol{A}^T\boldsymbol{A} + \delta\mathbf{I}_d)^{-1}\boldsymbol{A}^T \right)^2 \right].$$

**Excess risk in linear regression.** In the standard linear regression setup, we suppose that $\boldsymbol{A}$ is generated by taking $n$ independent $d$-dimensional samples from a centered distribution $\mathcal{D}_f$ which we assume to be standardized (mean 0 and expected sample-norm-squared 1). We let the matrix $\boldsymbol{\Sigma}_f \in \mathbb{R}^{d \times d}$ be the feature covariance of $\mathcal{D}_f$, that is

$$\boldsymbol{\Sigma}_f \stackrel{\text{def}}{=} \mathbb{E}[\boldsymbol{a}\boldsymbol{a}^T], \quad \text{where} \quad \boldsymbol{a} \sim \mathcal{D}_f. \tag{21}$$

Suppose there is a linear ("ground truth" or "signal") function $\beta : \mathbb{R}^d \to \mathbb{R}$, which for simplicity we suppose to have $\beta(0) = 0$. In this case, we identify $\beta$ with a vector using the representation $\boldsymbol{a} \mapsto \boldsymbol{\beta}^T\boldsymbol{a}$. We suppose that our data is drawn from a distribution $\mathcal{D}$ on $\mathbb{R}^d \times \mathbb{R}$, with the property that

$$\mathbb{E}[b \,|\, \boldsymbol{a}] = \boldsymbol{\beta}^T\boldsymbol{a}, \quad \text{where} \quad (\boldsymbol{a}, b) \sim \mathcal{D},$$

and the data $\boldsymbol{a} \sim \mathcal{D}_f$.

Hence we suppose that $[\boldsymbol{A} \,|\, \boldsymbol{b}]$ is a $\mathbb{R}^{n \times d} \times \mathbb{R}^{n \times 1}$ matrix on independent samples from $\mathcal{D}$. The vector $\boldsymbol{x}_t$ represents an estimate of $\boldsymbol{\beta}$, and the population risk is

$$\mathcal{R}(\boldsymbol{x}_t) \stackrel{\text{def}}{=} \frac{1}{2} \mathbb{E}[(b - \boldsymbol{x}_t^T\boldsymbol{a})^2 | \boldsymbol{x}_t] \quad \text{where} \quad (\boldsymbol{a}, b) \sim \mathcal{D},$$

where $(\boldsymbol{a}, b)$ is an sample independent of $\boldsymbol{x}_t$. This can be evaluated in terms of the feature covariance matrix $\boldsymbol{\Sigma}_f$ and the noise $\eta^2 \stackrel{\text{def}}{=} \mathbb{E}[(b - \boldsymbol{\beta}^T\boldsymbol{a})^2]$ to give

$$\mathcal{R}(\boldsymbol{x}_t) = \frac{1}{2}\eta^2 + \frac{1}{2}(\boldsymbol{\beta} - \boldsymbol{x}_t)^T \boldsymbol{\Sigma}_f (\boldsymbol{\beta} - \boldsymbol{x}_t). \tag{22}$$

It is important to note that the sequence $\{\boldsymbol{x}_{\lfloor tn \rfloor}\}_{t \geq 0}$ is generated from the iterates of SGD applied to the $\ell^2$-regularized least-squares problem (1).

In the case that $(\boldsymbol{a}, b)$ is jointly Gaussian, it follows that we may represent

$$\boldsymbol{a} = \boldsymbol{\Sigma}_f^{1/2}\boldsymbol{z}, \quad b = \boldsymbol{\beta}^T\boldsymbol{a} + \eta w, \quad \text{where} \quad (\boldsymbol{z}, w) \sim N(0, \mathbf{I}_d \oplus 1).$$

Therefore, it follows that the iterates $\boldsymbol{x}_{\lfloor nt \rfloor}$ are generated from the SGD algorithm applied to the problem:

$$\min_{\boldsymbol{x}} \frac{1}{2}\|\boldsymbol{A}\boldsymbol{x} - \boldsymbol{b}\|_2^2 + \frac{\delta}{2}\|\boldsymbol{x}\|_2^2 \quad \text{where} \quad \boldsymbol{b} = \boldsymbol{A}\boldsymbol{\beta} + \widehat{\eta}\boldsymbol{w},$$

and the vector $\boldsymbol{w}$ is iid $N(0, 1)$ random variables, independent of $\boldsymbol{A}$. This is also known as the generative model with noise.

Moreover, if $\mathcal{D}$ satisfies Assumption 7 (with $\boldsymbol{\Sigma} = \boldsymbol{\Sigma}_f$) then the population risk $\mathcal{R}(\boldsymbol{x}_{\lfloor tn \rfloor})$ is well approximated by $\Omega$:

**Theorem 7.** *Suppose $(\boldsymbol{A}, \boldsymbol{b})$ satisfy Assumption 7, $\delta > 0$ and $\boldsymbol{x}_0$ is iid centered subgaussian with $\mathbb{E}\left\|\boldsymbol{x}_0\right\|^2 = \widehat{R}$. For some $\epsilon > 0$, for all $T > 0$, and for all $D > 0$ there is a $C > 0$ such that*

$$
\Pr\left(\sup_{0 \le t \le T} \left\| \begin{pmatrix} \mathcal{L}(\boldsymbol{x}^{sgd}_{\lfloor tn \rfloor}) \\ \mathcal{R}(\boldsymbol{x}^{sgd}_{\lfloor tn \rfloor}) \end{pmatrix} - \begin{pmatrix} \Psi_t \\ \Omega_t \end{pmatrix} \right\|_2 > d^{-\epsilon} \right) \le C d^{-D},
$$

*where $\Psi_t$ solves (10) and $\Omega_t$ solves (9) with $\mathcal{R}$ given by (22).*

We remark that under Assumption 7 (and in-distribution) that $\eta^2 = \frac{\widetilde{R}}{d}$. In the case of out-of-distribution regression (see section below), we have that $\eta^2 \ne \frac{\widetilde{R}}{d}$ as the $\eta$ represents the population noise.

The loss function $\mathcal{L}$ evaluated at GF is the same as in (19) as is the limiting loss $\Omega_\infty$. For the test risk $\mathcal{R}$ in (22) evaluated at GF, we have the following expressions for

$$
\mathcal{R}(\mathfrak{X}^{\mathrm{gf}}_{\Gamma(t)}) = \frac{R}{2d} \operatorname{tr}\left( \boldsymbol{\Sigma}_f \Big( C(t) \boldsymbol{A}^T \boldsymbol{A} - \mathbf{I}_d \Big)^2 \right) + \frac{\widetilde{R}}{2d} \operatorname{tr}\left( \boldsymbol{\Sigma}_f \boldsymbol{A}^T \boldsymbol{A} C^2(t) \right)
$$

$$
+ \frac{\widehat{R}}{2d} \operatorname{tr}\left( \boldsymbol{\Sigma}_f \exp\left( -2(\boldsymbol{A}^T \boldsymbol{A} + \delta \mathbf{I}_d) \Gamma(t) \right) \right) + \frac{1}{2}\eta^2
$$

$$
\text{and} \quad \mathcal{R}(\mathfrak{X}^{\mathrm{gf}}_\infty) = \frac{R}{2d} \operatorname{tr}\left( \boldsymbol{\Sigma}_f \big( \boldsymbol{A}^T \boldsymbol{A} (\boldsymbol{A}^T \boldsymbol{A} + \delta \mathbf{I}_d)^{-1} - \mathbf{I}_d \big)^2 \right) \tag{23}
$$

$$
+ \frac{\widetilde{R}}{2d} \operatorname{tr}\left( \boldsymbol{\Sigma}_f \boldsymbol{A}^T \boldsymbol{A} \big( \boldsymbol{A}^T \boldsymbol{A} + \delta \mathbf{I}_d \big)^{-2} \right) + \frac{1}{2}\eta^2
$$

$$
\text{where} \quad C(t) \stackrel{\text{def}}{=} (\boldsymbol{A}^T \boldsymbol{A} + \delta \mathbf{I}_d)^{-1} \Big( \mathbf{I}_d - \exp\left( -(\boldsymbol{A}^T \boldsymbol{A} + \delta \mathbf{I}_d) \Gamma(t) \right) \Big).
$$

Using Theorem 3, we conclude that in the case that $\gamma(t) \to 0$ as $t \to \infty$, the excess risk of SGD tends to 0. More interestingly, in the interpolation regime, $\mathcal{L}(\mathfrak{X}^{\mathrm{gf}}_\infty) = 0$, i.e. the empirical risk tends to 0. In this case, even without taking $\gamma \to 0$, the excess risk of SGD tends to 0. If on the other hand it does not tend to 0 (i.e., $\gamma(t) \to \gamma$), we arrive at the formula for excess risk of SGD over the ridge estimator risk:

$$
\Omega_\infty - \mathcal{R}(\mathfrak{X}^{\mathrm{gf}}_\infty) = \mathcal{L}(\mathfrak{X}^{\mathrm{gf}}_\infty) \times \frac{\gamma}{2n} \frac{\operatorname{tr}\left( (\nabla^2 \mathcal{L}) \boldsymbol{\Sigma}_f \big( \nabla^2 \mathcal{L} + \delta \mathbf{I}_d \big)^{-1} \right)}{1 - \frac{\gamma}{2n} \operatorname{tr}\left( (\nabla^2 \mathcal{L})^2 \big( \nabla^2 \mathcal{L} + \delta \mathbf{I}_d \big)^{-1} \right)}
$$

$$
= \Psi_\infty \times \frac{\gamma}{n} \operatorname{tr}\left( \frac{(\nabla^2 \mathcal{L})}{\big( \nabla^2 \mathcal{L} + \delta \mathbf{I}_d \big)} \boldsymbol{\Sigma}_f \right). \tag{24}
$$

We note that the right-hand-side is proportional to $\Psi_\infty$ (c.f. Theorem 3), and hence this excess risk due to SGD will be small if the limiting empirical risk $\Psi_\infty$ is small. This also shows that the regularization term $\delta$ interacts with the excess risk due to SGD: if the spectrum of $\nabla^2 \mathcal{R}$ is heavy in that it has slowly decaying eigenvalues, the reduction in excess risk due to the regularization regularizer $\delta$ can be large.

**(Out-of-distribution) linear regression.** As before, we suppose that the data matrix $\boldsymbol{A}$ is generated by taking $n$ independent $d$-dimensional samples from a centered distribution $\mathcal{D}_f$ with feature covariance $\boldsymbol{\Sigma}_f$ (see (21)). We also suppose, as in the previous in-distribution example, that there is a linear ("ground truth" or "signal") function $\beta : \mathbb{R}^d \to \mathbb{R}$ which we identify with the vector $\boldsymbol{\beta} \in \mathbb{R}^d$ and for which $\mathbb{E}[b|\boldsymbol{a}] = \boldsymbol{\beta}^T \boldsymbol{a}$ where $(\boldsymbol{a}, b) \sim \mathcal{D}$ and the data $\boldsymbol{a} \sim \mathcal{D}_f$. We will generate our target $b$ from the distribution $(\boldsymbol{a}, b) \sim \mathcal{D}$. We then let $\boldsymbol{x}_t$ be the iterates generated by SGD applied to the optimization problem

$$
\min_{x \in \mathbb{R}^d} \frac{1}{2} \|\boldsymbol{A}\boldsymbol{x} - \boldsymbol{b}\|^2 + \frac{\delta}{2} \|\boldsymbol{x}\|^2, \quad \text{where} \quad (\boldsymbol{a}_i, b_i) \sim \mathcal{D}.
$$

The main distinction from the previous example is that we measure our generalization error using a different distribution than $\mathcal{D}$. Explicitly, there exists another centered distribution $\widehat{\mathcal{D}_f}$ (standardized)

with covariance features matrix $\widehat{\boldsymbol{\Sigma}}_f \in \mathbb{R}^{d \times d}$ from which we generate a vector $\widehat{\boldsymbol{a}} \sim \widehat{\mathcal{D}}_f$. Moreover, we generate a test point $(\widehat{\boldsymbol{a}}, \widehat{b})$ from a new distribution $\widehat{\mathcal{D}}$ such that $\mathbb{E}[\widehat{b}|\widehat{\boldsymbol{a}}] = \boldsymbol{\beta}^T \widehat{\boldsymbol{a}}$ with the same $\boldsymbol{\beta}$ as before and the distribution $\widehat{\mathcal{D}}$ has $\widehat{\boldsymbol{a}}$-marginal $\widehat{\mathcal{D}}_f$. We measure the population risk, $\mathcal{R} : \mathbb{R}^d \to \mathbb{R}$ as

$$\mathcal{R}(\boldsymbol{x}_t) \stackrel{\text{def}}{=} \frac{1}{2} \mathbb{E}[(\widehat{b} - \boldsymbol{x}_t^T \widehat{\boldsymbol{a}})^2 | \boldsymbol{x}_t] = \frac{1}{2} \eta^2 + \frac{1}{2} (\boldsymbol{x}_t - \boldsymbol{\beta})^T \widehat{\boldsymbol{\Sigma}}_f (\boldsymbol{x}_t - \boldsymbol{\beta}) \tag{25}$$

$$\text{where} \quad \eta^2 \stackrel{\text{def}}{=} \mathbb{E}[(\widehat{b} - \boldsymbol{\beta}^T \widehat{\boldsymbol{a}})^2].$$

In this setting, we can again derive the limiting excess risk, which has a similar formula for $\mathcal{R}(\mathbf{\mathcal{X}}_{\Gamma(t)}^{\text{gf}})$ as in (23) by replacing $\boldsymbol{\Sigma}_f$ with $\widehat{\boldsymbol{\Sigma}}_f$.

**Random features.** A central example where the quasi-random assumptions hold is the random features setting, which was introduced in [66] for scaling kernel machines. Random features models provide a rich but tractable class of models to gain further insights into the generalization phenomena [51, 44, 3, 2, 77]. These models are particularly of interest because of their connection to neural networks where the number of random features corresponds to model complexity [29, 55, 38] and because of its use as a practical method for data analysis [66, 73].

We suppose that the data matrix $\boldsymbol{X}$ is generated by taking $n$ independent $n_0$-dimensional samples from a centered distribution $\mathcal{D}_f$ with feature covariance

$$\boldsymbol{\Sigma}_f \stackrel{\text{def}}{=} \mathbb{E}[\boldsymbol{X}_i^T \boldsymbol{X}_i], \qquad \text{where } \boldsymbol{X}_i \in \mathbb{R}^{1 \times n_0} \text{ and } \boldsymbol{X}_i \sim \mathcal{D}_f.$$

We suppose for simplicity that $\boldsymbol{X}$ is a data matrix having dimension $n \times n_0$ whose iid rows are drawn from a multivariate Gaussian with covariance $\boldsymbol{\Sigma}_f$ and nice covariance structure:

**Assumption 8.** *The distribution $\mathcal{D}_f$ is multivariate normal and the covariance matrix $\boldsymbol{\Sigma}_f$ of the random features data satisfies for some $C > 0$*

$$\tfrac{1}{n_0} \text{tr}(\boldsymbol{\Sigma}_f) = 1 \quad \text{and} \quad \|\boldsymbol{\Sigma}_f\|_{op} \leq C.$$

This allows $\boldsymbol{X}$ to be represented equivalently as $\boldsymbol{X} = \boldsymbol{Z}\boldsymbol{\Sigma}^{1/2}/\sqrt{n_0}$ for a iid standard Gaussian matrix $\boldsymbol{Z}$. We suppose that $\boldsymbol{W}$ is an $(n_0 \times d)$ iid feature matrix having standard Gaussian entries and independent of $\boldsymbol{Z}$ so that $\boldsymbol{Z}\boldsymbol{\Sigma}^{1/2}\boldsymbol{W}/\sqrt{n_0}$ is a matrix whose rows are standardized.

We let $\sigma$ be an activation function satisfying:

**Assumption 9.** *The activation function satisfies for $C_0, C_1 \geq 0$*

$$|\sigma'(x)| \leq C_0 e^{C_1|x|}, \quad \text{for all} \quad x \in \mathbb{R}, \quad \text{and for standard normal } Z, \quad \mathbb{E}\,\sigma(Z) = 0.$$

We note that from the outset, the growth rate of the derivative of the activation function implies a similar bound on the growth rate of the underlying activation function $\sigma$. As before, we suppose the data $[\boldsymbol{X} \mid \boldsymbol{b}]$ is arranged in the matrix $\mathbb{R}^n \times (\mathbb{R}^{n_0} \times \mathbb{R})$ where each row is an independent sample from $\mathcal{D}$. We now transform the data $\boldsymbol{X} \in \mathbb{R}^{n \times n_0}$ by putting

$$\boldsymbol{A} = \sigma(\boldsymbol{X}\boldsymbol{W}/\sqrt{n_0}) \in \mathbb{R}^{n \times d},$$

where $\boldsymbol{W} \in \mathbb{R}^{n_0 \times d}$ is a matrix independent of $[\boldsymbol{X} \mid \boldsymbol{b}]$ of independent standard normals.[3] The activation function $\sigma : \mathbb{R} \to \mathbb{R}$ is applied element-wise.

We introduce the following notation

$$\boldsymbol{\Sigma}_\sigma(\boldsymbol{W}) \stackrel{\text{def}}{=} \mathbb{E}[\sigma(\boldsymbol{X}_i\boldsymbol{W}/\sqrt{n_0})^T \sigma(\boldsymbol{X}_i\boldsymbol{W}/\sqrt{n_0}) \mid \boldsymbol{W}] \tag{26}$$

$$\text{and} \quad \widehat{\sigma}(\boldsymbol{W}) \stackrel{\text{def}}{=} \mathbb{E}[\boldsymbol{X}_i^T \sigma(\boldsymbol{X}_i\boldsymbol{W}/\sqrt{n_0})|\boldsymbol{W}].$$

The population risk, $\mathcal{R} : \mathbb{R}^d \to \mathbb{R}$ as a random variable in $\boldsymbol{X}$ and $\boldsymbol{W}$, is

$$\begin{aligned}
\mathcal{R}(\boldsymbol{x}_t) &\stackrel{\text{def}}{=} \mathbb{E}[(b - \boldsymbol{x}_t^T \sigma(\boldsymbol{X}_i\boldsymbol{W}/\sqrt{n_0}))^2 | \boldsymbol{x}_t, \boldsymbol{W}] \\
&= \eta^2 + \mathbb{E}[(\boldsymbol{X}_i\boldsymbol{\beta} - \sigma(\boldsymbol{X}_i\boldsymbol{W}/\sqrt{n_0})\boldsymbol{x}_t)^2 \mid \boldsymbol{x}_t, \boldsymbol{W}] \\
&= \eta^2 + \boldsymbol{\beta}^T \boldsymbol{\Sigma}_f \boldsymbol{\beta} + \boldsymbol{x}_t^T \boldsymbol{\Sigma}_\sigma(\boldsymbol{W})\boldsymbol{x}_t - 2\boldsymbol{\beta}^T \widehat{\sigma}(\boldsymbol{W})\boldsymbol{x}_t,
\end{aligned} \tag{27}$$

$$\text{where } (\boldsymbol{X}_i, b) \sim \mathcal{D} \text{ and } \mathbb{E}[b \mid \boldsymbol{X}_i] = \boldsymbol{X}_i\boldsymbol{\beta}.$$

---

[3] In [51], the distribution of the columns are taken as independent uniform vectors on the sphere $\sqrt{d}\,\mathbb{S}^{d-1}$. The activation function $\sigma$ is a 1-Lipschitz function from $\mathbb{R} \to \mathbb{R}$ that is applied entrywise to the underlying matrix.

The $\ell^2$-regularized least-squares problem is now

$$\min_{\boldsymbol{x}} \frac{1}{2}\|\boldsymbol{A}\boldsymbol{x} - \boldsymbol{b}\|_2^2 + \frac{\delta}{2}\|\boldsymbol{x}\|_2^2 \quad \text{where} \quad \boldsymbol{b} = \boldsymbol{X}\boldsymbol{\beta} + \eta\boldsymbol{w},$$

which is the random features regression. This should be compared to a two-layer neural network model, in which the hidden layer has dimension $n_0$. However, the hidden layer weights are simply generated randomly in advance and are left untrained. The optimization is only performed on the final layers' weights ($\boldsymbol{x}$).

**Theorem 8.** *Suppose that $n, d, n_0$ are proportionally related. Suppose that the data matrix $\boldsymbol{X}$ satisfies Assumption 8, and the random features $\boldsymbol{W}$ are iid standard normal. Suppose $\boldsymbol{b} = \boldsymbol{X}\boldsymbol{\beta} + \eta\boldsymbol{w}$ with $\boldsymbol{\beta}, \boldsymbol{w}$ independent isotropic subgaussian vectors with $\mathbb{E}\|\boldsymbol{\beta}\|_2^2 = 1/n_0$ and $\mathbb{E}\|\boldsymbol{w}\|_2^2 = 1$ and $\eta$ bounded independent of $n$. Suppose the activation function satisfies Assumption 9. Suppose the initialization $\boldsymbol{x}_0$ is iid centered subgaussian with $\mathbb{E}\|\boldsymbol{x}_0\|_2^2 = \widehat{R}$. Then for some $\epsilon > 0$, for all $T > 0$, and for all $D > 0$ there is a $C > 0$ such that*

$$\Pr\left(\sup_{0 \leq t \leq T} \left\| \begin{pmatrix} \mathcal{L}(\boldsymbol{x}_{\lfloor tn \rfloor}^{sgd}) \\ \mathcal{R}(\boldsymbol{x}_{\lfloor tn \rfloor}^{sgd}) \end{pmatrix} - \begin{pmatrix} \Psi_t \\ \Omega_t \end{pmatrix} \right\| > d^{-\epsilon} \right) \leq Cd^{-D},$$

*where $\Psi_t$ solves (10) and $\Omega_t$ solves (9) with $\mathcal{R}$ given by (27).*

Finally, as in (24) we derive the excess risk of SGD ($\gamma(t) \to \gamma$) over ridge regression:

$$\begin{aligned}
\Omega_\infty - \mathcal{R}(\boldsymbol{\mathcal{X}}_\infty^{\text{gf}}) &= \mathcal{L}(\boldsymbol{\mathcal{X}}_\infty^{\text{gf}}) \times \frac{\gamma}{2n} \frac{\text{tr}\big((\nabla^2\mathcal{L})\boldsymbol{\Sigma}_\sigma(\boldsymbol{W})(\nabla^2\mathcal{L} + \delta\mathbf{I}_d)^{-1}\big)}{1 - \frac{\gamma}{2n}\text{tr}\big((\nabla^2\mathcal{L})^2(\nabla^2\mathcal{L} + \delta\mathbf{I}_d)^{-1}\big)} \\
&= \Psi_\infty \times \frac{\gamma}{2n}\text{tr}\left(\frac{(\nabla^2\mathcal{L})}{(\nabla^2\mathcal{L} + \delta\mathbf{I}_d)}(\boldsymbol{\Sigma}_\sigma(\boldsymbol{W}))\right).
\end{aligned} \tag{28}$$

### B.1.1 Discussion of Theorem 1 and motivating examples

In this section, we discuss the equivalence of SGD and homogenized SGD under quadratic statistics $\mathcal{R}$ satisfying Assumption 1 and quasi-random assumptions on the data matrix $\boldsymbol{A}$, initialization $\boldsymbol{x}_0$, and target vector $\boldsymbol{b}$ (see Appendix B). As the proof of Theorem 1 is quite mathematically involved and it does not add to the interpretation of the risk trajectories, we relegate this proof to [61, Theorem 1.3].

The proofs of Theorems 6, 7, and 8 follow immediately from Theorem 1 and Theorem 2. In each of the cases, the extra assumptions on the initialization, signal, and data matrix simplify the GF terms in (8) and (9).

## C   Volterra Equation for Streaming SGD

In this section we introduce constant learning rate ($\gamma(t) \equiv \gamma$) streaming SGD. Let $(\boldsymbol{a}_k, b_k)_{k=1}^\infty$ be iid samples from a $\mathbb{R}^d \times \mathbb{R}$-dimensional distribution $\mathcal{D}$. Streaming SGD using data from $\mathcal{D}$, which we denote $\mathcal{D}$-SGD, is the algorithm

$$\boldsymbol{s}_{k+1} = \boldsymbol{s}_k - \gamma\boldsymbol{a}_k(\boldsymbol{a}_k \cdot \boldsymbol{s}_k - b_k) \quad \text{for} \quad k \in \{0, 1, 2, \dots\}. \tag{29}$$

This naturally describes one-pass SGD, in which data points are used only once. If $\mathcal{R}$ is the expected risk (i.e. population risk), and if $\mathcal{R}$ is given by $\frac{1}{2}\mathbb{E}(\boldsymbol{a} \cdot \boldsymbol{x} - b)^2$ with $(\boldsymbol{a}, b) \sim \mathcal{D}$, then $\mathcal{D}$-SGD is directly solving the population risk minimization. This is an idealized situation as one does not have access to infinite data in practice.

**Deterministic behavior of streaming risks and comparison to HSGD.**   $\mathcal{D}$-SGD can encompass multi-pass SGD by letting $(\boldsymbol{a}_k, b_k)_{k=1}^n$ be the first $n$ iid samples from $\mathcal{D}$ and considering $\widehat{\mathcal{D}}$-SGD for

$$\widehat{\mathcal{D}}_n \stackrel{\text{def}}{=} \frac{1}{n}\sum_{k=1}^n \delta_{(\boldsymbol{a}_k, b_k)}. \tag{30}$$

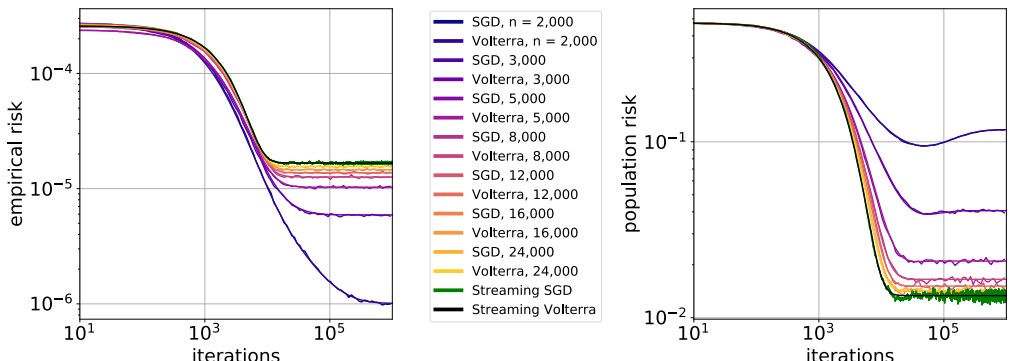

Figure 5: **Single runs of SGD vs. HSGD (Volterra) for a simple Gaussian linear regression problem**, with increasing number of samples $n$ and $d = 2000$. Empirical risk (left) increases monotonically with $n$ to a limit while population risk (right) decreases monotonically in $n$. Streaming corresponds to $n = \infty$. Covariance of Gaussian samples is $\mathbf{I}_d$, with target given by $b = a \cdot \beta + \eta Z$ for $\eta = 0.2$ and $\beta$ a unit vector and $Z \sim N(0, 1)$. For consistency across sample sizes, time is measured in iterations.

This means that a sample from $\widehat{\mathcal{D}}_n$ (*conditionally* on the dataset) is distributed like $(\boldsymbol{a}_I, b_I)$, where $I$ is an uniformly random choice of index.

To enable a comparison of streaming SGD to multi-pass SGD, we suppose for some $n$ the matrix $[\boldsymbol{A} \,|\, \boldsymbol{b}]$ is $n \times (d+1)$ matrix whose rows are iid samples from $\mathcal{D}$. We construct the empirical risk $\mathcal{L}$ from these $n$ samples as in (2). We then define the streaming loss

$$\mathcal{S}(\boldsymbol{x}) \stackrel{\text{def}}{=} \tfrac{1}{2} \, \mathbb{E}[(\boldsymbol{a} \cdot \boldsymbol{x} - b)^2] = \tfrac{1}{2n} \, \mathbb{E} \, \|\boldsymbol{A}\boldsymbol{x} - \boldsymbol{b}\|^2 = \tfrac{1}{n} \, \mathbb{E} \, \mathcal{L}(\boldsymbol{x}), \quad \text{where} \quad (\boldsymbol{a}, b) \sim \mathcal{D}.$$

The associated homogenized SGD representation, which we call homogenized $\mathcal{D}$-SGD, is

$$\mathrm{d}\boldsymbol{Y}_t \stackrel{\text{def}}{=} -\gamma(\nabla\mathcal{S}(\boldsymbol{Y}_t)) \, \mathrm{d}t + \gamma\sqrt{2\mathcal{S}(\boldsymbol{Y}_t)(\nabla^2\mathcal{S})} \, \mathrm{d}\boldsymbol{B}_t \quad \text{for} \quad t \geq 0. \tag{31}$$

This naturally leads to Volterra dynamics in which $\nabla^2\mathcal{L}$ is replaced by $\nabla^2\mathcal{S}$ in Eqs. (8)-(10) whose solution we denote with $\Psi_t^s$ and $\Omega_t^s$ (see Appendix D).

We prove a weak equivalence between HSGD and homogenized $\mathcal{D}$-SGD in the following theorem.

**Theorem 9.** *Suppose $\boldsymbol{X}_0 = \boldsymbol{Y}_0$. After rescaling time,*

$$\lim_{n \to \infty} \boldsymbol{X}_{t/n} = \boldsymbol{Y}_t \quad \text{uniformly on compact sets of time a.s.}$$

*Furthermore, GF converges, $\mathcal{X}_{\gamma t/n}^{gf} \to \mathcal{X}_{\gamma t}^{s\text{-}gf}$, and the Volterra equations converge, $\Psi_{t/n} \to \Psi_t^s$ and $\Omega_{t/n} \to \Omega_t^s$, uniformly on compact sets of time.*

We expect this to hold in much greater generality as suggested by Figures 1 and 5. This is an immediate consequence of the law of large numbers due to which $\frac{1}{n}\mathcal{L}$ converges almost surely to $\mathcal{S}$; the applications in which we are typically interested would take time large as a function of $n$ (the numerical results are extremely strong, see Figures 5 and 1), and we leave a deeper mathematical investigation of this point as an open question. We also note that for $t$ very large with $n$, there is likely another behavior that takes hold. For example, when $t = n$, observe that $\boldsymbol{X}_{t/n}$ will have used approximately $(1 - 1/e)n$ samples, whereas $\boldsymbol{Y}_n$ will have used $n$, and hence we expect a breakdown in the connection. Other works also examined the expected risk of streaming including [9, 86] but not as the limit of the dynamics of multi-pass SGD as we have done.

**Features of generalization.** While there are connections between streaming and multi-pass SGD, certain behavior is only accessible in the multi-pass setting. For example, early stopping can be a useful ingredient in avoiding overfitting when learning overparameterized models. However, late-time overfitting is only observable with multi-pass SGD and does not occur for streaming SGD (see the nonmonotonicity in population risk in Fig. 5).

Similarly, comparing streaming and multi-pass SGD has been suggested as a method for analyzing generalization. The bootstrap risk was introduced in [54] as an intepretable component in a decomposition of the population risk. It is defined as the excess risk of SGD for ERM $\mathcal{L}$ with $n$ iid samples from $\mathcal{D}$ when compared to $\mathcal{D}$-SGD, i.e.

$$\underbrace{\Omega_{t/n} - \Omega_t^s}_{\text{bootstrap risk}} = \underbrace{\mathcal{R}(\mathcal{X}_{\gamma t/n}^{\text{gf}}) - \mathcal{R}(\mathcal{X}_{\gamma t}^{\text{s-gf}})}_{\text{model risk}} + \underbrace{\int_0^{t/n} K(t/n, u; \nabla^2\mathcal{R})\Psi_u \, du - \int_0^t K^s(t, u; \nabla^2\mathcal{R})\Psi_u^s \, du}_{\text{SGD bootstrap risk}}.$$

(32)

Thus the Volterra equations can be used to give an exact expression for the bootstrap risk. In particular, we can predict the iteration at which the bootstrap risk becomes large as the streaming and multi-pass risks bifurcate. This allows for quantitative prediction of the stopping time in Claim 1 of [54], a central conjecture of their paper.

On taking time to infinity, we can further evaluate the SGD bootstrap risk.

**Theorem 10.** *If $\gamma \leq \min\{2(\frac{1}{n}\operatorname{tr}(\boldsymbol{A}^T\boldsymbol{A}))^{-1}, 2(\operatorname{tr}\nabla^2\mathcal{S})^{-1}\}$, then the limiting bootstrap risk is given by*

$$\lim_{t\to\infty} \Omega_{t/n} - \Omega_t^s = \mathcal{R}(\mathcal{X}_\infty^{\text{gf}}) - \mathcal{R}(\mathcal{X}_\infty^{\text{s-gf}}) + \frac{\gamma\Psi_\infty}{2n} \times \operatorname{tr}\left\{\frac{(\nabla^2\mathcal{R})(\nabla^2\mathcal{L})}{\nabla^2\mathcal{L} + \delta\boldsymbol{I}_d}\right\}$$

*Here $\Psi_\infty$ is the limiting training losses given by Theorem 3, i.e.*

$$\Psi_\infty = \mathcal{L}(\mathcal{X}_\infty^{\text{gf}}) \times \left(1 - \frac{\gamma}{2n}\operatorname{tr}\left\{\frac{(\nabla^2\mathcal{L})^2}{\nabla^2\mathcal{L} + \delta\boldsymbol{I}_d}\right\}\right)^{-1}.$$

## D  The Volterra SLD class and Concentration of HSGD

First, we state the Volterra equation for streaming SGD that we referenced in the main text.

> **Volterra Dynamics (streaming).** The following deterministic dynamical system is the high-dimensional equivalent for $\mathcal{L}(\boldsymbol{Y}_t)$ and $\mathcal{R}(\boldsymbol{Y}_t)$, respectively,
>
> $$\Psi_t^s = \mathcal{S}(\mathcal{X}_{\gamma t}^{\text{s-gf}}) + \int_0^t K^s(t, u; \nabla^2\mathcal{S})\Psi_u^s \, du, \quad \text{for} \quad t \geq 0 \qquad \text{(Empirical risk)} \qquad (33)$$
>
> $$\Omega_t^s = \mathcal{R}(\mathcal{X}_{\gamma t}^{\text{s-gf}}) + \int_0^t K^s(t, u; \nabla^2\mathcal{R})\Psi_u^s \, du \quad \text{for} \quad t \geq 0 \qquad \text{(Population risk)} \qquad (34)$$
>
> where the kernel $K$, for any $d \times d$ matrix $\boldsymbol{P}$, is
>
> $$K^s(t, u; \boldsymbol{P}) = \gamma^2 \operatorname{tr}\left((\nabla^2\mathcal{S})\boldsymbol{P}\exp(-2\gamma(\nabla^2\mathcal{S})(t-u))\right) \qquad (35)$$
>
> and *GF for streaming*, $\mathcal{X}_t^{\text{s-gf}}$, is the solution of
>
> $$d\mathcal{X}_t^{\text{s-gf}} \overset{\text{def}}{=} -\nabla\mathcal{S}(\mathcal{X}_t^{\text{s-gf}}) \, dt \quad \text{and} \quad \mathcal{X}_0^{\text{s-gf}} = \boldsymbol{Y}_0.$$

We will enlarge the class of SLDs that we consider to what we will call as the *Volterra SLD class* defined as

$$d\boldsymbol{X}_t \overset{\text{def}}{=} -\gamma(t)\nabla f(\boldsymbol{X}_t) \, dt + \gamma(t)\sqrt{\mathcal{L}(\boldsymbol{X}_t)\mathcal{M}_t + \mathcal{A}_t} \, d\boldsymbol{B}_t, \qquad (36)$$

where $\mathcal{M}_t$ and $\mathcal{A}_t$ are two deterministic positive definite functions which we assume to be normalized to satisfy:

**Assumption 10.** *The covariance processes $\mathcal{M}$ and $\mathcal{A}$ satisfy for some absolute constants $c > 0$ and $\epsilon > 0$*

$$\sup_{t\geq 0}\left(\operatorname{tr}\mathcal{M}_t + \operatorname{tr}\mathcal{A}_t\right) \leq c < \infty \quad \text{and} \quad \sup_{t\geq 0}\left(\|\mathcal{M}_t\|_{op} + \|\mathcal{A}_t\|_{op}\right) < d^{-\epsilon}.$$

The $\mathcal{M}_t$ represents noise in the data either because the data is randomly sampled or the data is transformed by multiplicative transformation. In contrast $\mathcal{A}_t$ represents an additive noise at each step. It is any noise which does not multiple the state $\boldsymbol{x}$, for example, label noise.

**Volterra SLD class.** The Volterra SLD class is so-named because the expected loss satisfies a Volterra type integral equation. Define for $t \geq s \geq 0$ and positive semidefinite $\boldsymbol{P}$,

$$\Gamma(t) = \int_0^t \gamma(s) \, \mathrm{d}s, \text{ and } \begin{cases} K(t,s;\boldsymbol{P}) = \gamma^2(s) \operatorname{tr}\left( \mathcal{M}_s \boldsymbol{P} \exp\left(-2(\boldsymbol{A}^T\boldsymbol{A} + \delta \mathbf{I}_d)(\Gamma(t) - \Gamma(s))\right) \right) \\[2mm] A(t,s;\boldsymbol{P}) = \gamma^2(s) \operatorname{tr}\left( \mathcal{A}_s \boldsymbol{P} \exp\left(-2(\boldsymbol{A}^T\boldsymbol{A} + \delta \mathbf{I}_d)(\Gamma(t) - \Gamma(s))\right) \right) \end{cases}. \quad (37)$$

We shall suppose throughout that $\mathcal{X}_t^{\mathrm{gf}}$ is the canonical GF

$$\mathrm{d}\mathcal{X}_t^{\mathrm{gf}} = -\nabla f(\mathcal{X}_t^{\mathrm{gf}}) \quad \text{and} \quad \mathcal{X}_0^{\mathrm{gf}} = \boldsymbol{X}_0.$$

The loss $\mathcal{L}(\boldsymbol{X}_t)$ concentrates around the solution $\Psi_t$ of the Volterra integral equation (see Theorem 11 for a precise formulation):

$$\Psi_t = \mathcal{L}\left(\mathcal{X}_{\Gamma(t)}^{\mathrm{gf}}\right) + \int_0^t A(t,s;\boldsymbol{A}^T\boldsymbol{A}) \, \mathrm{d}s + \int_0^t K(t,s;\boldsymbol{A}^T\boldsymbol{A})\Psi_s \, \mathrm{d}s. \quad (38)$$

We give a formal proof of the concentration result in Section D.1. For other quadratics $\mathcal{R}$, the loss $\mathcal{R}(\boldsymbol{X}_t)$ concentrates around

$$\Omega_t = \mathcal{R}\left(\mathcal{X}_{\Gamma(t)}^{\mathrm{gf}}\right) + \int_0^t A(t,s;\nabla^2\mathcal{R}) \, \mathrm{d}s + \int_0^t K(t,s;\nabla^2\mathcal{R})\Psi_s \, \mathrm{d}s. \quad (39)$$

## D.1 Volterra Concentration

In this section, we prove the concentration result. In this section, we prove that homogenized SGD concentrates around its mean provided that the expected loss in is in the Volterra SLD class. The result, in this section, Theorem 11, is more general than Theorem 2 which follows by setting $\mathcal{M}_t \equiv \frac{1}{n}\nabla^2\mathcal{L}$ and $\mathcal{A}_t \equiv 0$.

**Theorem 11.** *Under Assumption 10, the loss $\mathcal{L}(\boldsymbol{X}_t)$ concentrates around $\Psi_t$ the solution of the Volterra equation*

$$\Psi_t = \mathcal{L}\left(\mathcal{X}_{\Gamma(t)}^{gf}\right) + \int_0^t A(t,s;\boldsymbol{A}^T\boldsymbol{A}) \, \mathrm{d}s + \int_0^t K(t,s;\boldsymbol{A}^T\boldsymbol{A})\Psi_s \, \mathrm{d}s,$$

*in that for any $T, D > 0$ there is a $C(T, D, \|\boldsymbol{A}\|_{op}, \|\boldsymbol{b}\|, \epsilon) > 0$ sufficiently large that*

$$\Pr\left[ \sup_{0 \leq t \leq T} |\mathcal{L}(\boldsymbol{X}_t) - \Psi_t| > Cd^{-\epsilon/2} \right] \leq Cd^{-D}.$$

*Furthermore, for another quadratic $\mathcal{R}(\boldsymbol{x}) = 1/2\boldsymbol{x}^T\boldsymbol{T}\boldsymbol{x} + \boldsymbol{u}^T\boldsymbol{x} + c$ with $\boldsymbol{T}$ symmetric matrix having $\|\nabla^2\mathcal{R}\|_{op} \leq C, \|\nabla\mathcal{R}\|_2 \leq C, \|\nabla\mathcal{R}(0)\| \leq 1$ are independent of the Brownian motion,*

$$\Pr\left[ \sup_{0 \leq t \leq T} \left| -\mathcal{R}(\boldsymbol{X}_t) + \mathcal{R}\left(\mathcal{X}_{\Gamma(t)}^{gf}\right) + \int_0^t A(t,s;\nabla^2\mathcal{R}) \, \mathrm{d}s + \int_0^t K(t,s;\nabla^2\mathcal{R})\Psi_s \, \mathrm{d}s \right| > Cd^{-\epsilon/2} \right] \leq Cd^{-D}.$$

*Proof. Step 1. Volterra equation for the expected loss.*

Define $\boldsymbol{Q}_t = \exp((\boldsymbol{A}^T\boldsymbol{A} + \delta\mathbf{I}_d)\Gamma(t))$ and apply Itô's rule to $\boldsymbol{Q}_t\boldsymbol{X}_t$, derive

$$\mathrm{d}(\boldsymbol{Q}_t\boldsymbol{X}_t) = \gamma(t)\boldsymbol{Q}_t\boldsymbol{A}^T\boldsymbol{b} \, \mathrm{d}t + \gamma(t)\boldsymbol{Q}_t\sqrt{\mathcal{L}(\boldsymbol{X}_t)\mathcal{M}_t + \mathcal{A}_t} \, \mathrm{d}\boldsymbol{B}_t.$$

Hence

$$\boldsymbol{Q}_t\boldsymbol{X}_t = \boldsymbol{Q}_0\boldsymbol{X}_0 + \int_0^t \gamma(s)\boldsymbol{Q}_s\boldsymbol{A}^T\boldsymbol{b} \, \mathrm{d}s + \int_0^t \gamma(s)\boldsymbol{Q}_s\sqrt{\mathcal{L}(\boldsymbol{X}_s)\mathcal{M}_s + \mathcal{A}_s} \, \mathrm{d}\boldsymbol{B}_s.$$

Note that on setting $\mathcal{M}_s = \mathcal{A}_s = 0$, this gives GF $\mathcal{X}_{\Gamma(t)}^{\mathrm{gf}}$, and hence we have representation

$$\boldsymbol{X}_t = \mathcal{X}_{\Gamma(t)}^{\mathrm{gf}} + \boldsymbol{Q}_t^{-1}\int_0^t \gamma(s)\boldsymbol{Q}_s\sqrt{\mathcal{L}(\boldsymbol{X}_s)\mathcal{M}_s + \mathcal{A}_s} \, \mathrm{d}\boldsymbol{B}_s.$$

Expanding the quadratic,

$$\mathcal{L}(\boldsymbol{X}_t) = \mathcal{L}\big(\mathfrak{X}^{\mathrm{gf}}_{\Gamma(t)}\big) + \nabla\mathcal{L}(\mathfrak{X}^{\mathrm{gf}}_{\Gamma(t)})^T \boldsymbol{Q}_t^{-1} \int_0^t \gamma(s)\boldsymbol{Q}_s \sqrt{\mathcal{L}(\boldsymbol{X}_s)\mathcal{M}_s + \mathcal{A}_s}\, \mathrm{d}\boldsymbol{B}_s$$
$$+ \frac{1}{2}\left\| \boldsymbol{A}\boldsymbol{Q}_t^{-1}\int_0^t \gamma(s)\boldsymbol{Q}_s\sqrt{\mathcal{L}(\boldsymbol{X}_s)\mathcal{M}_s + \mathcal{A}_s}\, \mathrm{d}\boldsymbol{B}_s \right\|^2. \tag{40}$$

It follows that with $\mathcal{F}_t$ the sigma-algebra generated by $(\boldsymbol{X}_0, \mathcal{L}, (\boldsymbol{B}_s : 0 \le s \le t))$ if we compute the $\mathcal{F}_0$-conditional expectation, the Brownian integral vanishes, and we are left with two contributions from the second norm-squared process

$$\mathbb{E}\big[\mathcal{L}(\boldsymbol{X}_t) \mid \mathcal{F}_0\big] = \mathcal{L}\big(\mathfrak{X}^{\mathrm{gf}}_{\Gamma(t)}\big) + \frac{1}{2}\int_0^t \gamma^2(s)\,\mathrm{tr}\Big(\boldsymbol{A}^T\boldsymbol{A}\boldsymbol{Q}_t^{-2}\boldsymbol{Q}_s^2\mathcal{A}_s\Big)\,\mathrm{d}s$$
$$+ \frac{1}{2}\int_0^t \gamma^2(s)\,\mathrm{tr}\Big(\boldsymbol{A}^T\boldsymbol{A}\boldsymbol{Q}_t^{-2}\boldsymbol{Q}_s^2\mathcal{M}_s\Big)\,\mathbb{E}\big[\mathcal{L}(\boldsymbol{X}_s) \mid \mathcal{F}_0\big]\,\mathrm{d}s. \tag{41}$$

This is the claimed Volterra equation (see e.g., $\Psi_t \overset{\text{def}}{=} \mathbb{E}\left[\mathcal{L}(\boldsymbol{X}_t) \mid \mathcal{F}_0\right]$ in (8); here $\mathcal{A}_t = 0$.)

*Step 2. High probability boundedness of $\mathcal{L}$.* We observe before beginning that many of the quantities that appear in the expressions above are bounded. The GF $\mathfrak{X}^{\mathrm{gf}}_{\Gamma(t)}$ satisfies a uniform bound, solely in terms of its initial conditions and, in particular, the boundedness of $\mathcal{L}(\mathfrak{X}^{\mathrm{gf}}_{\Gamma(t)})$ satisfies $\mathcal{L}(\mathfrak{X}^{\mathrm{gf}}_{\Gamma(t)}) \le \mathcal{L}(\boldsymbol{X}_0)$. The matrix $\boldsymbol{Q}_t^{-1}\boldsymbol{Q}_s$ is uniformly bounded in norm by 1 for all $t \ge s$. We have also assumed that $\|\boldsymbol{A}\|_{\mathrm{op}}$ and $\|\boldsymbol{b}\|_2$ are bounded. By applying Itô's formula to the norm $u_t = \frac{1}{2}\|\boldsymbol{X}_t\|_2^2$, we have from (36) that

$$\mathrm{d}u_t = -\gamma(t)\boldsymbol{X}_t^T(\boldsymbol{A}^T(\boldsymbol{A}\boldsymbol{X}_t - \boldsymbol{b}))\,\mathrm{d}t + \gamma(t)\boldsymbol{X}_t^T\sqrt{\mathcal{L}(\boldsymbol{X}_t)\mathcal{M}_t + \mathcal{A}_t}\,\mathrm{d}\boldsymbol{B}_t + \frac{\gamma^2(t)}{2}\,\mathrm{tr}\big(\mathcal{L}(\boldsymbol{X}_t)\mathcal{M}_t + \mathcal{A}_t\big)\,\mathrm{d}t$$

From the norm boundedness of $\boldsymbol{A}$ and $\boldsymbol{b}$, we can bound $\mathcal{L}(\boldsymbol{X}_t) \le 2\|\boldsymbol{A}\|_{\mathrm{op}}^2 u_t + 2\|\boldsymbol{b}\|_2^2 \le C(u_t + 1)$. Likewise, increasing $C$ as need be, using the boundedness of $\gamma(t)$, $\mathrm{tr}\,\mathcal{M}_t$ and $\mathrm{tr}\,\mathcal{A}_t$, we conclude

$$\mathrm{d}\langle u_t \rangle = \gamma^2(t)\,\mathrm{tr}\big(\boldsymbol{X}_t\boldsymbol{X}_t^T(\mathcal{L}(\boldsymbol{X}_t)\mathcal{M}_t + \mathcal{A}_t)\big) \le C(u_t + 1)^2.$$

It follows that $z_t \overset{\text{def}}{=} \log(1 + u_t) - Ct$ is supermartingale with $\langle z_t \rangle \le C$ for some sufficiently large $C$ and all $t \le T$. Hence with probability at least $1 - e^{-2C(T)(\log d)^{3/2}}$,

$$z_t \le (\log d)^{3/4}$$

for all $t \le T$. On this same event it follows for a sufficiently large cosntant $C > 0$

$$f(t) = \mathcal{L}(\boldsymbol{X}_t) + \delta u_t \quad \text{and} \quad \mathcal{L}(\boldsymbol{X}_t) \le C(u_t + 1) \le C^2 e^{Ct + (\log d)^{3/4}}$$

for all $t \le T$.

*Step 3. Concentration of the loss.* We may now control the difference of the loss from its expectation. Specifically, in comparing (40) and (41), we may express the difference $\Delta_t \overset{\text{def}}{=} \mathcal{L}(\boldsymbol{X}_t) - \mathbb{E}[\mathcal{L}(\boldsymbol{X}_t) \mid \mathcal{F}_0]$ as

$$\Delta_t = \boldsymbol{M}_t^{(1)} + \boldsymbol{M}_t^{(2)} + \frac{1}{2}\int_0^t \gamma^2(s)\,\mathrm{tr}\Big(\boldsymbol{A}^T\boldsymbol{A}\boldsymbol{Q}_t^{-2}\boldsymbol{Q}_s^2\mathcal{M}_s\Big)\Delta_s\,\mathrm{d}s, \quad \text{where}$$

$$\boldsymbol{M}_t^{(1)} = \nabla\mathcal{L}(\mathfrak{X}^{\mathrm{gf}}_{\Gamma(t)})^T\boldsymbol{Q}_t^{-1}\int_0^t \gamma(s)\boldsymbol{Q}_s\sqrt{\mathcal{L}(\boldsymbol{X}_s)\mathcal{M}_s + \mathcal{A}_s}\,\mathrm{d}\boldsymbol{B}_s, \quad \text{and}$$

$$\boldsymbol{M}_t^{(2)} = \frac{1}{2}\left\| \boldsymbol{A}\boldsymbol{Q}_t^{-1}\int_0^t \gamma(s)\boldsymbol{Q}_s\sqrt{\mathcal{L}(\boldsymbol{X}_s)\mathcal{M}_s + \mathcal{A}_s}\,\mathrm{d}\boldsymbol{B}_s \right\|^2 \tag{42}$$
$$- \frac{1}{2}\int_0^t \gamma^2(s)\,\mathrm{tr}\Big(\boldsymbol{A}^T\boldsymbol{A}\boldsymbol{Q}_t^{-2}\boldsymbol{Q}_s^2(\mathcal{L}(\boldsymbol{X}_s)\mathcal{M}_s + \mathcal{A}_s)\Big)\,\mathrm{d}s.$$

We claim that both processes $\boldsymbol{M}_t^{(1)}$ and $\boldsymbol{M}_t^{(2)}$ are small, whose proof we defer. Specifically, with probability $1 - C(T)e^{-(\log d)^{3/2}}$ we have

$$\max_{0 \le t \le T}\big\{|\boldsymbol{M}_t^{(1)}| + |\boldsymbol{M}_t^{(2)}|\big\} \le d^{-3\epsilon/4}.$$

From the uniform boundedness in norm of $\boldsymbol{A}^T \boldsymbol{A}$, we then conclude from (42) for all $t \leq T$

$$|\Delta_t| \leq d^{-3\epsilon/4} + \int_0^t \|\boldsymbol{A}^T \boldsymbol{A}\|_{\mathrm{op}} |\Delta_s| \, \mathrm{d}s.$$

Using Gronwall's inequality,

$$|\Delta_t| \leq \|\boldsymbol{A}^T \boldsymbol{A}\|_{\mathrm{op}}^{-1} \left( e^{\|\boldsymbol{A}^T \boldsymbol{A}\|_{\mathrm{op}} t} - 1 \right) d^{-3\epsilon/4}.$$

Thus we conclude by increasing the constants in the claimed bound that the desired inequality holds.

*Step 4 (Deferred). Concentration of the martingales.* We introduce two martingales, for each fixed $t \in [0, T]$,

$$\boldsymbol{M}_u^{(1,t)} = \nabla \mathcal{L}(\mathfrak{X}_{\Gamma(t)}^{\mathrm{gf}})^T \boldsymbol{Q}_t^{-1} \int_0^u \gamma(s) \boldsymbol{Q}_s \sqrt{\mathcal{L}(\boldsymbol{X}_s) \mathfrak{M}_s + \mathcal{A}_s} \, \mathrm{d}\boldsymbol{B}_s.$$

$$\boldsymbol{M}_u^{(2,t)} = \frac{1}{2} \left\| \boldsymbol{A} \boldsymbol{Q}_t^{-1} \int_0^u \gamma(s) \boldsymbol{Q}_s \sqrt{\mathcal{L}(\boldsymbol{X}_s) \mathfrak{M}_s + \mathcal{A}_s} \, \mathrm{d}\boldsymbol{B}_s \right\|^2$$
$$- \frac{1}{2} \int_0^u \gamma^2(s) \operatorname{tr} \left( \boldsymbol{A}^T \boldsymbol{A} \boldsymbol{Q}_t^{-2} \boldsymbol{Q}_s^2 (\mathcal{L}(\boldsymbol{X}_s) \mathfrak{M}_s + \mathcal{A}_s) \right) \mathrm{d}s.$$

We first show that if we fix any $t \leq T$, then for all $d$ sufficiently large with respect to $T$ and with probability at least $1 - 2e^{-(\log d)^{3/2}}$,

$$\max_{0 \leq u \leq t} \left\{ |\boldsymbol{M}_u^{(1,t)}| + |\boldsymbol{M}_u^{(2,t)}| \right\} \leq d^{-7\epsilon/8}.$$

We will then need to use a meshing argument to complete the argument. We show the details for the first. Those for the second are similar.

We simply need to bound the quadratic variation of each. Note

$$\langle \boldsymbol{M}_u^{(1,t)} \rangle = \int_0^u \gamma^2(s) \operatorname{tr} \left( \boldsymbol{Q}_t^{-1} \boldsymbol{Q}_s \nabla \mathcal{L}(\mathfrak{X}_{\Gamma(t)}^{\mathrm{gf}}) \nabla \mathcal{L}(\mathfrak{X}_{\Gamma(t)}^{\mathrm{gf}})^T \boldsymbol{Q}_t^{-1} \boldsymbol{Q}_s (\mathcal{L}(\boldsymbol{X}_s) \mathfrak{M}_s + \mathcal{A}_s) \right) \mathrm{d}u.$$

Here we use the norm boundedness of $\mathfrak{M}_t + \mathcal{A}_t$, by $2d^{-\epsilon}$. We further bound the other terms in norm to produce

$$\langle \boldsymbol{M}_u^{(1,t)} \rangle \leq 2d^{-\epsilon} \|\boldsymbol{A}^T \boldsymbol{A}\|_{\mathrm{op}}^2 \left( C^2 e^{Cu + (\log d)^{3/4}} \right) \mathcal{L}(\mathfrak{X}_{\Gamma(t)}^{\mathrm{gf}}).$$

We note that $\mathcal{L}(\mathfrak{X}_{\Gamma(t)}^{\mathrm{gf}}) \leq \mathcal{L}(\mathfrak{X}_0^{\mathrm{gf}})$. Hence with probability at least $1 - e^{-(\log d)^{3/2}}$ (for all $d$ sufficiently large with respect to $T, \|\boldsymbol{A}\|_{\mathrm{op}}, \|\boldsymbol{b}\|_2, \epsilon$),

$$\max_{0 \leq u \leq t} |\boldsymbol{M}_u^{(1,t)}| \leq d^{-7\epsilon/8}/2.$$

*Step 5 (Deferred). Mesh argument.* Finally, we use a union bound to gain the control from Step 4 over a mesh of $[0, T]$ of spacing $d^{-100}$. From the union bound, we therefore have for all these mesh points $\{t_k\}$

$$\max_k \max_{0 \leq u \leq t_k} \left\{ |\boldsymbol{M}_u^{(1,t_k)}| + |\boldsymbol{M}_u^{(2,t_k)}| \right\} \leq d^{-7\epsilon/8},$$

and this holds with probability $1 - 2Td^{100} e^{-(\log d)^{3/2}}$. For $t \in [t_k, t_{k+1}]$, we just use that

$$\|\nabla \mathcal{L}(\mathfrak{X}_{\Gamma(t)}^{\mathrm{gf}})^T \boldsymbol{Q}_t^{-1} - \nabla \mathcal{L}(\mathfrak{X}_{\Gamma(t_{k+1})}^{\mathrm{gf}})^T \boldsymbol{Q}_{t_{k+1}}^{-1}\| \leq C(T, \boldsymbol{A}, \boldsymbol{b}) d^{-100},$$

and thus on the event that $\mathcal{L}(\boldsymbol{X}_s)$ is bounded, we have for $t \in [t_k, t_{k+1}]$

$$|\boldsymbol{M}_t^{(1)} - \boldsymbol{M}_t^{(1,t_{k+1})}| \leq C(T, \boldsymbol{A}, \boldsymbol{b}) d^{-50}$$

for all $d$ sufficiently large with respect to $T, \|\boldsymbol{A}\|_{\mathrm{op}}$, and $\|\boldsymbol{b}\|_2$.

*Step 6. Other quadratics.* Hence, if we take $\Psi_t$ as a solution to the Volterra equation

$$\Psi_t = \mathcal{L}(\mathfrak{X}_{\Gamma(t)}^{\mathrm{gf}}) + \frac{1}{2} \int_0^t \gamma^2(s) \operatorname{tr} \left( \boldsymbol{A}^T \boldsymbol{A} \boldsymbol{Q}_t^{-2} \boldsymbol{Q}_s^2 \mathcal{A}_s \right) \mathrm{d}s + \frac{1}{2} \int_0^t \gamma^2(s) \operatorname{tr} \left( \boldsymbol{A}^T \boldsymbol{A} \boldsymbol{Q}_t^{-2} \boldsymbol{Q}_s^2 \mathfrak{M}_s \right) \Psi_s \, \mathrm{d}s,$$

then we have a high-quality approximation for the loss $\mathcal{L}(\boldsymbol{X}_t)$, and moreover, applying Itô's equation, we may always represent another quadratic $\mathcal{R} : \mathbb{R}^d \to \mathbb{R}$ of the SLD by (analogously to (40))

$$\mathcal{R}(\boldsymbol{X}_t) = \mathcal{R}\big(\mathcal{X}_{\Gamma(t)}^{\mathrm{gf}}\big) + \nabla\mathcal{R}(\mathcal{X}_{\Gamma(t)}^{\mathrm{gf}})^T \boldsymbol{Q}_t^{-1} \int_0^t \gamma(s) \boldsymbol{Q}_s \sqrt{\mathcal{L}(\boldsymbol{X}_s)\mathcal{M}_s + \mathcal{A}_s}\, \mathrm{d}\boldsymbol{B}_s$$

$$+ \frac{1}{2}\bigg(\boldsymbol{Q}_t^{-1}\int_0^t \gamma(s)\boldsymbol{Q}_s \sqrt{\mathcal{L}(\boldsymbol{X}_s)\mathcal{M}_s + \mathcal{A}_s}\,\mathrm{d}\boldsymbol{B}_s\bigg)^T (\nabla^2\mathcal{R}) \boldsymbol{Q}_t^{-1}\int_0^t \gamma(s)\boldsymbol{Q}_s \sqrt{\mathcal{L}(\boldsymbol{X}_s)\mathcal{M}_s + \mathcal{A}_s}\,\mathrm{d}\boldsymbol{B}_s$$

By comparing this to the same expression, where we replace the losses $\mathcal{L}(\boldsymbol{X}_s)$ by $\Psi_s$ and compute expectations over the Brownian terms, we arrive at (compare (41))

$$\mathcal{R}(\boldsymbol{X}_t) = \boldsymbol{M}_t^{(3)} + \mathcal{R}\big(\mathcal{X}_{\Gamma(t)}^{\mathrm{gf}}\big) + \frac{1}{2}\int_0^t \gamma^2(s)\,\mathrm{tr}\Big((\nabla^2\mathcal{R})\boldsymbol{Q}_t^{-2}\boldsymbol{Q}_s^2\mathcal{A}_s\Big)\,\mathrm{d}s$$

$$+ \frac{1}{2}\int_0^t \gamma^2(s)\,\mathrm{tr}\Big((\nabla^2\mathcal{R})\boldsymbol{Q}_t^{-2}\boldsymbol{Q}_s^2\mathcal{M}_s\Big)\Psi_s\,\mathrm{d}s.$$

Provided the Hessian $(\nabla^2\mathcal{R})$ and gradient $\nabla\mathcal{R}(0)$ are bounded independently of $d$ uniformly on $T$, the concentration of $\boldsymbol{M}_t^{(3)}$ now follows exactly as in Steps 4 and 5. $\qquad\square$

# E   Limiting values of the excess risk

In this section, we prove the limiting excess risk values, Theorem 3. We will, in fact, prove a more general version of Theorem 3 which holds for a wider class, so called the Volterra SLD class, as discussed in (37). Theorems 12 (constant learning rate) and 13 (time dependent learning rate) immediately imply Theorem 3 by setting $\mathcal{M}_t \equiv \frac{1}{n}\nabla^2\mathcal{L}$ and $\mathcal{A}_t \equiv 0$. By using the Volterra SLD class, we also recover the result for streaming, Theorem 10.

**Excess risk in the constant case.**    Under the stronger assumptions of constant learning rate, and constant variance profile, this can be further simplified. That is, suppose

**Assumption 11.** *Suppose that the covariance processes $\mathcal{M}$ and $\mathcal{A}$ are constant and satisfy for some absolute constants $c > 0$ and $\epsilon > 0$*

$$(\gamma(t), \mathcal{M}_t, \mathcal{A}_t) \equiv (\gamma, \mathcal{M}, \mathcal{A}), \quad where \quad (\mathrm{tr}\,\mathcal{M} + \mathrm{tr}\,\mathcal{A}) \le c < \infty \quad and \quad (\|\mathcal{M}\|_{op} + \|\mathcal{A}\|_{op}) < d^{-\epsilon}.$$

Under this assumption the kernels in the Volterra equation simplify to be:

$$K(t, s; \boldsymbol{P}) = K(t - s; \boldsymbol{P}) = \gamma^2\,\mathrm{tr}\Big(\mathcal{M}\boldsymbol{P}\exp\big(-2\gamma(\boldsymbol{A}^T\boldsymbol{A} + \delta\mathbf{I}_d)(t - s)\big)\Big),$$

$$A(t, s; \boldsymbol{P}) = A(t - s; \boldsymbol{P}) = \gamma^2\,\mathrm{tr}\Big(\mathcal{A}\boldsymbol{P}\exp\big(-2\gamma(\boldsymbol{A}^T\boldsymbol{A} + \delta\mathbf{I}_d)(t - s)\big)\Big) \tag{43}$$

The theory of convolution-type Volterra equations is substantially simpler than those of non-convolution type. In particular, we can completely rates of convergence and the limiting loss, as well as convergence guarantees (note that if the training loss of the underlying GF does not tend to 0 and or $\mathcal{A} \ne 0$, then the loss does not tend to 0, and so this is neighborhood convergence).

**Theorem 12** (Limit risk values, constant learning rate ). *Suppose the learning rate is constant, $\gamma(t) \equiv \gamma$. Under Assumption 11, the Volterra SLD is (neighborhood) convergent if and only if*

$$\mathcal{I}(\gamma) \stackrel{def}{=} \int_0^\infty K(t; \boldsymbol{A}^T\boldsymbol{A})\,\mathrm{d}t = \frac{\gamma}{2}\,\mathrm{tr}\Big(\mathcal{M}(\boldsymbol{A}^T\boldsymbol{A})\big(\boldsymbol{A}^T\boldsymbol{A} + \delta\mathbf{I}_d\big)^{-1}\Big) < 1. \tag{44}$$

*In the case that $\mathcal{I}(\gamma) < 1$, $\Psi_t$ converges as $t \to \infty$ to*

$$\Psi_\infty \stackrel{def}{=} (1 - \mathcal{I})^{-1}\bigg(\mathcal{L}(\mathcal{X}_\infty^{\mathrm{gf}}) + \frac{\gamma}{2}\,\mathrm{tr}\big(\mathcal{A}(\boldsymbol{A}^T\boldsymbol{A})\big(\boldsymbol{A}^T\boldsymbol{A} + \delta\mathbf{I}_d\big)^{-1}\big)\bigg). \tag{45}$$

*Likewise, the population risk $\Omega_t$ converges as $t \to \infty$ to*

$$\Omega_\infty \stackrel{def}{=} \mathcal{R}(\mathcal{X}_\infty^{\mathrm{gf}}) + \frac{\gamma}{2}\,\mathrm{tr}\big((\mathcal{A} + \mathcal{M}\Psi_\infty)(\nabla^2\mathcal{R})\big(\boldsymbol{A}^T\boldsymbol{A} + \delta\mathbf{I}_d\big)^{-1}\big). \tag{46}$$

*Proof of Theorem 12.* follows immediately from limiting values of renewal equations (see [22] and [5]). □

By setting $\mathcal{M} = \frac{1}{n}\nabla^2 \mathcal{L}$ and $\mathcal{A} \equiv 0$, we recover the multi-pass SGD setting discussed in the main portion of this paper. When the learning rate satisfies $\gamma(t) \equiv \gamma \in (0, 2(\frac{1}{n}\operatorname{tr}\{\frac{(\boldsymbol{A}^T\boldsymbol{A})^2}{\boldsymbol{A}^T\boldsymbol{A}+\delta\boldsymbol{I}_d}\})^{-1}$, it follows that $\mathcal{I}(\gamma) < 1$. Consequently, Theorem 12 proves Theorem 3 when the learning rate is constant.

**Excess risk when learning rate is time dependent.** In the case that the learning rate is time dependent, we prove the following result for the limiting dynamics under the expanded Volterra SLD class. Here we will still assume that the covariance processes $\mathcal{M}$ and $\mathcal{A}$ are constant.

**Assumption 12.** *Suppose that the covariance processes $\mathcal{M}$ and $\mathcal{A}$ are constant and satisfy for some absolute constants $c > 0$ and $\epsilon > 0$*

$$(\mathcal{M}_t, \mathcal{A}_t) \equiv (\mathcal{M}, \mathcal{A}), \quad where \quad \big(\operatorname{tr}\mathcal{M} + \operatorname{tr}\mathcal{A}\big) \le c < \infty \quad and \quad \big(\|\mathcal{M}\|_{op} + \|\mathcal{A}\|_{op}\big) < d^{-\epsilon}.$$

The time-dependent learning rate excess risk is given below.

**Theorem 13** (Time infinity risk values for SLD class). *Suppose Assumption 12 holds for the Volterra SLD class and the integrated learning rate satisfies $\Gamma(t) \to \infty$ and $\gamma(t) \to \gamma \in [0, \infty)$. Let the limiting learning rate value $\gamma$ be chosen such that the kernel norm is less than 1, that is,*

$$\mathcal{I}(\gamma) \stackrel{def}{=} \int_0^\infty K(t, s; \boldsymbol{A}^T\boldsymbol{A}) \, \mathrm{d}t = \frac{\gamma}{2}\operatorname{tr}\Big(\mathcal{M}(\boldsymbol{A}^T\boldsymbol{A})\big(\boldsymbol{A}^T\boldsymbol{A} + \delta\boldsymbol{I}_d\big)^{-1}\Big) < 1. \tag{47}$$

*Then with $\Psi_\infty$ given by the limiting empirical risk,*

$$\Psi_\infty = \Big(1 - \frac{\gamma}{2}\operatorname{tr}\Big\{\frac{\mathcal{M}\boldsymbol{A}^T\boldsymbol{A}}{\boldsymbol{A}^T\boldsymbol{A} + \delta\boldsymbol{I}_d}\Big\}\Big)^{-1} \times \Big(\mathcal{L}(\mathfrak{X}_\infty^{gf}) + \frac{\gamma}{2}\operatorname{tr}\Big\{\frac{\mathcal{A}\boldsymbol{A}^T\boldsymbol{A}}{\boldsymbol{A}^T\boldsymbol{A} + \delta\boldsymbol{I}_d}\Big\}\Big), \tag{48}$$

*the excess risk converges to*

$$\Omega_t - \mathcal{R}\big(\mathfrak{X}_{\Gamma(t)}^{gf}\big) \to \frac{\gamma}{2} \times \operatorname{tr}\Big\{\frac{(\mathcal{A} + \mathcal{M}\Psi_\infty)(\nabla^2\mathcal{R})}{\boldsymbol{A}^T\boldsymbol{A} + \delta\boldsymbol{I}_d}\Big\}.$$

*Proof.* First suppose that the limiting loss value of $\Psi_t$, defined in (38), is bounded and it exists at infinity. We show under this condition on $\Psi_t$ that the limiting risk value holds for $\Omega_\infty$, defined in (39). A simple computation with a change of variables gives

$$\begin{aligned}
&\lim_{t\to\infty} \Omega_t - \mathcal{R}(\mathfrak{X}_{\Gamma(t)}^{gf}) \\
&= \lim_{t\to\infty} \int_0^t \gamma^2(s)\operatorname{tr}\Big((\nabla^2\mathcal{R})\mathcal{A}\exp\big(-2(\boldsymbol{A}^T\boldsymbol{A} + \delta\boldsymbol{I}_d)(\Gamma(t) - \Gamma(s))\big)\Big)\,\mathrm{d}s \\
&\quad + \lim_{t\to\infty} \int_0^t \gamma^2(s)\operatorname{tr}\Big((\nabla^2\mathcal{R})\mathcal{M}\exp\big(-2(\boldsymbol{A}^T\boldsymbol{A} + \delta\boldsymbol{I}_d)(\Gamma(t) - \Gamma(s))\big)\Big)\Psi_s\,\mathrm{d}s \\
&= \lim_{t\to\infty} \int_0^{\Gamma(t)} \gamma(s)\operatorname{tr}\Big(\nabla^2\mathcal{R})\mathcal{A}\exp\big(-2(\boldsymbol{A}^T\boldsymbol{A} + \delta\boldsymbol{I}_d)(\Gamma(t) - s)\big)\Big)\,\mathrm{d}s \\
&\quad + \lim_{t\to\infty} \int_0^{\Gamma(t)} \gamma(s)\operatorname{tr}\Big((\nabla^2\mathcal{R})\mathcal{M}\exp\big(-2(\boldsymbol{A}^T\boldsymbol{A} + \delta\boldsymbol{I}_d)(\Gamma(t) - s)\big)\Big)\Psi_{\Gamma^{-1}(s)}\,\mathrm{d}s \\
&= \lim_{t\to\infty} \int_0^{\Gamma(t)} \gamma(\Gamma(t) - v)\operatorname{tr}\Big(\nabla^2\mathcal{R})\mathcal{A}\exp\big(-2(\boldsymbol{A}^T\boldsymbol{A} + \delta\boldsymbol{I}_d)v\big)\Big)\,\mathrm{d}s \\
&\quad + \lim_{t\to\infty} \int_0^{\Gamma(t)} \gamma(\Gamma(t) - v)\operatorname{tr}\Big((\nabla^2\mathcal{R})\mathcal{M}\exp\big(-2(\boldsymbol{A}^T\boldsymbol{A} + \delta\boldsymbol{I}_d)v\big)\Big)\Psi_{\Gamma^{-1}(\Gamma(t)-v)}\,\mathrm{d}v.
\end{aligned} \tag{49}$$

Dominated convergence theorem allows us to interchange the integral and limit as $\Psi_t$ and $\gamma(t)$ are bounded. We pull out the limiting values of $\lim_{t\to\infty}\gamma(t) = \gamma$ and $\Psi_\infty$. By integrating, we deduce

$$
\begin{aligned}
&\lim_{t\to\infty}\Omega_t - \mathcal{R}(\mathfrak{X}^{\mathrm{gf}}_{\Gamma(t)}) \\
&= \lim_{t\to\infty}\int_0^{\Gamma(t)}\gamma(\Gamma(t)-v)\operatorname{tr}\left(\nabla^2\mathcal{R})\mathcal{A}\exp\left(-2(\boldsymbol{A}^T\boldsymbol{A}+\delta\mathbf{I}_d)v\right)\right)\mathrm{d}s \\
&\quad + \lim_{t\to\infty}\int_0^{\Gamma(t)}\gamma(\Gamma(t)-v)\operatorname{tr}\left((\nabla^2\mathcal{R})\mathcal{M}\exp\left(-2(\boldsymbol{A}^T\boldsymbol{A}+\delta\mathbf{I}_d)v\right)\right)\Psi_{\Gamma^{-1}(\Gamma(t)-v)}\,\mathrm{d}v \\
&= \gamma\int_0^\infty\operatorname{tr}\left((\nabla^2\mathcal{R})\mathcal{A}\exp\left(-2(\boldsymbol{A}^T\boldsymbol{A}+\delta\mathbf{I}_d)v\right)\right)\mathrm{d}v \\
&\quad + \gamma\Psi_\infty\int_0^\infty\operatorname{tr}\left((\nabla^2\mathcal{R})\mathcal{M}\exp\left(-2(\boldsymbol{A}^T\boldsymbol{A}+\delta\mathbf{I}_d)v\right)\right)\mathrm{d}v \\
&= \gamma\operatorname{tr}\left((\nabla^2\mathcal{R})(\mathcal{M}\Psi_\infty+\mathcal{A})(2(\boldsymbol{A}^T\boldsymbol{A}+\delta\mathbf{I}_d))^{-1}\right).
\end{aligned}
\tag{50}
$$

The result for the limiting risk value $\lim_{t\to\infty}\Omega_t - \mathcal{R}(\mathfrak{X}^{\mathrm{gf}}_{\Gamma(t)})$ follows.

It remains to show that $\Psi_t$ is bounded and exists at infinity with its limiting value given by (48). Recall the loss kernel for $\Psi_t$ given by

$$
K(t,s) \stackrel{\text{def}}{=} K(t,s;\boldsymbol{A}^T\boldsymbol{A}) = \gamma^2(s)\operatorname{tr}\left(\mathcal{M}\boldsymbol{A}^T\boldsymbol{A}\exp\left(-2(\boldsymbol{A}^T\boldsymbol{A}+\delta\mathbf{I}_d)(\Gamma(t)-\Gamma(s))\right)\right), \quad (51)
$$

so that $\Psi_t$ is the solution to the Volterra equation

$$
\Psi_t = \mathcal{L}(\mathfrak{X}^{\mathrm{gf}}_{\Gamma(t)}) + \int_0^t K(t,s)\Psi_s\,\mathrm{d}s. \tag{52}
$$

Under the kernel norm bounded by 1, (47), we show that the kernel $K(s,t)$ is of $L^\infty$-type on $[0,\infty)$. A kernel is $L^\infty$-type if $\|K\|_{L^\infty(J)} < \infty$ for a set $J \subset \mathbb{R}$ where $\|K\|_{L^\infty(J)} = \sup_{t\in J}\int_J |K(s,t)|\,\mathrm{d}s$ [22, Chapter 9.2]. For this, we see that for each $t$ and $s$

$$
K(t,s) \le \widehat{\gamma}\cdot\gamma(s)\operatorname{tr}\left(\mathcal{M}\boldsymbol{A}^T\boldsymbol{A}\exp\left(-2(\boldsymbol{A}^T\boldsymbol{A}+\delta\mathbf{I}_d)(\Gamma(t)-\Gamma(s))\right)\right). \tag{53}
$$

This implies by change of variables that

$$
\begin{aligned}
\int_0^t K(t,s)\,\mathrm{d}s &\le \int_0^{\Gamma(t)}\widehat{\gamma}\operatorname{tr}\left(\mathcal{M}\boldsymbol{A}^T\boldsymbol{A}\exp\left(-2(\boldsymbol{A}^T\boldsymbol{A}+\delta\mathbf{I}_d)(\Gamma(t)-s)\right)\right)\mathrm{d}s \\
&\le \frac{\widehat{\gamma}}{2}\operatorname{tr}\left(\mathcal{M}\boldsymbol{A}^T\boldsymbol{A}(\boldsymbol{A}^T\boldsymbol{A}+\delta\mathbf{I}_d)^{-1}\right) < \infty.
\end{aligned}
\tag{54}
$$

Hence, it follows that the kernel $K$ is $L^\infty$-type on $[0,\infty)$. To prove the boundedness assumption of $\Psi_t$, we will need something slightly stronger. We show that there exists a finite number of intervals $J_i$ such that $\cup_i J_i = [0,\infty)$ and $\|K\|_{L^\infty(J_i)} \le 1$. From this and Theorem 9.3.13 in [22], it will follow that the resolvent is also of type $L^\infty$ on $[0,\infty)$. Since $\gamma(t)\to\gamma$, there exists a $t_0$ such that for all $t \ge t_0$, $\gamma(t) \le \gamma + \varepsilon$. This $\varepsilon > 0$ can be chosen sufficiently small such that $\gamma + \varepsilon < 2\left(\operatorname{tr}(\mathcal{M}\boldsymbol{A}^T\boldsymbol{A}(\boldsymbol{A}^T\boldsymbol{A}+\delta\mathbf{I}_d)^{-1})\right)^{-1}$ (see (47) which gives an upper bound on $\gamma$). First, we observe that

$$
\sup_{t\ge 0}\sup_{0\le s\le t_0}K(t,s) \le \widehat{\gamma}^2\operatorname{tr}\left(\mathcal{M}(\boldsymbol{A}^T\boldsymbol{A})e^{2(\boldsymbol{A}^T\boldsymbol{A}+\delta\mathbf{I}_d)\Gamma(t_0)}\right) < \infty.
$$

We break up the interval $[0,t_0]$ into finitely many intervals of length each of which has a length strictly less than $\left(\widehat{\gamma}^2\operatorname{tr}\left(\mathcal{M}\boldsymbol{A}^T\boldsymbol{A}e^{2(\boldsymbol{A}^T\boldsymbol{A}+\delta\mathbf{I}_d)\Gamma(t_0)}\right)\right)^{-1}$. If we denote these intervals by $J_i$, then it immediately follows by bounding the integral using the sup of $K$ multiplied by the length of the interval $J_i$ that

$$
\|K\|_{L^\infty(J_i)} = \sup_{t\in J_i}\int_{J_i}K(t,s)\,\mathrm{d}s < 1.
$$

It only remains to show on the tail, that is, $J_\infty \overset{\text{def}}{=} (t_0, \infty)$, for which $\||K|\|_{L^\infty(J_\infty)} < 1$. Using the same change of variables as in (53) and our choice of $t_0$, we have that for all $t \geq t_0$

$$\int_{t_0}^t K(t, s) \, \mathrm{d}s \leq \int_{t_0}^t (\gamma + \varepsilon)\gamma(s) \operatorname{tr}\left(\mathcal{M}\boldsymbol{A}^T\boldsymbol{A}\exp\left(-2(\boldsymbol{A}^T\boldsymbol{A} + \delta\mathbf{I}_d)(\Gamma(t) - \Gamma(s)))\right)\right) \mathrm{d}s$$

$$= \int_{\Gamma(t_0)}^{\Gamma(t)} (\gamma + \varepsilon)\operatorname{tr}\left(\mathcal{M}\boldsymbol{A}^T\boldsymbol{A}\exp\left(-2(\boldsymbol{A}^T\boldsymbol{A} + \delta\mathbf{I}_d)(\Gamma(t) - s))\right)\right)\mathrm{d}s$$

$$\leq \frac{\gamma + \varepsilon}{2} \operatorname{tr}\left(\mathcal{M}\boldsymbol{A}^T\boldsymbol{A}(\boldsymbol{A}^T\boldsymbol{A} + \delta\mathbf{I}_d)^{-1}\right) < 1.$$

The last inequality following by our assumption on $\gamma + \varepsilon$ being sufficiently small. By Theorem 9.3.13 in [22], we have that the resolvent is also of type $L^\infty$ on $[0, \infty)$. We also have that $K(t, s)$ is of bounded type, that is the kernel is bounded (see [22, Definition 9.5.2] for precise definition). Since the forcing term $\mathcal{L}(\mathcal{X}_{\Gamma(t)}^{\text{gf}})$ and $\int_0^t A(t, s; \boldsymbol{A}^T\boldsymbol{A}) \, \mathrm{d}s$ are bounded, then it follows by [22, Theorem 9.5.4] that the solution to the Volterra equation (52), $\Psi_t$, is bounded.

We now show that $\Psi_t$ exists at infinity. Fix a $\varepsilon > 0$. By the assumptions on the learning rate, there exists a $t_0 > 0$ such that for all sufficiently large $t \geq s \geq t_0$

$$\gamma - \varepsilon \leq \gamma(t) \leq \gamma + \varepsilon \quad \text{and} \quad (\gamma - \varepsilon)(t - s) \leq \Gamma(t) - \Gamma(s) \leq (\gamma + \varepsilon)(t - s). \tag{55}$$

Using these inequalities for $\gamma(t)$, we get an upper bound and lower bound on the kernel $K(t, s)$ which we denote by $\overline{K}(t, s)$ and $\underline{K}(t, s)$, respectively. Specifically for all $t, s \geq t_0$,

$$K(t, s) \leq \overline{K}(t, s) \overset{\text{def}}{=} (\gamma + \varepsilon)^2 \operatorname{tr}\left(\mathcal{M}\boldsymbol{A}^T\boldsymbol{A}\exp\left(-2(\boldsymbol{A}^T\boldsymbol{A} + \delta\mathbf{I}_d)(\gamma - \varepsilon)(t - s)\right)\right)$$

$$K(t, s) \geq \underline{K}(t, s) \overset{\text{def}}{=} (\gamma - \varepsilon)^2 \operatorname{tr}\left(\mathcal{M}\boldsymbol{A}^T\boldsymbol{A}\exp\left(-2(\boldsymbol{A}^T\boldsymbol{A} + \delta\mathbf{I}_d)(\gamma + \varepsilon)(t - s)\right)\right). \tag{56}$$

The kernels $\overline{K}(t, s)$ and $\underline{K}(t, s)$ are substantially nicer than the original $K(t, s)$ because they are proper convolution kernels. Here one can define $\overline{K} : [0, \infty) \to \mathbb{R}$ by

$$\overline{K}(t) \overset{\text{def}}{=} (\gamma + \varepsilon)^2 \operatorname{tr}\left(\mathcal{M}\boldsymbol{A}^T\boldsymbol{A}\exp\left(-2(\boldsymbol{A}^T\boldsymbol{A} + \delta\mathbf{I}_d)(\gamma - \varepsilon)t\right)\right).$$

Then it follows that $\overline{K}(t, s) = \overline{K}(t - s)$. A similar result holds for $\underline{K}(t, s)$.

For ease of notation, define the forcing function: for $t \geq t_0$

$$F(t) \overset{\text{def}}{=} \mathcal{L}(\mathcal{X}_{\Gamma(t)}^{\text{gf}}) + \int_0^{t_0} K(t, s)\Psi_s \, \mathrm{d}s + \int_0^t A(t, s; \boldsymbol{A}^T\boldsymbol{A}) \, \mathrm{d}s, \tag{57}$$

where $\Psi_t$ is a solution to (52). Similar to the definitions of $\overline{K}(t)$ and $\underline{K}(t)$, we define $\overline{F}(t)$ and $\underline{F}(t)$ respectively as

$$\underline{F}(t) \leq F(t) \leq \overline{F}(t), \tag{58}$$

$$\text{where} \quad \overline{F}(t) \overset{\text{def}}{=} \mathcal{L}(\mathcal{X}_{\Gamma(t)}^{\text{gf}}) + \int_0^{t_0} K(t, s)\Psi_s \, \mathrm{d}s + \int_0^{t_0} A(t, s; \boldsymbol{A}^T\boldsymbol{A}) \, \mathrm{d}s$$

$$+ \int_0^t (\gamma + \varepsilon)^2 \operatorname{tr}\left(\mathcal{A}\boldsymbol{A}^T\boldsymbol{A}\exp\left(-2(\boldsymbol{A}^T\boldsymbol{A} + \delta\mathbf{I}_d)(\gamma - \varepsilon)(t - s)\right)\right)$$

$$\text{and} \quad \underline{F}(t) \overset{\text{def}}{=} \mathcal{L}(\mathcal{X}_{\Gamma(t)}^{\text{gf}}) + \int_0^{t_0} K(t, s)\Psi_s \, \mathrm{d}s + \int_0^{t_0} A(t, s; \boldsymbol{A}^T\boldsymbol{A}) \, \mathrm{d}s$$

$$+ \int_0^t (\gamma - \varepsilon)^2 \operatorname{tr}\left(\mathcal{A}\boldsymbol{A}^T\boldsymbol{A}\exp\left(-2(\boldsymbol{A}^T\boldsymbol{A} + \delta\mathbf{I}_d)(\gamma + \varepsilon)(t - s)\right)\right) \mathrm{d}s \tag{59}$$

Because $\Psi_s$ is bounded, it follows that $\lim_{t\to\infty} \int_0^{t_0} K(t, s)\Psi_s = \lim_{t\to\infty} A(t, s; \boldsymbol{A}^T\boldsymbol{A}) = 0$. Also it is clear that the $F(t), \overline{F}(t)$, and $\underline{F}(t)$ are bounded.

Using the upper/lower bound on the kernel (56), we can squeeze the value of $\Psi_t$ between two expressions: for $t, s \geq t_0$,

$$\underline{F}(t) + \int_{t_0}^t \underline{K}(t, s) \Psi_s \, \mathrm{d}s \leq \Psi_t \leq \overline{F}(t) + \int_{t_0}^t \overline{K}(t, s) \Psi_s \, \mathrm{d}s. \tag{60}$$

Using a similar argument for $K(t, s)$ and choosing $\varepsilon$ sufficiently small, $\overline{K}(t, s)$ and $\underline{K}(t, s)$ are $L^\infty$-type on $[0, \infty)$. Moreover using a similar argument as we did for $K$ itself, the norms $\left\| \overline{K} \right\|_{L^\infty([0,\infty))} < 1$ and $\left\| \underline{K} \right\|_{L^\infty([0,\infty))} < 1$. Here we used the upper bound on $\gamma$ in (47) and a sufficiently small $\varepsilon$. Note we do not need to break up into finite intervals. As before, the resolvent then is of $L^\infty$-type on $[0, \infty)$ [22, Corollary 9.3.10]. Further because of non-negativity, Proposition 9.8.1 in [22] yields that the resolvents are also non-negative.

Consider the upper bound (a similar argument will hold for the lower bound). We can apply Gronwall's inequality (60) [22, Theorem 9.8.2]. It follows that $\Psi_t$ is upper bounded (lower bounded) by the solutions $\overline{\Psi}_t$ ($\underline{\Psi}_t$) to the following convolution Volterra equations

$$\overline{\Psi}_t = \overline{F}(t) + \int_{t_0}^t \overline{K}(t, s) \overline{\Psi}_s \, \mathrm{d}s \quad \text{and} \quad \underline{\Psi}_t = \underline{F}(t) + \int_{t_0}^t \underline{K}(t, s) \underline{\Psi}_s \, \mathrm{d}s.$$

Specifically, we have $\underline{\Psi}_t \leq \Psi_t \leq \overline{\Psi}_t$ for all $t \geq t_0$. Since $\overline{\Psi}_t$ and $\underline{\Psi}_t$ are solutions to a proper convolution-type Volterra equation and both functions $\overline{F}(t)$, and $\underline{F}(t)$ have limits at infinity ($\overline{F}(\infty) \stackrel{\text{def}}{=} \lim_{t \to \infty} \overline{F}(t)$ and $\underline{F}(\infty) \stackrel{\text{def}}{=} \lim_{t \to \infty} \underline{F}(t)$), by [5], for $t \geq t_0$

$$\limsup_{t \to \infty} \Psi_t \leq \limsup_{t \to \infty} \overline{\Psi}_t = \overline{F}(\infty) \left( 1 - \left\| \overline{K} \right\|_{L^\infty([t_0,\infty))} \right)^{-1} \leq \overline{F}(\infty) \left( 1 - \left\| \overline{K} \right\|_{L^\infty([0,\infty))} \right)^{-1}, \tag{61}$$

and similarly, the lower bound gives

$$\liminf_{t \to \infty} \Psi_t \geq \liminf_{t \to \infty} \underline{\Psi}_t \leq \underline{F}(\infty) \left( 1 - \left\| \underline{K} \right\|_{L^\infty([0,\infty))} \right)^{-1}. \tag{62}$$

A simple computation yields that

$$\left\| \overline{K} \right\|_{L^\infty([0,\infty))} = \frac{(\gamma + \varepsilon)^2}{\gamma - \varepsilon} G(\mathcal{M}) \quad \text{and} \quad \left\| \underline{K} \right\|_{L^\infty([0,\infty))} = \frac{(\gamma - \varepsilon)^2}{\gamma + \varepsilon} G(\mathcal{M})$$

$$\text{and} \quad \overline{F}(\infty) = \mathcal{L}(\mathcal{X}_\infty^{\mathrm{gf}}) + \frac{(\gamma + \varepsilon)^2}{\gamma - \varepsilon} G(\mathcal{A}) \quad \text{and} \quad \underline{F}(\infty) = \mathcal{L}(\mathcal{X}_\infty^{\mathrm{gf}}) + \frac{(\gamma - \varepsilon)^2}{\gamma + \varepsilon} G(\mathcal{A}) \tag{63}$$

$$\text{where} \quad G(\mathcal{H}) \stackrel{\text{def}}{=} \tfrac{1}{2} \operatorname{tr} \left( \mathcal{H} \boldsymbol{A}^T \boldsymbol{A} (\boldsymbol{A}^T \boldsymbol{A} + \delta \mathbf{I}_d)^{-1} \right).$$

So for any sufficiently small $\varepsilon > 0$, we have that

$$\left( 1 - \frac{(\gamma - \varepsilon)^2}{\gamma + \varepsilon} G(\mathcal{M}) \right)^{-1} \times \left\{ \mathcal{L}(\mathcal{X}_\infty^{\mathrm{gf}}) + \frac{(\gamma - \varepsilon)^2}{\gamma + \varepsilon} G(\mathcal{A}) \right\} \leq \liminf_{t \to \infty} \Psi_t$$

$$\leq \limsup_{t \to \infty} \Psi_t \leq \left( 1 - \frac{(\gamma + \varepsilon)^2}{\gamma - \varepsilon} G(\mathcal{M}) \right)^{-1} \times \left\{ \mathcal{L}(\mathcal{X}_\infty^{\mathrm{gf}}) + \frac{(\gamma + \varepsilon)^2}{\gamma - \varepsilon} G(\mathcal{A}) \right\}. \tag{64}$$

As this holds for any sufficiently small $\varepsilon$, the result follows by sending $\varepsilon \to 0$. $\qquad \square$

# F  Algorithmic regularization

In this section, we discuss the exact asymptotic convergence rates for SGD and full batch momentum algorithms on high-dimensional $\ell^2$-regularized least squares problems. The results in this section (e.g., Theorems 4 and 5) were shown in a series of papers [58–60] that explored exact trajectories of loss function.

## F.1  Convergence rates of SGD

To characterize the rates, we define $\lambda_{\min}$ as the smallest non-zero eigenvalue of $\boldsymbol{A}^T \boldsymbol{A}$. Then for generic initial conditions, (in particular almost surely if $\boldsymbol{X}_0$ is isotropic norm 1), then

$$\lim_{t \to \infty} \left( \mathcal{L}(\mathcal{X}_t^{\mathrm{gf}}) - \mathcal{L}(\mathcal{X}_\infty^{\mathrm{gf}}) \right)^{1/t} = \begin{cases} e^{-\gamma(\lambda_{\min} + \delta)}, & \text{if } \delta > 0, \\ e^{-2\gamma \lambda_{\min}}, & \text{otherwise.} \end{cases}$$

The rate of convergence of $\Psi_t$ to $\Psi_\infty$ is given by, (for small $\gamma$), the rate above. For larger $\gamma$, another rate can frustrate the convergence. Recall the *Malthusian exponent* of the convolution Volterra equation in (11) is given by

$$\lambda_* = \inf\left\{ x : 1 = \int_0^\infty e^{xt} K(t; \boldsymbol{A}^T \boldsymbol{A}) \, \mathrm{d}t = \gamma^2 \int_0^\infty e^{xt} \operatorname{tr}\left( \mathcal{M} \boldsymbol{A}^T \boldsymbol{A} \exp\left(-2\gamma(\boldsymbol{A}^T \boldsymbol{A} + \delta \mathbf{I})t\right) \right) \, \mathrm{d}t \right\}. \quad (65)$$

The set may be empty, in which case the infimum is $\infty$. We recall below Theorem 4.

**Theorem 14.** *For $\gamma > 0$ satisfying $\mathcal{I}(\gamma) < 1$ (see (44)), define*

$$\Xi(\gamma) \stackrel{def}{=} \begin{cases} \min\{\gamma(\lambda_{\min} + \delta), \lambda_*(\gamma)\} & \text{if } \delta > 0, \\ \min\{2\gamma\lambda_{\min}, \lambda_*(\gamma)\} & \text{if } \delta = 0. \end{cases} \quad (66)$$

*Then the rates of convergence of both the training and test loss are*

$$\lim_{t\to\infty} \left( \Psi_t - \Psi_\infty \right)^{1/t} = e^{-\Xi(\gamma)} = \lim_{t\to\infty} \left( \Omega_t - \Omega_\infty \right)^{1/t}$$

*Furthermore, when $\gamma = n/\operatorname{tr}(\boldsymbol{A}^T \boldsymbol{A})$, we have the rate guarantee $\Xi(\gamma) \geq \frac{\lambda_{\min} n}{2 \operatorname{tr}(\boldsymbol{A}^T \boldsymbol{A})}$.*

*Proof.* See [59, Theorem 1.2] for proof. $\qquad\square$

### F.2 Momentum GD (MGD) rates

In this section, we consider a popular *deterministic* or *full-batch* algorithm for solving the ridge regression problem in (1), that is, gradient descent with momentum (a.k.a Polyak momentum). Throughout this section, we use the notation, $\boldsymbol{x}_t^{\text{m-gd}} = \boldsymbol{x}_t$. Gradient descent with momentum (MGD), initialized at $\boldsymbol{x}_0 \in \mathbb{R}^d$ and $\boldsymbol{x}_1 = \boldsymbol{x}_0 - \frac{\gamma}{1+m} \nabla f(\boldsymbol{x}_0)$, iterates for $k \geq 1$

$$\boldsymbol{x}_{k+1} = \boldsymbol{x}_k + m(\boldsymbol{x}_k - \boldsymbol{x}_{k-1}) - \gamma \nabla f(\boldsymbol{x}_k), \quad (67)$$

where $\gamma, m > 0$ are the stepsize and momentum parameters respectively. From Proposition 3.1 in [60], there exists $k$-degree polynomials $P_k$ and $Q_k$ such that the iterates of GD+M satisfy the following

$$\boldsymbol{x}_k = P_k(\boldsymbol{A}^T \boldsymbol{A} + \delta \mathbf{I})\boldsymbol{x}_0 + Q_k(\boldsymbol{A}^T \boldsymbol{A} + \delta \mathbf{I})\boldsymbol{A}^T \boldsymbol{b}, \quad \text{with } P_k(\lambda) = 1 - (\lambda)Q_k(\lambda) \quad (68)$$

and the coefficients of $P_k$ and $Q_k$ only depend on the largest and smallest eigenvalue of $\boldsymbol{A}^T \boldsymbol{A}$. For Polyak, similar to the work in [60, Section 3.1], we can give an explicit representation for these polynomials $P_k$ and $Q_k$.

**Proposition 1** (Polynomial representation of MGD). *Suppose $\boldsymbol{x}_0 \in \mathbb{R}^d$ and fix a stepsize $\gamma > 0$ and momentum parameter $m > 0$. For the iterates of GD+M on (1) with ridge parameter $\delta > 0$, we have the following representation for the polynomials*

$$\boldsymbol{x}_k = P_k(\boldsymbol{A}^T \boldsymbol{A} + \delta \mathbf{I})\boldsymbol{x}_0 + Q_k(\boldsymbol{A}^T \boldsymbol{A} + \delta \mathbf{I})\boldsymbol{A}^T \boldsymbol{b}, \quad (69)$$

*where $P_k$ and $Q_k$ are $k$-degree polynomials satisfying*

$$P_k(\lambda) = m^{k/2} \left( \frac{2m}{1+m} T_k(\sigma(\lambda)) + \left(1 - \frac{2m}{1+m}\right) U_k(\sigma(\lambda)) \right) \quad \text{and} \quad Q_k(\lambda) = \frac{1 - P_k(\lambda)}{\lambda}$$

$$\text{where} \quad \sigma(\lambda) = \frac{1 + m - \gamma\lambda}{2\sqrt{m}}$$

*and $T_k$, $U_k$ are Chebyshev polynomials of the 1st and 2nd kind respectively.*

$$(70)$$

*Proof.* The proof can be found in [60, Appendix A.2] or [19, Chapter 11]. We include a sketch of the proof. From the recurrence in (67) and the gradient of the ridge regression, the polynomials $P_k$ that generate GD+M satisfy the following three-term recurrence,

$$P_{k+1}(\lambda) = (1 + m - \gamma\lambda)P_k(\lambda) - mP_{k-1}(\lambda)$$

$$P_k(\lambda) = \frac{P_{k+1}(\lambda) + mP_k(\lambda)}{1 + m - \gamma\lambda}. \quad (71)$$

We define the polynomial generating function for $P_k$ as $\mathcal{G}(\lambda, t) = \sum_{k=0}^{\infty} t^k P_k(\lambda)$. Using the recurrence in (71), we get that

$$\mathcal{G}(\lambda, t) = 1 + \frac{1}{t(1 - m + \gamma\lambda)} \sum_{k=2}^{\infty} t^k P_k(\lambda) - \frac{mt}{1 - m + \gamma\lambda} \sum_{k=0}^{\infty} t^k P_k(\lambda)$$

$$= 1 + \frac{1}{t(1 - m + \gamma\lambda)} \left[ \mathcal{G}(\lambda, t) - 1 - t(1 - \tfrac{\gamma}{1+m}\lambda) \right] - \frac{mt}{1 - m + \gamma\lambda} \mathcal{G}(\lambda, t).$$

By solving this expression for the generating polynomial, we have

$$\mathcal{G}(\lambda, t) = \frac{1 + t(m - (\gamma + \frac{\gamma}{1+m}))}{1 - t(1 - m + \gamma\lambda) - mt^2}.$$

This generating function for MGD closely resembles the generating function for Chebyshev polynomials of the 1st and 2nd kind. Under simple transformations (e.g., $t \mapsto \frac{t}{\sqrt{m}}$), this is exactly the case. These transformations yield the expression in (70). $\qquad\square$

The role of $\sigma(\lambda)$ is to transform the eigenvalues of $\boldsymbol{A}^T\boldsymbol{A} + \delta\mathbf{I}_d$ within a specific range controlled by the learning rate and momentum. It is known that the Chebyshev polynomials are well-behaved on the interval of $[-1, 1]$ and grow exponentially off of this region.

Moreover for a generic quadratic applied to $\boldsymbol{x}_k$, the rate of convergence will be controlled by $P_k(\lambda)$. Using standard asymptotic behavior of Chebyshev polynomials, we can derive asymptotic rates based on $\lambda_{\min} \overset{\text{def}}{=} \lambda_{\min}(\boldsymbol{A}^T\boldsymbol{A} + \delta\mathbf{I}_d)$ and $\lambda_{\max} \overset{\text{def}}{=} \lambda_{\max}(\boldsymbol{A}^T\boldsymbol{A} + \delta\mathbf{I}_d)$, the smallest (non-zero) and largest eigenvalues of $\boldsymbol{A}^T\boldsymbol{A}$ respectively. We record this result below

**Proposition 2** (Asymptotic rates of MGD). *The asymptotic rate of MGD is*

$$\limsup_{k \to \infty} \sqrt[k]{P_k} = \begin{cases} \sqrt{m} & \text{if } \gamma \in \left[ \frac{(1-\sqrt{m})^2}{\lambda_{\min}}, \frac{(1+\sqrt{m})^2}{\lambda_{\max}} \right] \\ \sqrt{m}(|\sigma(\lambda_{\min})| + \sqrt{\sigma(\lambda_{\min})^2 - 1}) & \text{if } \gamma \in \left[ 0, \min\left\{ \frac{2(1+m)}{\lambda_{\min}+\lambda_{\max}}, \frac{(1-\sqrt{m})^2}{\lambda_{\min}} \right\} \right] \\ \sqrt{m}(|\sigma(\lambda_{\max})| + \sqrt{\sigma(\lambda_{\max})^2 - 1}) & \text{if } \gamma \in \left[ \max\left\{ \frac{2(1+m)}{\lambda_{\min}+\lambda_{\max}}, \frac{(1+\sqrt{m})^2}{\lambda_{\max}} \right\}, \frac{2(1+m)}{\lambda_{\max}} \right] \\ \geq 1 & \text{otherwise.} \end{cases}$$

$$(72)$$

*Proof.* See [62] for a complete proof. The result follows from knowing that the iterates are given by Chebyshev polynomials and then applying well-known asymptotics of Chebyshev polynomials to get the convergence rate. $\qquad\square$

We can minimize over the rate to find the optimal parameters. In this case, they become the parameters used in the Heavy-Ball algorithm [65] where

$$m = \left( \frac{\sqrt{\lambda_{\max}} - \sqrt{\lambda_{\min}}}{\sqrt{\lambda_{\max}} + \sqrt{\lambda_{\min}}} \right)^2 \quad \text{and} \quad \gamma = \left( \frac{2}{\sqrt{\lambda_{\max}} + \sqrt{\lambda_{\min}}} \right)^2. \tag{73}$$

A simple computation yields that the asymptotic rate for Heavy-Ball is $\frac{\sqrt{\lambda_{\max}} - \sqrt{\lambda_{\min}}}{\sqrt{\lambda_{\max}} + \sqrt{\lambda_{\min}}}$.

# G   Numerical simulations

To illustrate our theoretical results and conjectures we report simulations and experiments using SGD with constant learning rate on the $\ell^2$-regularized least squares problem. In all simulations for the random $\ell^2$-regularized least-square problem, the vectors $\boldsymbol{\eta}$, and $\boldsymbol{\beta}$ are sampled from a standard Gaussian and the initialization vector $\boldsymbol{x}_0 = \boldsymbol{0}$ (for Figures 1 and 5) and $N(0, 4\mathbf{I}_d)$ (Figure 6). For the random features model (see Section B.1 and Figure 1, a standardized ReLu activation function was applied, that is

$$\sigma(\cdot) = \frac{\max\{\cdot, 0\} - 0.5(\pi)^{-1}}{0.5 - 0.5\pi^{-1}}. \tag{74}$$

The entries of the hidden weight matrix $\boldsymbol{W} \in \mathbb{R}^{n_0 \times d}$ in the random feature model are standard normal.

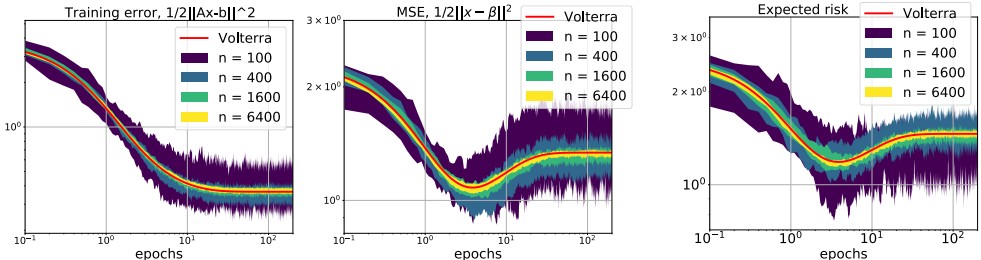

Figure 6: **Concentration of mean squared error (MSE) and expected test loss, $\frac{1}{2}\|x - \beta\|_2^2$, for SGD** on a Gaussian random $\ell^2$-regularized least-squares problem (Section B) where $\boldsymbol{\beta} \sim N(\mathbf{0}, \mathbf{I}_d)$ is the ground truth signal and a generative model $\boldsymbol{b} = \boldsymbol{A}\boldsymbol{\beta} + \boldsymbol{\eta}$ where entries of $\boldsymbol{\eta}$ iid standard normal with $\|\boldsymbol{\eta}\|_2^2 = 2.25$, $n = 0.9d$ with $\ell^2$-regularization parameter $\delta = 0.1$. SGD with constant learning rate $\gamma = 0.8$ was initialized at $\boldsymbol{x}_0 \sim N(\mathbf{0}, 4\mathbf{I}_d)$ (independent of $\boldsymbol{A}$, $\boldsymbol{\beta}$); an $80\%$ confidence interval (shaded region) over 10 runs for each $n$. Any quadratic statistic, such as the MSE, becomes non-random in the large limit and all runs of SGD converge to a deterministic function $\Omega_t$ (red) solving a Volterra equation (9). This is an illustration of Theorem 1 and Theorem 2.

**Volterra: computing theoretical dynamics.** When the entries of $\boldsymbol{A}$ are generated by standard Gaussians, a celebrated work [50] gives an explicit limiting density for the eigenvalues when $d$ and $n$ are proportional. In this case, the Volterra equation (8) for the loss function $\mathcal{L}$ is computable without needing to input the empirical eigenvalues of the data matrix $\boldsymbol{A}\boldsymbol{A}^T$. Since the covariance of standard Gaussians is explicitly $\boldsymbol{\Sigma}_f = \mathbf{I}_d$ (see Appendix B.1), one can also directly solve for the expected risk (9) for applications such as in-distribution expected risk. As such, the Volterra equation is completely determined. To solve it, a Chebyshev quadrature was used to derive a numerical approximation for the kernel, $K$, (8). The size of the grid points used to compute the numerical integration does effect the Volterra equations convergence to the theoretical limit. We suggest that the number of epochs be equal to the number of grid points used in the numerical quadrature rule. Next, to generate the solution $\mathcal{L}$ of the Volterra equation, we implement a Picard iteration which finds a fix point to the Volterra equation by repeatedly convolving the kernel and adding the forcing term. Despite the numerical approximations to integrals, the resulting solutions to the Volterra equation ($\Psi$ and $\Omega$) model the true behavior of SGD remarkably well. Similarly, by evaluating contour integrals, random features with Gaussian $\boldsymbol{X}$ and $\boldsymbol{W}$ known explicit formulas for the limiting densities of eigenvalues and eigenvectors (see e.g., [2]). This approach was used to compute the theoretical dynamics in Figure 6.

When the limiting eigenvalues and eigenvectors are unavailable, as in the case of real data sets, an empirical Volterra equation solver was used. We computed the svd of the data matrix $\boldsymbol{A}$ and calculated an empirical covariance for $\boldsymbol{\Sigma}_f$ (see Appendix B.1). The singular values and vectors of $\boldsymbol{A}$ and $\boldsymbol{\Sigma}_f$ were then used to compute the forcing term (i.e., the GF terms $\mathcal{L}(\mathcal{X}^{\text{gf}})$ and $\mathcal{R}(\mathcal{X}^{\text{gf}})$) and kernel $K$ (10). As before, a Chebyshev quadrature was used to derive the integral for the kernel $K$ and a Picard iteration to find the fix point of the Volterra was applied. This method was used to compute the theoretical dynamics $\Omega_t$ and $\Psi_t$ in Figures 1 and 5.

**Real data.** The CIFAR-5m [54] example (Figures 1 is shown to demonstrate that large-dimensional random matrix predictions often work for large dimensional real data. Random features models were used to predict the car/plane class vector which has approximately 1 million samples. The data sets were all standardized and pre-processed to have mean 0 and variance 1 before applying the random features model with standardized ReLu.

We give specific simulation/experimental details below:

- *CIFAR-5m streaming, Figure 1*: Plots of single runs of SGD on CIFAR-5m [54] using the car/plane class vector (samples $n = 1$ million, features $n_0 = 32 \times 32 \times 3$) on a random features model with standardized ReLu (see (74)). CIFAR-5m car/plane data set was standardized so that entries were mean 0 and variance 1. Standard Gaussian weight matrix $\boldsymbol{W} \in \mathbb{R}^{n_0 \times d}$ with fixed $d = 6,000$ used in the random features set-up (see Appendix B.1). Multi-pass SGD with constant learning rate $\gamma = 0.8$ applied to various sample size $n = 1000 \cdot [4, 6, 10, 20, 40]$

Table 1: **Summary of the eigenvalues in ICR** with normalized trace equal to 1.0, i.e., $\frac{1}{n}\operatorname{tr}(\boldsymbol{AA}^T) = 1.0$. All data sets were standardized before applying any transformations (e.g., random features). For random features (RF), standard Gaussian $\boldsymbol{W} \in \mathbb{R}^{n_0 \times d}$ applied to the data set followed by entry-wise application of standardized ReLu (see (74) and Appendix B.1 and Appendix G for exact set-up).

| | | | Eigenvalues of $\boldsymbol{AA}^T$ | |
| --- | --- | --- | --- | --- |
| **Data set** | **Samples** $(n)$ | **Features** $(d)$ | *Largest* | *Smallest* |
| CIFAR-10[1] (all) | 50,000 | 3,072 | 11,118.80 | $4.7 \cdot 10^{-4}$ |
| CIFAR-10[1] RF large $d$ | 50,000 | 5,551 | 8,162.84 | $2.8 \cdot 10^{-1}$ |
| CIFAR-10[1] RF small $d$ | 50,000 | 452 | 8,403.31 | 13.18 |
| CIFAR-5m[2] (all) | 5 million | 3,072 | 1,195,595.52 | $1.03 \cdot 10^{-1}$ |
| CIFAR-5m[2] (car/plane) | 1 million | 3,072 | 258,599.09 | $1.7 \cdot 10^{-2}$ |
| Gaussian *under parameterized* | 2,000 | 100 | 29.35 | 12.4 |
| Gaussian *equal* | 2,000 | 1,930 | 4.06 | $3.4 \cdot 10^{-4}$ |
| Gaussian *over parameterized* | 2,000 | 100,000 | 1.30 | $7.4 \cdot 10^{-1}$ |
| Gaussian-RF *under parameterized* | 2,000 | 100 | 66.38 | 3.95 |
| Gaussian-RF *equal* | 2,000 | 1,467 | 27.15 | $6.2 \cdot 10^{-3}$ |
| Gaussian-RF *over parameterized* | 2,000 | 316,227 | 21.77 | $6.6 \cdot 10^{-2}$ |
| MNIST[3] (all) | 60,000 | 784 | 5,562.79 | $1.1 \cdot 10^{-2}$ |
| MNIST[3] RF large $d$ | 60,000 | 5,551 | 4,249.29 | $2.8 \cdot 10^{-1}$ |
| MNIST[3] RF small $d$ | 60,000 | 452 | 4,564.77 | 15.09 |

[1] [34]  [2] [54]  [3] [37]

on (1) with $\delta = 0.01$. Empirical volterra solver was applied to match the multi-pass setting using the same variables. An empirical covariance $\boldsymbol{\Sigma}_\sigma(\boldsymbol{W})$ computed using all 1 million samples. Streaming SGD using constant learning rate $\gamma = 0.8$ applied to the expected risk using the empirical covariance $\boldsymbol{\Sigma}_\sigma(\boldsymbol{W})$. As the $\ell^2$ regularization parameter $\delta$ is hit by a factor of $n$, in the streaming setting, the regularization is set to 0.0. Empirical Volterra using the eigenvalues of $\boldsymbol{\Sigma}_\sigma(\boldsymbol{W})$ with $\gamma = 0.8$ and $\delta = 0.0$ matched the SGD steaming setting.

- *Random features theory.*

- *ICR, Figure 2*: Graph of the ICR under the assumption that the normalized trace of $\nabla^2 \mathcal{L}$ is 1.0, that is, $\frac{1}{n}\operatorname{tr}(\nabla^2 \mathcal{L}) = 1.0$. All data sets, MNIST, CIFAR-10, and CIFAR-5m are standardized (i.e., entries normalized so that mean 0.0 and variance 1.0). Largest and smallest (non-zero) eigenvalues of the feature covariance reported. For the random features set-up (RF), standard Gaussian matrix $\boldsymbol{W} \in \mathbb{R}^{n_0 \times d}$ where $n_0$ is the underlying number of features from the data set and $d$ ranged from $10^{2.5}$ to $10^{3.9}$ was applied to the data set followed by an entry-wise activation standardized ReLu. Reported (dashed lines) are the largest and smallest eigenvalues after applying the standardized ReLu and making the normalized trace equal to 1.0. In the Gaussian set-up, the number of samples $n$ was fixed at 2000 and $d$ ranged from $10^2$ to $10^5$; entries of $\boldsymbol{A}$ standard Gaussians. In the random features Gaussian (Gaussian-RF), we fixed the samples $n = 2000$ and $n_0 = 100$ and varied

the $d = 10^2$ to $10^{5.5}$. Largest and smallest eigenvalues of $\sigma(\boldsymbol{XW})^T \sigma(\boldsymbol{XW})$ reported after making the normalized trace 1.0.

- *Gaussian linear regression streaming, Figure 5*: Simple linear regression with targets from a generative model, $\boldsymbol{b} = \boldsymbol{A}\boldsymbol{\beta} + \boldsymbol{\xi}$; signal $\boldsymbol{\beta} \sim N(\boldsymbol{0}, \frac{1}{d}\mathbf{I}_d)$ and noise $\boldsymbol{\xi} \sim N(\boldsymbol{0}, \frac{0.04}{d}\mathbf{I})$. A $(n \times 2000)$ data matrix $\boldsymbol{A}$ with $\boldsymbol{A}_{ij} \sim N(0, 1/2000)$ with various $n$ values (see figure). SGD with constant learning rate $\gamma = 0.8$ initialized at $\boldsymbol{x}_0 = \boldsymbol{0}$ was applied to the linear regression problem with a regularization parameter of 0.01, see training loss and excess risk in linear regression in Appendix B.1. In this setting, the covariance of the expected risk is explicitly given by $\mathbf{I}_d/d$. A new data point $\boldsymbol{a} \sim N(0, \frac{1}{d}\mathbf{I}_d)$ and $b = \boldsymbol{a}\boldsymbol{\beta} + 0.2Z$ with $Z \sim N(0,1)$ generated and the expected risk computed as $(\boldsymbol{a}\boldsymbol{x}_t - b)^2$ where $\boldsymbol{x}_t$ are the iterates of SGD. Empirical volterra solver used with grid points $\approx$ number of iterations of SGD.

- *Gaussian linear regression concentration, Figure 6*: Simple linear regression with targets from generative model, $\boldsymbol{b} = \boldsymbol{A}\boldsymbol{\beta} + \boldsymbol{\xi}$; signal $\boldsymbol{\beta} \sim N(\boldsymbol{0}, \frac{1}{d}\mathbf{I}_d)$, noise $\boldsymbol{\xi} \sim N(\boldsymbol{0}, \frac{1.5^2}{n}\mathbf{I}_n)$. Matrix $\boldsymbol{A} \in \mathbb{R}^{n \times d}$ is row normalized and $\frac{d}{n} = 0.9$ for $n = \{100, 400, 1600, 6400\}$. 10 runs of SGD with constant learning $\gamma = 0.8$ started at $\boldsymbol{x}_0 \sim N(0, \frac{4}{n}\mathbf{I}_d)$ applied to the $\ell^2$-regularized least squares problem with $\delta = 0.1$, see training loss and excess risk in linear regression in Appendix B.1. 80% confidence interval (shaded) depicted in Figure 6. Volterra equation solver used with grid points approximately the same as epochs. Expected risk computed as in Figure 5. Concentration around the Volterra equation occurs as $n$ (or $d$) $\rightarrow \infty$ across different risk functions.