# OpenReview forum: "Implicit Regularization or Implicit Conditioning? Exact Risk Trajectories of SGD in High Dimensions"
_NeurIPS.cc/2022/Conference — NeurIPS 2022 Accept_

### Official Review · Reviewer_Nf1A · 2022-07-03

**Rating:** 5
**Confidence:** 4
**Soundness:** 2 fair
**Presentation:** 2 fair
**Contribution:** 2 fair

**Summary:**

This paper uses SDE (so-called HSGD) to study the optimization property of both streaming and multi-pass SGD over the quadratic functions. First, they show that the risk of SGD is no better than GD, showing that there is no implicit regularization in this setting. Next, they show the connection between SGD and HSGD, enabling them to use HSGD to study the properties of SGD. Their main contribution is to show that SGD enjoys a different condition number that sometimes leads to a better convergence rate over GD. Moreover, they also study the streaming SGD using the same framework and relate them to compare the generalization.

**Questions:**

Please see the weakness.

**Limitations:**

Please see the weakness.

**Strengths And Weaknesses:**

### Strengths
* The most interesting result is to show that SGD has a different condition number for **convex quadratic functions**. In particular, SGD outperforms GD if the spectrum of the Hessian has some outliers, which is usual in realistic applications, see Figure 2.
* The Volterra Dynamic, though not first proposed by this paper, its analysis might be of independent interest to study the behavior of SGD.

### Weaknesses
* In Figure 2, they run random feature models for realistic applications, i.e., CIFAR-10, CIFAR-5m, showing that in these cases the Hessian is ill-conditioned, supporting their main arguments that SGD outperforms GD on these realistic applications. However, note that the random feature model is a convex optimization problem, which differs a lot from realistic neural networks. Hence, small ICR on the random feature model cannot be used to show that SGD outperforms GD on realistic optimization problems **directly**. Next, the author should provide the "real" convergence comparison between SGD and GD even for the random feature models to support their main observation. ICR is just an indicator for the comparison between SGD and GD. There might be an approximation error due to the usage of HSGD and also the dimension, iteration time is finite. Hence, without this simulation, I am not convinced that SGD does outperform GD in terms of convergence speed.

* The argument that there is no implicit regularization might be trivial. For convex optimization, **any** algorithms, if convergent, then they should converge to the same function value (for strongly convex cases, they converge exactly to the same point). To my best knowledge, all the previous papers studying the implicit regularization focus on nonconvex optimization.

* The introduction of Volterra Dynamics is limited, which is very unfriendly to those unfamiliar with it. It would be good if additional background information can be provided.
* Missing title in the reference [36].

---

> ### Author Response · Authors · 2022-08-02
> **Response to Reviewer Nf1A**
>
> **Response to Weaknesses**
>
> *1. In Figure 2, they run random feature models for realistic applications, i.e., CIFAR-10, CIFAR-5m, showing that in these cases the Hessian is ill-conditioned, supporting their main arguments that SGD outperforms GD on these realistic applications. However, note that the random feature model is a convex optimization problem, which differs a lot from realistic neural networks. Hence, small ICR on the random feature model cannot be used to show that SGD outperforms GD on realistic optimization problems **directly**.*
>
> Indeed, for non-least squares problems the ICR changes over time in the optimization.  So we only show the problem at initialization in some sense. For neural networks with wide hidden layers and whose dynamics are well-approximated by the neural tangent kernel, the ICR of the NTK will not vary over time and our analysis should apply. Of course, one could debate whether such a problem is in fact realistic, and for situations in which the NTK does evolve substantially over time our analysis would not directly apply.
>
> Having said that, we would like to add that part of the motivation to do a thorough analysis of linear regression is that when presented with a truly challenging problem, like the analysis of a deep neural network, having good information about linear regression provides an important prior and useful insight about what we might expect to observe because of high-dimensionality itself, even in the convex quadratic setting. In fact, we find it quite remarkable that there are still things to learn about optimization of convex quadratics, and our novel results highlight the importance of studying optimization in high dimensions.
>
> *2. Next, the author should provide the "real" convergence comparison between SGD and GD even for the random feature models to support their main observation. ICR is just an indicator for the comparison between SGD and GD. There might be an approximation error due to the usage of HSGD and also the dimension, iteration time is finite. Hence, without this simulation, I am not convinced that SGD does outperform GD in terms of convergence speed.*
>
> We think you make a fair point, and so present a simulation that we believe adds some strength to the meaning of ICR, showing how as we vary the ICR, SGD vs GD performance changes. The following is the performance on a least-square problem of fixed aspect ratio, where we can change the ICR by varying the power-law exponent of the covariance spectra.
>
> [https://anonymous.4open.science/r/revisionstuff-C899/BetaSmallICR_overdetermined.pdf](https://anonymous.4open.science/r/revisionstuff-C899/BetaSmallICR_overdetermined.pdf)
>
> (Notes: dotted (GD), dashed (SGD), solid (MGD).  All overdetermined, but with L2 regularizer to stabilize MGD.  ICR plotted at the bottom.  Time measured in dot-products.)
>
> *3. The argument that there is no implicit regularization might be trivial. For convex optimization, **any** algorithms, if convergent, then they should converge to the same function value (for strongly convex cases, they converge exactly to the same point). To my best knowledge, all the previous papers studying the implicit regularization focus on nonconvex optimization.*
>
> We agree that Lemma 1 is a triviality, and we are not trying to claim this as a novelty. Your argument is very clear, and ourproof is a direct consequence of Jensen’s inequality and is a 1-line proof.  We have formulated it as a lemma to emphasize that in this case, implicit regularization cannot occur.
>
> *4. The introduction of Volterra Dynamics is limited, which is very unfriendly to those unfamiliar with it. It would be good if additional background information can be provided.*
>
> Additional background and references for Volterra dynamics, which appear frequently in filtering, population dynamics, and the renewal problem, to name a few, are now available in the new version of the uploaded Supplementary Materials (Appendix A).
>
> *5. Missing title in the reference [36].*
>
> Thanks. We have fixed this.

---

> > ### Comment · Reviewer_Nf1A · 2022-08-07
> > **Thanks for your response**
> >
> > Thanks for the author's response! Most of my concerns have been addressed by the author. In particular, I like the introduction about the Volterra Dynamics in Appendix A. Hence, I decided to increase my score from 4 to 5.
> >
> > However, about the first point, I still have some concerns. As pointed out by the author, the ICR changes over time in the optimization. However, if the argument is correct, then it might not be difficult to show that the ICR has a consistent pattern along the training process. For example, the ICR is always smaller than some threshold in the case that SGD outperforms GD. Hence, I highly suggest the author add the simulation about the evolution of ICR during the training process for realistic applications.

---

> > > ### Author Response · Authors · 2022-08-09
> > > **Response to Reviewer Nf1A Questions**
> > >
> > > We agree with the reviewer that it is very interesting to examine the evolution of the ICR during training. We have added an experiment that shows the ICR is roughly constant over training **[see Supplementary Materials zip file]**, yielding at least some preliminary empirical support for the practical utility of our analysis. We are currently running additional experiments on more architecture and datasets, which we will include in the final version (the computational burden is significant and these results may take a while to be finalized).
> > >
> > >
> > > We would like to emphasize though that conducting a rigorous and compelling analysis is more challenging than it might seem, owing to a number of reasons:
> > >
> > > 1. It is not immediately clear which matrix to compute the ICR of in the non-convex setting, owing to the presence of negative eigenvalues. Perhaps the most natural choices would be the Gauss-Newton approximation to the Hessian or the Neural Tangent Kernel. In the case of squared error, these matrices have the same spectrum (and as such was the choice for the above experiments), but for other loss functions they may differ and there is some ambiguity regarding which matrix to study.
> > >
> > > 2. For realistic models and dataset sizes, there are serious computational hurdles to extracting the ICR, especially in obtaining high-precision estimates of the minimum eigenvalue. Indeed, constructing efficient estimators for spectral statistics of curvature matrices has been the subject of an active recent research direction [1-2]. While these advances have made possible the estimation of the spectrum during training, they are still not able to reliably achieve high-precision information about the smallest eigenvalues, which would be necessary for estimating the ICR.
> > >
> > > 3. There is no analogue of the Nesterov optimal momentum parameters for the non-convex setting, and therefore it is not clear how to construct the appropriate baselines against which to compare SGD. Grid search is a possibility, but it is expensive and still there would be ambiguity with respect to the criterion for optimality, since the late-time asymptotics will not be accessible experimentally.
> > >
> > > 4. For natural datasets and realistic architectures, there is not always a well-motivated “knob” that would allow for tuning the ICR while leaving everything else fixed. (We have looked at a number of preprocessing operations, e.g. regularized ZCA whitening, but for many architectures they do not have a strong impact on the initial ICR.) As such, it is challenging to directly investigate the impact of ICR on the favorability of SGD vs M-GD.
> > >
> > > [1] Ghorbani, B., Krishnan, S., and Xiao, Y. *An Investigation into Neural Net Optimization via Hessian Eigenvalue Density*, ICML, 2019
> > >
> > > [2] Adams, RP., Pennington, J., Johnson, MJ, Smith, J, Ovadia, Y, Patton, B, and Saunderson J., *Estimating the Spectral Density of Large Implicit Matrices*, 2018

---

### Official Review · Reviewer_wJVQ · 2022-07-11

**Rating:** 6
**Confidence:** 4
**Soundness:** 3 good
**Presentation:** 3 good
**Contribution:** 3 good

**Summary:**

The authors of the paper present the following contributions:
- They show that, in the high dimensional limit, multipass SGD behaves, in terms of empirical and population loss, as a SDE with a particular noise covariance
- They further show that, in the high dimensional limit, this SDE converges to a deterministic Volterra Dynamics
- They analyse this dynamics to compare to the one of the gradient flow showing that noise negatively impact the population loss but can accelerate convergence.


**Questions:**

I have one question for the SDE limit of the model: could the authors explain why the quadratic case is special to derive the SDE limit ? How could it be extended to a general convex loss ? or even to a non-convex setup ?



**Limitations:**

As I already discussed the limitations in the previous boxes, I'll put the conclusion of my review here. I truly think this is a nice paper, with solid result and an interesting high-dimensional limit dynamics.  I now put a weak accept, but will be happy to raise my score if the authors temper a bit the overselling part, correct the minor flaws and the referencing.

**Strengths And Weaknesses:**

### **Strengths**

- First I have to say that the paper is well written, pleasant to follow and well illustrated by the experiments.
- Second and more importantly, I really like the results concerning the high dimensional set-up: comparing the losses dynamics to the one of an explicit SDE that can be studied is a good idea. And even I had no time to check the proof, the result seems sound. Then, concentration to Lotka-Volterra dynamics is also a nice development of the analysis!


### **Weaknesses**

- Perhaps the first weakness is the limitation of the result: the authors are very precise in the specific setting of linear regression, provide some good material to analyse it, but it is hard to conclude anything deep from their analysis.
- I know this is a large tendency in the ML community, but I think that overselling the results is counter productive when writing a paper. A good example is the title of the paper where the fact that the focus is on Least-squares should be written. Also, trying to systematically conclude that SGD is "better" or that this can explain the amazing performance of SGD in practice is a bit overselling. The results of the authors being already strong, there is no need to oversell the paper like this.
- In the same direction, I find the introduction confusing and the references to the literature incomplete. I think that the authors should be more specific and really focus on the least squares literature, and not try to refer to the deep learning one (or maybe just at the end to motivate further investigations). In terms of literature, lign 50, the multipass is not properly covered and [1,2] among others are worth mentioning.
- The paragraph on **Diffusion approximations and homogenized SGD** should be rewritten. A lot of work has been done to model SGD as a diffusion. [3] properly explains how to model SGD with a SDE, [4] analyses it respecting the geometry of the noise and the sentence *we lack a precise connection between a concrete SLD and a practical nonconvex learning problem* is simply wrong as [5] studied exactly a SDE model in a non-convex setting. Finally, to be consistent with the rest of the paper, it would be great that $\gamma$ appears in the noise term of equation (5). Note also that in the isotropic case, if $\Sigma$ is proportional to the identity, the invariant measure is not proportional to $e^{-f}$ but to $e^{- C_{\gamma} f}$, with $C_{\gamma}>0$ some constant.


[1] Junhong Lin and Lorenzo Rosasco. Optimal rates for multi-pass stochastic gradient methods. Journal of Machine Learning Research, 18(97):1–47, 2017

[2] Loucas Pillaud-Vivien, Alessandro Rudi, and Francis Bach. Statistical optimality of stochastic gradient descent on hard learning problems through multiple passes. In Advances in Neural Information Processing Systems, pages 8125–8135, 2018.

[3] Qianxiao Li, Cheng Tai, and Weinan E. Stochastic modified equations and dynamics of stochastic gradient algorithms i: Mathematical foundations. Journal of Machine Learning Research, 20(40):1–47, 2019.

[4] Stephan Wojtowytsch. Stochastic gradient descent with noise of machine learning type. Part II: Continuous time analysis, *Preprint*, 2021.

[5] Scott Pesme, Loucas Pillaud-Vivien, and Nicolas Flammarion. Implicit bias of sgd for diagonal linear networks: a provable benefit of stochasticity. Advances in Neural Information Processing Systems, 34, 2021.


### **Minor Flaws**
- [36] is an empty reference
- lign 92: miss a $dt$ in equation (5) for the drift
- lign 166: explain with you take $tn$ as the time for SGD
- lign 250: parenthesis problems

---

> ### Author Response · Authors · 2022-08-02
> **Response to Reviewer wJVQ (1/2)**
>
> **Response** (1/2)
>
> We thank the reviewer for their careful reading and constructive feedback. We have fixed the minor errors found by the reviewer in the new version of the main paper and supplementary materials. **We encourage the reviewer to look at the new revisions.**
>
> **Comments on Weaknesses**
>
> *1. Perhaps the first weakness is the limitation of the result: the authors are very precise in the specific setting of linear regression, provide some good material to analyse it, but it is hard to conclude anything deep from their analysis.*
>
> Indeed our analysis is restricted to the case of linear regression, and we agree that it is hard to draw general conclusions beyond this setting. We might expect our results to extend to the specific case of neural networks with wide enough layers and for which the dynamics are well approximated by the neural tangent kernel. But it is true that truly non-convex models, the dynamics of feature learning, etc., are all well outside the scope of our analysis.
>
> Having said that, we would like to add that part of the motivation to do a thorough analysis of linear regression is that when presented with a truly challenging problem, like the analysis of a deep neural network, having good information about linear regression provides an important prior and useful insight about what we might expect to observe because of high-dimensionality itself, even in the convex setting. In fact, we find it quite remarkable that there are still things to learn about optimization of convex quadratics, and our novel results highlight the importance of studying optimization in high dimensions.
>
> *2. I know this is a large tendency in the ML community, but I think that overselling the results is counter productive when writing a paper. A good example is the title of the paper where the fact that the focus is on Least-squares should be written. Also, trying to systematically conclude that SGD is "better" or that this can explain the amazing performance of SGD in practice is a bit overselling. The results of the authors being already strong, there is no need to oversell the paper like this.*
>
> We completely agree with the reviewer about this point and are very sorry that our paper came across this way. Our intention was not to argue that SGD is generally “better,” and in fact our results do not even support that conclusion in the least squares setting. To the contrary,  we rather sought to argue that SGD actually incurs a mild generalization performance loss, but point out that it *can* have optimization advantages that become evident in high dimensions. To fully understand the performance of SGD, it is necessary to examine its effect on both generalization and on convergence rates, though it is often hard to disentangle these effects in empirical analyses in practical models. Our in-depth analysis in the least squares setting allows for us to do this theoretically, and provides some baselines for future analyses in the non-convex setting, and also underscores the importance of accounting for high-dimensionality.
>
> *3. In the same direction, I find the introduction confusing and the references to the literature incomplete. I think that the authors should be more specific and really focus on the least squares literature, and not try to refer to the deep learning one (or maybe just at the end to motivate further investigations). In terms of literature, lign 50, the multipass is not properly covered and [1,2] among others are worth mentioning.*
>
> *[1] Junhong Lin and Lorenzo Rosasco. Optimal rates for multi-pass stochastic gradient methods. Journal of Machine Learning Research, 18(97):1–47, 2017*
>
> *[2] Loucas Pillaud-Vivien, Alessandro Rudi, and Francis Bach. Statistical optimality of stochastic gradient descent on hard learning problems through multiple passes. In Advances in Neural Information Processing Systems, pages 8125–8135, 2018.*
>
> Thank you for the references.  We have added them into the new version of the Supplemental Materials (see Appendix A) in which we have a non-exhaustive additional related work section. We encourage the reviewer to look at the new uploaded Supplementary Materials.

---

> > ### Author Response · Authors · 2022-08-02
> > **Response to Reviewer wJVQ (2/2)**
> >
> > **Response** (2/2)
> >
> > *4. The paragraph on **Diffusion approximations and homogenized SGD** should be rewritten. A lot of work has been done to model SGD as a diffusion. [3] properly explains how to model SGD with a SDE, [4] analyses it respecting the geometry of the noise and the sentence we lack a precise connection between a concrete SLD and a practical nonconvex learning problem is simply wrong as [5] studied exactly a SDE model in a non-convex setting. Finally, to be consistent with the rest of the paper, it would be great that γ appears in the noise term of equation (5).*
> >
> > *[3] Qianxiao Li, Cheng Tai, and Weinan E. Stochastic modified equations and dynamics of stochastic gradient algorithms i: Mathematical foundations. Journal of Machine Learning Research, 20(40):1–47, 2019.*
> >
> > *[4] Stephan Wojtowytsch. Stochastic gradient descent with noise of machine learning type. Part II: Continuous time analysis, Preprint, 2021.*
> >
> > *[5] Scott Pesme, Loucas Pillaud-Vivien, and Nicolas Flammarion. Implicit bias of sgd for diagonal linear networks: a provable benefit of stochasticity. Advances in Neural Information Processing Systems, 34, 2021.*
> >
> > Thanks for the references.  We discussed [3] and the related body of work in a response above.  Paper [4] we agree should be added.  [5] is very nice work, and we agree it should be referenced somewhere, though it may not really constitute “a practical nonconvex learning problem,” at least as we envisioned that phrase to mean, and we would object somewhat to characterizing that sentence as “simply wrong.” Nevertheless, we agree with the spirit of the reviewer’s complaint and will reword that sentence and rewrite the entire paragraph. To that end, we have added a subsection on *Diffusion Approximations and SGD*  into the new version of the Supplemental Materials (see Appendix A) in which we have a non-exhaustive description of prior work on SDEs and SGD and a detailed description of differences between SME and HSGD. We encourage the reviewer to look at the new uploaded Supplementary Materials, Appendix A. We will incorporate this material into the main text using the additional page available for the camera ready version.
> >
> > *5. Note also that in the isotropic case, if Σ is proportional to the identity, the invariant measure is not proportional to e−f but to e−Cγf, with Cγ>0 some constant.*
> >
> > Thanks! This is fixed in the new uploaded version of the paper.
> >
> > **Response to Questions:**
> >
> > *1. I have one question for the SDE limit of the model: could the authors explain why the quadratic case is special to derive the SDE limit ? How could it be extended to a general convex loss ? or even to a non-convex setup ?*
> >
> > GLMs (or some class of them) might be  solvable.  We thought about the linear hidden-layer  setup some, but there are important phenomenological differences.  The model of [5] could possibly be approached.

---

> > > ### Comment · Reviewer_wJVQ · 2022-08-02
> > > **Rebuttal**
> > >
> > > I would like to thank the authors for their rebuttal and their work.

---

### Official Review · Reviewer_XeAd · 2022-07-12

**Rating:** 5
**Confidence:** 4
**Soundness:** 3 good
**Presentation:** 2 fair
**Contribution:** 3 good

**Summary:**

This paper studies the generalization ability of the multi-pass SGD on high-dimensional convex quadratics by relating it to a stochastic differential equation called homogenized stochastic gradient descent (HSGD). The authors show that using the HSGD, a precise risk trajectory of SGD can be established, which reveals the conditions on the data distribution that SGD is more efficient than GD. The authors further extend the analysis to streaming SGD and show its inability to capture certain salient features compared to multi-pass SGD.



**Questions:**

Please refer to the weakness section.

**Limitations:**

Please refer to the weakness section.

**Strengths And Weaknesses:**

Overall, the strengths of this paper include

(1) establishing an approximation result between SGD and HSGD.
(2) showing that SGD negatively impacts generalization performance.
(3) showing how SGD accelerates convergence.
(4) showing the inability of streaming SGD.

Weaknesses are as follows:
(1) Although the authors provide a precise characterization of the generalization error of SGD, the formula of the $Psi_t$ and $\Omega_t$ are still difficult to follow. There lacks a good interpretation of the developed results, especially in the non-asymptotic setting that $t$ is not approaching infinity. Note that for many high-dimensional linear regression problems, GD/SGD with early stopping can give good generalizable solutions while the overfitting will occur if $t\rightarrow \infty$.

(2) Additionally, in some cases the condition number $\kappa$ could be extremely large (when the matrix $T$ or $A$ has a fast decaying eigenspectrum). Then the convergence rate to $\Psi_\infty$ and $\Omega_\infty$ may not be that interesting as they will be super slow and people tend to early stop the optimization algorithm.

(3) In section 3.2, a detailed explanation about why SGD is more efficient than GD is missing. It seems that the authors only claim that SGD with a constant learning rate can match the convergence of GD, but how this lead to the argument that SGD is more efficient is missing. The authors may need to provide a rigorous comparison between the efficiency of SGD and GD/M-GD.

(4) Lemma 1 is not new in the literature (at least in the case of $\delta=0$). The following work (see their Theorem 1) has shown that for any feasible learning rate, the excess risk of multi-pass SGD must be greater than or equal to GD,  then taking the limit $learning rate \rightarrow 0$ can imply the results of Lemma for the case of $\delta=0$ The authors may need to comment this in the surrounding text.

D. Zou, J. Wu, V. Braverman, Q. Gu, and S. M. Kakade. Risk Bounds of Multi-Pass SGD for Least Squares in the Interpolation Regime. arXiv preprint arXiv:2203.03159, 2022.

---

> ### Author Response · Authors · 2022-08-02
> **Response to Reviewer XeAd (1/2)**
>
> **Comments on Weaknesses** (1/2)
>
> *1. Although the authors provide a precise characterization of the generalization error of SGD, the formula of the Psit and Ωt are still difficult to follow.*
>
> We agree that $\Omega_t$ and $\Psi_t$ may at first appear complicated as they are not defined through closed-form expressions but rather in terms of solutions to an integral (Volterra) equation. The level of complexity here is similar to that of a linear differential equation, and as such we believe the presentation is not unduly complicated or unwarranted given the power of the results. Having said that, we recognize that not everyone has familiarity with Fredholm theory or the theory of integral equations and we added some background material to Appendix A.  We have also provided code for the evaluation of these quantities: they can be numerically evaluated efficiently by what amounts to the power method.  We have also analyzed their asymptotic behavior, which can be examined somewhat more explicitly. We believe that these various efforts are sufficient to help readers follow and understand these formulas. If the reviewer has additional specific requests in this direction we would be happy to accommodate.
>
>  *There lacks a good interpretation of the developed results, especially in the non-asymptotic setting that t is not approaching
> infinity.*
>
> We agree that the behavior for intermediate $t$  is important, and one of the main strengths of our results is that our formulas give precise predictions for the entire learning trajectory. As is often the case in the perhaps more familiar setting of linear differential equations, some of the details at intermediate $t$ may not be determinable without solving the equations numerically; we have performed such numerical evaluations in many settings, see e.g. Figs 1 and 3, which we believe should aid in the interpretation of the results. In certain cases, some high-level conclusions can be determined more generally, and we have developed those analyses as well; see e.g. Lemma 1, which shows that SGD generalizes worse than gradient flow at all intermediate times.
>
> *Note that for many high-dimensional linear regression problems, GD/SGD with early stopping can give good generalizable solutions while the overfitting will occur if t→∞.*
>
> Yes, it is often the case that in problems with low SNR or insufficient explicit regularization that both GD and SGD generalize better with early stopping than if t->infinity (see e.g., Figure 1).  But it's also true that SGD, with early stopping, generalizes worse than the corresponding gradient flow (Lemma 1), and MGD actually struggles even more on problems with large ICR.  Note MGD performance gains usually appear when solving for the ‘last eigenmodes’, which are frequently exactly the ones which are being screened by the practitioner by early stopping – actually appears nicely in
>
> [https://anonymous.4open.science/r/revisionstuff-C899/BetaSmallICR_overdetermined.pdf](https://anonymous.4open.science/r/revisionstuff-C899/BetaSmallICR_overdetermined.pdf)
>
> (Notes: dotted (GD), dashed (SGD), solid (MGD).  All overdetermined, but with L2 regularizer to stabilize MGD.  ICR plotted at the bottom.  Time measured in dot-products.)
>
> So the story at early stopping is largely the same as t=infinity.
>
> *2. Additionally, in some cases the condition number κ could be extremely large (when the matrix T or A has a fast decaying eigenspectrum). Then the convergence rate to Ψ∞ and Ω∞ may not be that interesting as they will be super slow and people tend to early stop the optimization algorithm.*
>
> This is certainly an excellent point.  And we give the actual finite time formula, which is expressed using the Volterra equation.  But as you note, this is complicated, and so it’s helpful to have a simplification that allows for an easy proxy for the behavior of Psi and Omega which are helpful in some cases.
>
> Also, early stopping can be interpreted as screening some of the eigenvalues from the problem, as they are too small to have participated in the optimization.  In this case, you are actually using the smallest ‘effective’ eigenvalue.  In every real-world optimization analysis, you have to do this.  An example of this is MNIST itself, which has a few eigenvalues 10^{-5} smaller in magnitude than the next smallest one.  These need to be removed to understand the correct smallest ‘effective’ eigenvalue.  And the correct way to compute Psi_infty and Omega_infty is with this smallest effective eigenvalue.  This would also be the case with early stopping, where you remove eigenvalues smaller than the threshold for participation.

---

> > ### Author Response · Authors · 2022-08-02
> > **Response to Reviewer XeAd (2/2)**
> >
> > **Comments on Weaknesses** (2/2)
> >
> > *3. In section 3.2, a detailed explanation about why SGD is more efficient than GD is missing. It seems that the authors only claim that SGD with a constant learning rate can match the convergence of GD, but how this lead to the argument that SGD is more efficient is missing. The authors may need to provide a rigorous comparison between the efficiency of SGD and GD/M-GD.*
> >
> > Theorems 4 and 5 establish the asymptotic convergence rates for SGD and M-GD, and lines 228-229 provide the interpretation of these results: SGD will converge in an ICR-multiple number of epochs that M-GD requires (so lower ICR favors SGD). Of course, these are just asymptotic rates, and the reviewer is absolutely right that they might not fully reflect what happens in practice or at finite $t$. As such, we have conducted some additional simulations that we believe adds some strength to the meaning of ICR.
> >
> > The following is the performance on a least-square problem of fixed aspect ratio, where we can change the ICR by varying the power-law exponent of the covariance spectra.
> >
> > [https://anonymous.4open.science/r/revisionstuff-C899/BetaSmallICR_overdetermined.pdf](https://anonymous.4open.science/r/revisionstuff-C899/BetaSmallICR_overdetermined.pdf)
> >
> > (Notes: dotted (GD), dashed (SGD), solid (MGD).  All overdetermined, but with L2 regularizer to stabilize MGD.  ICR plotted at the bottom.  Time measured in dot-products.)
> >
> > We would like to emphasize that we are not trying to argue that SGD ‘is more efficient’ than GD as a blanket statement.  It’s rather that there is a type of problem which is poorly conditioned (meaning low ICR) in which full batch methods struggle and SGD will perform better.  We’re open to other suggestions on how to provide a rigorous comparison between the efficiency of SGD and GD/M-GD, especially at finite $t$.
> >
> > *4. Lemma 1 is not new in the literature (at least in the case of δ=0). The following work (see their Theorem 1) has shown that for any feasible learning rate, the excess risk of multi-pass SGD must be greater than or equal to GD, then taking the limit learningrate→0 can imply the results of Lemma for the case of δ=0*
> >
> > *The authors may need to comment this in the surrounding text.*
> >
> > *D. Zou, J. Wu, V. Braverman, Q. Gu, and S. M. Kakade. Risk Bounds of Multi-Pass SGD for Least Squares in the Interpolation Regime. arXiv preprint arXiv:2203.03159, 2022.*
> >
> > Zou-Wu-Braverman-Gu-Kakade is a nice paper, and we’re happy to discuss it further.  Lemma 1 is a 1-sentence application of Jensen's inequality, and we don’t mean to claim it as a novelty. We have added the reference (actually their other paper as it deals with multi-pass SGD) to the new version of the main paper.

---

### Official Review · Reviewer_xitN · 2022-07-14

**Rating:** 4
**Confidence:** 2
**Ethics Flag:** Yes
**Soundness:** 2 fair
**Presentation:** 1 poor
**Contribution:** 2 fair

**Summary:**

This work proposes HSGD as a continuous approximation to SGD (multi pass version). It first shows an approximation theorem that justifies the closeness between the continuous HSGD and the discrete SGD. Then by studying HSGD it characterizes the limiting training and population risks through Volterra dynamics. Based on the theorems, the paper concludes that (1) multi-pass SGD does not have implicit bias over GD, at least in the setting of least square, (2) SGD has an effect of implicitly conditioning, that accelerates the convergence.

**Questions:**

Please see above.

**Ethics Review Area:**

["I don’t know"]

**Limitations:**

Please see above. It seems most of the theorems are from existing works. Thus this particular work presents little delta to me. Another thing bothers me is that, the authors did not try to explicitly discuss this issue in their main text.

**Strengths And Weaknesses:**

# Strengths
+ The implicit conditioning could be an interesting perspective to understand the effectiveness of SGD.

# Weakness
- I find the statement of related literature could be improved.

- please correct the bib formate of reference [36]

- Line 49, could have mentioned [55]. In particular, [55] showed a similar result that multi-pass SGD generalizes worse than GD in the linear regression setting. The authors should mention this when stating their contribution and in Sec 3.1

- Thm 1 and Thm2 seem to from [39]. The authors should explicitly mention [39] after their Thms 1 and 2. Moreover, could you please explain the delta of thms 1 & 2 compared to that in [39]?

- Thms 4 and 5 are from [36, 37, 38] (as explicitly mentioned in line 901 in Appendix). In this sense, the authors should explicitly clarify this issue in their main text. The current writing has caused a huge misunderstanding about its true contribution when I first go through the paper.

- Line 90. Could you explain why SGD with momentum degenerates to SGD?

- Lemma 1. What is $\gamma$ here? Note that a somewhat related lemma has been shown in [55].

- Line 119-124. I am actually confused, because it has been long known that a continuous SME approximates SGD, see [L 2017] and [H 2017].

- Line 148. I believe [21] is a wrong citation here. Could you point out where in [21] iid sub gaussian is assumed?

- Line 158. The notation is a bit confusing, is $\gamma$ still referring to the stepsize?

- Thm 1 is hard for me to interpret. First of all, how do you compare this result to the SME approximation proved by [L 2017] and [H 2017]?
Secondly, Could you provide some examplar regime where the approximation error is small? It seems for the error to be small one needs $d$ to be large, but on the other hand that implies $n$ needs to be large. How is the error affected by stepsize?

- Thm 2 is also hard for me to interpret.













[L 2017] Li Q, Tai C, Weinan E. Stochastic modified equations and adaptive stochastic gradient algorithms. InInternational Conference on Machine Learning 2017 Jul 17 (pp. 2101-2110). PMLR.

[H 2017] Hu W, Li CJ, Li L, Liu JG. On the diffusion approximation of nonconvex stochastic gradient descent. arXiv preprint arXiv:1705.07562. 2017 May 22.

---

> ### Author Response · Authors · 2022-08-02
> **Response to Reviewer xitN (1/2)**
>
> **Comments on Weaknesses** (1/2)
>
> * *I find the statement of related literature could be improved.*
>
> We have included an extensive longer related work section in Appendix A of the new version of the supplementary material and we encourage the reviewer to look at the new Appendix A in the Supplement. If the reviewer has additional papers that we missed, we would greatly appreciate the references. Unfortunately due to current space requirements, we could not add additional references to the main 9 pages, but we will utilize the extra page of the camera ready to do so, if the paper is accepted.
>
> * *Line 49, could have mentioned [55]. In particular, [55] showed a similar result that multi-pass SGD generalizes worse than GD in the linear regression setting. The authors should mention this when stating their contribution and in Sec 3.1*
>
> Our original discussion of paper [55] was perhaps limited because that work appeared on the arXiv only two months before the submission deadline. Nevertheless, we agree that the results are relevant, and the new version of our paper comments on [55] in the main text and expands on the discussion in Appendix A. We invite the reviewer to look at the new Supplementary material.
>
> * *Thm 1 and Thm2 seem to from [39]. The authors should explicitly mention [39] after their Thms 1 and 2. Moreover, could you please explain the delta of thms 1 & 2 compared to that in [39]?*
>
> Theorem 1 is indeed from [39]; however, we would like to emphasize that [39] only appeared on the arXiv in May and that it is a much more mathematical paper with a different target audience. Theorem 2 is novel and does not appear in [39]. Theorem 2 extends prior work to cover more SLDs and in particular applies to settings such as streaming.  We will clarify these points in the next revision.
>
> * *Thms 4 and 5 are from [36, 37, 38] (as explicitly mentioned in line 901 in Appendix). In this sense, the authors should explicitly clarify this issue in their main text. The current writing has caused a huge misunderstanding about its true contribution when I first go through the paper.*
>
> Thanks for checking the proof, but actually Theorems 4 and 5 are novel and do not appear in [36,37,38], which do not deal at all with generalization properties of SGD. It is true that papers [36,37,38] contain similar results for the convergence rates for the empirical risk, but they do not discuss the population risk at all. Given the other results of this paper, particularly Theorems 1 and 2, Theorems 4 and 5 largely follow with small additional arguments from the results of [36,37,38]. The adaptation of the proofs is not complicated, because for both cases, in the least squares setup, the population risk can be expressed as a function of empirical risk.  As we have the behavior of the empirical risk, we can derive the corresponding asymptotic behavior of the population risk.
>
> * *Line 90. Could you explain why SGD with momentum degenerates to SGD?*
>
> The citation has the theoretical argument.  Heuristically, in high dimensions, gradient estimators can be expected to be orthogonal.  If the momentum parameter is not sent to 1 with the dimensionality of the problem, then the lifetime of a single gradient estimate in the memory of the algorithm is shorter than the time it takes to interact with the other gradient estimates.  So for a fixed momentum parameter m (eg. 0.9) and in high dimensions, single batch SGD+momentum is actually pathwise-equivalent to SGD with effective learning rate $\gamma*m/(1-m)$: this comes from summing all contributions of a single gradient estimator as though they all occurred in orthogonal spaces.
>
> * *Lemma 1. What is γ here? Note that a somewhat related lemma has been shown in [55].*
>
> Gamma is defined in (5).  We should emphasize that Lemma 1 is not really claimed as a novelty: it is a direct consequence of Jensen’s inequality and is a 1-line proof.  We rather consider it as a triviality but formulate it as a lemma to emphasize that in this case, implicit regularization cannot occur.  Of course [55] is a very nice paper and we are happy to further our discussion of it.

---

> > ### Author Response · Authors · 2022-08-02
> > **Response to Reviewer xitN (2/2)**
> >
> > **Comments on Weaknesses** (2/2)
> >
> > * *Line 119-124. I am actually confused, because it has been long known that a continuous SME approximates SGD, see [L 2017] and [H 2017].*
> >
> > For space considerations in the original submission, we had to remove a longer discussion about these papers., but we certainly agree that they are a substantial and important theoretical contribution to the theory of diffusions in optimization/machine learning.  If accepted, we will happily restore some of this discussion, which we have temporarily added to Appendix A.
> > Essentially, there are a few critical differences between our setting and that of  L2017 and H2017:
> >
> > 1. L2017 and H2017 actually concern a different diffusion, the SME, which has a different covariance structure than H-SGD.  Even in the least squares setting, this covariance structure is quite complicated.  In particular, there is no analogue of the Volterra equation, and so it actually cannot be analyzed in the same way as H-SGD.  The fact that the generalization dynamics of H-SGD do not have dimension dependence is the key difference.
> >
> > 2. The time scale in L2017 and H2017, when transported to our setting, is a fixed number of _iterations_.  On that time scale, when the dimensions d ~~ n are large, SGD does nothing.  You need to run on the order of n iterations.  The theory from L2017 and H2017 no longer applies to this case.  In contrast, note that in Theorem 1, the number of iterations over which SGD and H-SGD are close is the same magnitude as the problem size (there is an n in the x_[nt]!)
> >
> > * *Line 148. I believe [21] is a wrong citation here. Could you point out where in [21] iid sub gaussian is assumed?*
> >
> > Rather, our analysis holds with the subgaussian assumption.  [21] is related, but just some finite number of moments is needed.
> >
> > * *Line 158. The notation is a bit confusing, is γ still referring to the stepsize?*
> >
> > Yes, $\gamma$ is always the stepsize constant.
> >
> > * *Thm 1 is hard for me to interpret. First of all, how do you compare this result to the SME approximation proved by [L 2017] and [H 2017]? Secondly, Could you provide some examplar regime where the approximation error is small? It seems for the error to be small one needs d to be large, but on the other hand that implies n needs to be large. How is the error affected by stepsize?*
> >
> > (See above for comparison to  [L 2017] and [H 2017])
> >
> > Your discussion is exactly correct.  When the problem-size is large (d and n are large), SGD can be compared to H-SGD.  The philosophy here is to make approximations to the behavior of SGD for large-d and large-n problems.
> >
> > If d=n=10, we have nothing to say.  When d and n are 10^4 or larger, then H-SGD and SGD will have a tiny difference. (Figure 3, e.g. – which is a good example of where the theory applies, here n and d vary, T=100, and the Volterra and SGD are on top of each other – HSGD would be too.)  Even d and n in the hundreds show good empirical agreement.
> >
> > The error has dependence on the stepsize through the constants, but the theorem actually covers non-convergent and convergent stepsizes alike (so SGD converges if and only if HSGD converges).  The key is that you can get a meaningful statement by fixing a gamma and changing the dimension.  When the data is standardized (row norms of A are 1), the interval [0,2) are the convergent values of gamma.  So it’s not necessary to put n–dependence into gamma.

---

### Author Response · Authors · 2022-08-02
**Comments to all reviewers**

We thank the reviewers for their constructive feedback and careful reading of our paper. We have implemented changes in the **newly uploaded version of our paper and Supplementary Materials**, which we encourage the reviewers to look at. Space limitations prevented us from including all changes in the main text itself, but we will do so for the camera ready. The main changes include:

1. Minor typos and fixes suggested by the reviewers
2. Reference to [Zou et al 2022]  in Lemma 1
3. Expanded non-exhaustive “Related work” section which includes all the references suggested by the reviewers plus a few more in Appendix A of the Supplementary Material. Space limited us to include this in the main text.
4. Expanded discussion of the “Diffusion Approximations and SGD” that includes detailed differences and limitations between SME [Li et al 2017] and its analysis [Hu et al 2017]. This is available in the new Supplementary Materials, Appendix A. (See more discussion below for individual questions).
5. Added background references and some basic convergence and limiting behavior results for Volterra equations. This is available in the new Supplementary Materials, Appendix A.
6. Added some additional experiments highlighting the role of ICR in the performance of SGD versus momentum ([see here](https://anonymous.4open.science/r/revisionstuff-C899/BetaSmallICR_overdetermined.pdf)). When the ICR goes below approximately 1, SGD begins to outperform momentum as *exactly* predicted. The experiment was performed on a least-square problem of fixed aspect ratio, where we can change the ICR by varying the power-law exponent of the covariance spectra.

---

> ### Author Response · Authors · 2022-08-09
> **Figures added**
>
> Dear reviewers and AC:
>
> The links to the anonymous github are proving unreliable.  The extra figures have been included in the supplementary material ZIP file.
> They are:
> 1) A figure showing the effect of ICR on training in a synthetic least-squares setup.  This was generated with data covariance which is power-law but bounded below.  By changing the power law exponent, we are able to tune the ICR and hence illustrate that low-ICR implies SGD is favored.
> 2) A figure showing loss curves and ICR in a 3-layer fully connected neural network with width 512 for CIFAR classification (planes vs. automobiles). The loss is the mean squared error. SGD applied with batch size 100, learning rate 0.1 over 40 epochs. Losses/Validation appear on the right axis. The ICR remains largely stable over the course of the training.
>
> Thanks for your consideration!

---

### Meta-Review · Area_Chair_SKpm · 2022-08-21

**Recommendation:** Accept
**Confidence:** Certain

**Metareview:**

The paper addresses an important question regarding the trajectories of SGD in high-dimensional settings. The theoretical derivations of the paper builds on top of prior works but nonetheless is sound. Most reviewers agree that the paper advances the knowledge in this area.

**Award:**

No

---

### Decision · Program_Chairs · 2022-09-14

Accept